# GROUNDING AS FEEDBACK: VISION-LANGUAGE ALIGNMENT VIA SAMPLING-BASED VISUAL PROJECTION

## ABSTRACT

Vision-Language Models (VLMs) have demonstrated remarkable potential in integrating visual and linguistic information, but frequently produce text that is unfaithful to the visual world, leading to object hallucinations and inaccurate descriptions that limit their real-world applicability. To address this core problem, we introduce *Sampling-based Visual Projection* (SVP), a framework for feedback-driven self-training that efficiently enhances vision-language alignment. Our key insight is to use a pre-trained grounding model as an expert guide to provide feedback on descriptions generated by the VLM itself. SVP uses this feedback in an iterative loop to score, select, and adapt the VLM on high-quality, feedback-refined samples. This process transfers the spatial reasoning skills of the expert guide into the generalist VLM without requiring new, manually curated text-image pairs or preference annotation. Our experiments show that SVP yields significant gains across a range of tasks, including a 14% average improvement in captioning and a 12% increase in object recall, significantly reduced hallucinations, while maintaining question-answering capabilities. The result is a more robust and reliable VLM, demonstrating that targeted, feedback-driven improvement is a powerful method for enhancing vision-language alignment.

## 1 INTRODUCTION

Vision-Language Models (VLMs (Bordes et al., 2024; Zhang et al., 2024a)) are essential to deploying expert level artificial intelligence, as human intelligence is predominantly multimodal.

Generative VLMs (Li et al., 2024; 2022a; Ye et al., 2024; Chen et al., 2024a) built upon Large Language Models (LLMs) have shown great promises in zero-shot abilities on various downstream vision-linguistic tasks (Fig. 5.(*iv*)), unlocking new multimodal capacities and providing powerful generalization to specialized machine learning models. By learning a mapping between linguistic tokens and visual features, such VLMs enjoy the strong generation capabilities of LLMs (Brown et al., 2020; Touvron et al., 2023) and the understanding of the physical world of computer vision models (Radford et al., 2021; Dosovitskiy et al., 2020).

However, VLMs derived from pretrained backbones are known to be impacted by the hallucinations and biases from LLMs (Sasse et al., 2024; Rahmanzadehgervi et al., 2024). It is fre-

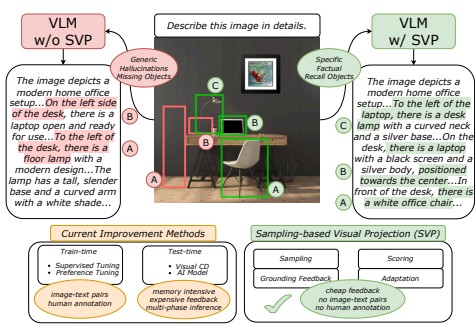

**Figure 1: Improving Vision-Language Alignment.** Vision-language models (VLMs) often produce descriptions lacking specificity and accuracy, frequently hallucinating objects or missing important elements (left). Our *Sampling-based Visual Projection* (SVP) addresses these issues by leveraging self-captioning and grounding feedback. SVP enhances visual-language alignment without requiring human annotations, curated image-text pairs, or expensive AI feedback (right). This leads to models with greater contextual relevance, fewer hallucinations, and enhanced object recall. Appx 11 for details.

quently observed that these VLMs fail to produce text consistent with the visual content (left side Fig. 1), i.e., the generated text describes entities not present in the input image or misses relevant entities altogether, generating content not grounded in the visual input (Collerton et al., 2023; Bai et al., 2024). Addressing these shortcomings is crucial for future deployment of VLMs in high-stakes, real-world applications across the frontiers of scientific discovery (He et al., 2024) and engineering (Picard et al., 2023; Song et al., 2024).

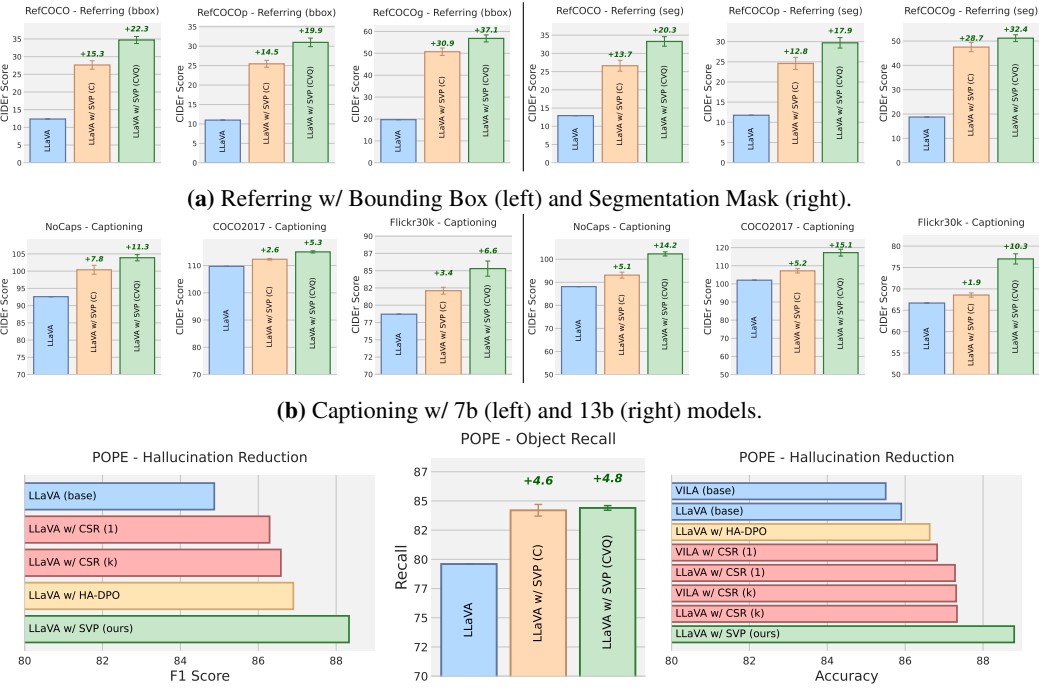

**(a)** Referring w/ Bounding Box (left) and Segmentation Mask (right).

**(b)** Captioning w/ 7b (left) and 13b (right) models.

**(c)** Object Recall and Hallucination Reduction.

**Figure 2: Benchmark Results** comparing base models to our SVP-adapted model on captioning (CIDEr), referring (CIDEr), hallucination control (F1), and object recall (R). Models were adapted using three sets of 1,000 images from the COCO2014 training set, with self-captioning and grounding feedback. Higher scores indicate better performance. SVP demonstrates significant improvements in captioning, referring, object recall, and hallucination reduction.

Researchers have explored various approaches to solve the above problem in VLMs (bottom Fig. 1). Most of these works focus on fine-tuning VLMs with supervised (carefully curated) data to improve grounding (Peng et al., 2024; Beyer et al., 2024; Yuan et al., 2021; You et al., 2023; Zhang et al., 2025) and vision-language alignment (Liu et al., 2024a; Sun et al., 2023). Unfortunately, this data approach tends to be costly and sample-inefficient, requiring large amounts of image-text annotations even for small models to resolve the above stated problem (Yuan et al., 2021).

Preference-based post-training methods (Ouyang et al., 2022; Christiano et al., 2017; Rafailov et al., 2024) as another popular approach align VLM outputs with visual inputs (Zhou et al., 2024a; Sun et al., 2023) but require curated preference pairs (Sun et al., 2023; Favero et al., 2024). And, test-time approaches (Wan et al., 2024; Leng et al., 2024; Favero et al., 2024; Yin et al., 2023) improve grounding without architectural changes, yet their computational demands and model-specific heuristics limit broad applicability.

To address the challenge of improving VLM alignment without costly new annotations, we introduce a feedback-driven self-training framework. We propose to leverage an external, pre-trained grounding model as an expert guide, using its feedback to efficiently enhance the alignment between visual and linguistic modalities in a task-agnostic manner (right side Fig. 1). Our fundamental hypothesis is that better modality alignment is crucial for developing high-performing and reliable VLMs, and that improved alignment will ultimately result in a more effective model.

*Drawing inspiration from human learning, we propose to emulate the way humans efficiently align sensory experiences with language by* grounding *new information in tangible visual examples leveraging feedback* (Hattie & Timperley, 2007; Tenenbaum & Goldring, 1989; Tenenbaum et al., 2011). We hypothesize that spatial and positional reasoning is the key for connecting the low-level visual elements and high-level linguistic representations (Peirce, 2015; Oquab et al., 2014; Vallar, 2007), and that an external visual grounding model (Liu et al., 2023), agnostic to the VLM's shortcomings, can be used as feedback to extract latent information in the models.

Specifically, in this work, we introduce SVP (*Sampling-based Visual Projection*, Fig. 5), an algorithm founded on two core principles: self-training and grounding feedback. The self-training approach (Zelikman et al., 2022; Anthony et al., 2017; Gulcehre et al., 2023) utilizes the model's own outputs

to enhance its performance. And, the grounding feedback provides the VLM with a mechanism to improve its output and select informative samples. Our goal is not to directly build a specialist grounding model, but to *leverage grounding as feedback to elicit latent information in the model*, with the aim of better aligning language and visual representations without the need of new costly image-text annotations (Sun et al., 2023; Peng et al., 2024), preference data (Ouyang et al., 2022; Rafailov et al., 2024), or multi-step inference workflows (Yin et al., 2023; Wan et al., 2024). See Sec A for extended related work.

*It is important to clarify that our objective is not to construct a new generative model with grounding capabilities. Rather, we utilize an existing grounding model as an expert guide, distilling its spatial reasoning skills to unlock latent capabilities within generative vision-language models. While traditional knowledge distillation transfers generative capacities from a teacher to a student, SVP focuses on distilling the geometric alignment between vision and language. This process effectively functions as alignment distillation achieved through feedback-driven self-training.*

`SVP` is a three-step process:

*(i) **Sampling**:* A base VLM generates detailed and comprehensive image descriptions. These descriptions are then processed by a pre-trained grounding model (Liu et al., 2023). The resulting spatially enriched grounding output serves as feedback, conditioning the same VLM to generate text tokens that better align with the visual information (Fig. 5.$(i)$).

*(ii) **Scoring**:* This step employs a scoring and ranking mechanism to select grounded samples that are more informative and better aligned with the visual input (Fig. 5.$(ii)$).

*(iii) **Adaptation**:* The base VLM undergoes adaptation (Hu et al., 2021) on the filtered dataset. Importantly, the grounding information is not shown during the fine-tuning process but is utilized during inference (Fig. 5.$(iii)$).

**Contributions** Our key contributions are:

- We introduce *Sampling-based Visual Projection* (`SVP`), a novel framework that enhances vision-language alignment through iterative feedback-driven self-training, leveraging self-captioning and visual grounding techniques without requiring additional expensive image-text annotations or preference data.

- We develop a principled formulation based on hierarchical sampling, and feedback-driven optimization, where grounding guides the sampling process toward better vision-language alignment. Our design ensures easy applicability across various VLM architectures and scales while providing interpretable vision-language alignment.

- We demonstrate `SVP`'s effectiveness through comprehensive experiments across 10 diverse vision-language benchmarks, including captioning, referring expressions, visual question answering, and hallucination control, using only a small set of curated images and a pretrained grounding model.

## 2 BACKGROUND

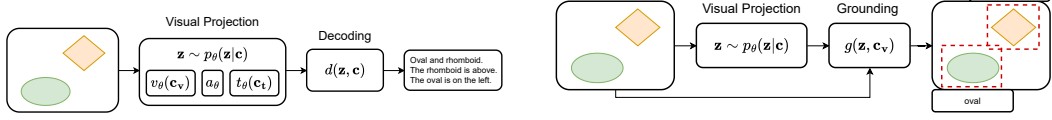

**(a)** Vision-Language Generation    **(b)** Vision-Language Grounding

**Figure 3:** Visual Projection for Generation (left) and Grounding (right).

**Notation** We use $p(\mathbf{x}|\mathbf{c})$ and $p(\mathbf{z}|\mathbf{c})$ to denote auto-regressive distributions, where $\mathbf{c}$ is the conditioning information (image and prompt), $\mathbf{z}$ is a visual projection using grounding feedback, and $\mathbf{x}$ is the task-specific output. These distributions follow $p(\mathbf{x}|\mathbf{c}) = p(\mathbf{x}_T|\mathbf{c})\prod_{t=1}^{T} p(\mathbf{x}_{t-1}|\mathbf{x}_t, \mathbf{c})$, with similar form for $p(\mathbf{z}|\mathbf{c})$. For latent variables, $\mathbf{z}$ represents trajectories $\mathbf{z}_{1:T_z}$. We assume a deterministic output distribution $p(\mathbf{x}|\mathbf{z}, \mathbf{c}) = \delta(\mathbf{x} - d(\mathbf{z}, \mathbf{c}))$, as is common in tokenization-based models. Given context $\mathbf{c} = (\mathbf{c_v}, \mathbf{c_t})$ with visual input $\mathbf{c_v}$ and text prompt $\mathbf{c_t}$, we define a Visual Projection as $p(\mathbf{z}|\mathbf{c})$ and its grounded version as $q(\mathbf{z}|\mathbf{c}, \mathbf{g})$ when conditioning on grounding $\mathbf{g}$. The conditional entropy is $\mathbb{H}[\mathbf{z}|\mathbf{c}] = -\int_{\mathbf{z}} p(\mathbf{z}|\mathbf{c}) \log p(\mathbf{z}|\mathbf{c})$.

**Vision-Language Models** Generative VLMs are multimodal systems processing both text and images. LLaVA-like architectures (Fig. 3a, left) integrate a visual encoder $v_\theta(\mathbf{c_v})$, text encoder $t_\theta(\mathbf{c_t})$, visual-text alignment adapter $a_\theta$, and large language model. The model $p_\theta$ generates token trajectories $\mathbf{z}$ from conditioning $\mathbf{c}$ for various downstream tasks. These systems undergo three training phases:

multimodal pre-training, visual-text alignment, and instruction tuning (Zhu et al., 2023; Liu et al., 2024b; Li et al., 2022a), enabling broad cross-modal capabilities.

**Vision-Language Grounding** Grounding links language descriptions to spatial regions in images. A grounding model $g(\mathbf{z}, \mathbf{c_v})$ processes visual $\mathbf{c_v}$ and textual $\mathbf{z}$ inputs to produce open-set detection labels and bounding boxes (Fig. 3b, right). While traditional object detection uses fixed-class classification, modern approaches like GLIP (Li et al., 2022b) and GroundingDINO (Liu et al., 2023) reframe detection as text-guided grounding. This flexibility enables broader applications in detection and spatial understanding tasks.

## 3 METHOD

**Table 1:** Overview of symbols and components in the SVP framework, detailing input modalities, grounding mechanisms, and model outputs.

| Symbol | Meaning | Description |
|---|---|---|
| $\mathbf{c} : (\mathbf{c}_v, \mathbf{c}_t)$ | Input (image + prompt) | COCO image + "*Describe this image in details...*" |
| $\mathbf{z}_p$ | Base caption (without grounding) | "*A desk with a lamp and laptop....*" |
| $\mathbf{g}$ | Grounding feedback | `lamp [0.33,0.47,0.16,0.23]`, `laptop [0.48,...]` |
| $\mathbf{z}$ | Guided caption (with grounding) | "*A silver desk lamp on the left, a black laptop in the center...*" |
| $p_\theta(\mathbf{z}_p \mid \mathbf{c})$ | Base Vision-Language Model | Input: image; Output: self-generated caption |
| $g(\mathbf{z}_p, \mathbf{c}_v)$ | Grounding Model | Input: image + base caption; Output: grounding (noun phrases + bounding boxes) |
| $q(\mathbf{z} \mid \mathbf{c}, \mathbf{g})$ | Guided Vision-Language Model | Input: image + grounding; Output: self-generated improved caption |

---

**Algorithm 1** SVP Algorithm

---

**Require:** Base VLM $p_\theta$, grounding model $g$, images $\mathcal{C}$, $K$ samples/image
1: $\mathcal{D} \leftarrow \emptyset$     ▷ Self-Training Dataset
2: **for** each image $\mathbf{c} \in \mathcal{C}$ **do**
3:     **for** $k = 1$ to $K$ **do**
4:         $\mathbf{z}_p \sim p_\theta(\cdot|\mathbf{c})$     ▷ Base Sample
5:         $\mathbf{g} \leftarrow g(\mathbf{z}_p, \mathbf{c})$     ▷ Grounding
6:         $\mathbf{z} \sim q(\cdot|\mathbf{c}, \mathbf{g})$     ▷ Guided Sample
7:         $s \leftarrow S(\mathbf{z}, p_\theta, q)$     ▷ Scoring, Eq. 4-6
8:     **end for**
9:     $\mathcal{D} \leftarrow \mathcal{D} \cup \texttt{TopK}(\mathbf{z}, s)$     ▷ Select best
10: **end for**
11: Fine-tune $p_\theta$ on $\mathcal{D}$     ▷ Adaptation, Eq. 7
12: **return** $p_{\theta'}$

---

We present *Sampling-based Visual Projection* (SVP), a framework designed to improve vision-language alignment through guided self-training. SVP draws inspiration from self-training iterative techniques for reasoning in language models (Zelikman et al., 2022; 2024; Gulcehre et al., 2023) and sampling in latent variable models (Jordan et al., 1999; Hoffman et al., 2024).

The process involves:

*(i)* **Sampling**, where a base VLM generates initial captions that are then refined using feedback from a pre-trained grounding model;

*(ii)* **Scoring**, where we identify the most informative and well-aligned captions by measuring the difference between the guided and original model outputs; and

*(iii)* **Adaptation**, where we fine-tune the base VLM on this curated set of high-quality, self-generated data.

In our experiments, we evaluate two variants of this framework. The primary variant, SVP (C), uses only these self-generated captions for fine-tuning. A second variant, SVP (CVQ), augments this data with visual queries from the VLM's original training set to ensure that its question-answering capabilities are preserved during adaptation.

The core idea of SVP is to generate a task-agnostic language-based representation $\mathbf{z}$, referred to as Visual Projection (VP), for the visual input $\mathbf{c}$.

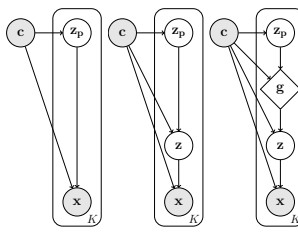

**Figure 4:** Graphical Models for the sampling processes. Left: standard sampling. Center: hierarchical sampling. Right: hierarchical sampling with internal structure.

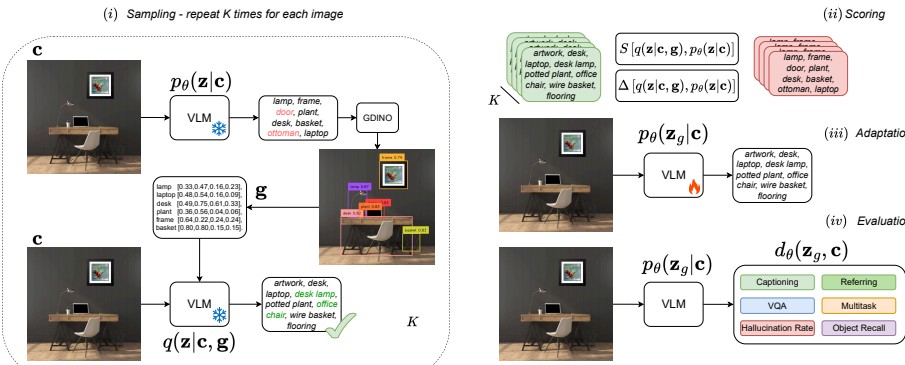

**Figure 5: SVP Overview.** Our framework operates as a nested loop for guided self-training, consisting of a data-generation and model adaptation loops. *(i) Sampling*: For each image $\mathbf{c}$, we generate $K$ samples. First, the base VLM $p_\theta(\mathbf{z}|\mathbf{c})$ generates a draft caption. This caption's text is fed to an expert grounding model (GDINO) to produce spatial feedback $\mathbf{g} \leftarrow g(\mathbf{z}, \mathbf{c})$ (e.g., bounding boxes). This grounding information is provided as text and used as new context to prompt the VLM $q(\mathbf{z}|\mathbf{c}, \mathbf{g})$, which generates a refined, grounded caption. *(ii) Scoring*: These $K$ refined samples are scored to quantify the improvement from the grounding feedback. The top-k of high-scoring, well-aligned samples are selected to build a new training dataset. *(iii) Adaptation*: The base VLM's parameters are then fine-tuned (adapted) using LoRA on this newly curated dataset. *(iv) Evaluation*: Finally, the adapted VLM is evaluated on ten benchmarks across six tasks to measure its improved alignment and reduced hallucinations. Full VLM output in Appx Fig. 11. Prompt structure in Appx G.

These VPs function as latent variables or generalized captions, and SVP aims to refine them through self-training iterative methods, strengthening the alignment between vision and language modalities to enhance the base VLM's performance across diverse tasks.

**Problem Formulation** For a VLM represented with a conditional model $p_\theta(\mathbf{x}|\mathbf{c})$, where $\mathbf{c} = (\mathbf{c}_v, \mathbf{c}_t)$ contains visual input and optional text prompt, direct sampling often yields poor alignment between visual and textual modalities. To address this, we introduce a visual projection as a latent variable (Fig 4, left):

$$p_\theta(\mathbf{x}, \mathbf{z}_p|\mathbf{c}) = p(\mathbf{x}|\mathbf{z}_p, \mathbf{c})p_\theta(\mathbf{z}_p|\mathbf{c}), \tag{1}$$

where $\mathbf{z}_p$ acts as an intermediate visual projection bridging vision and language, similar to chain-of-thought approaches in LLMs. To enhance flexibility and control through ancestral sampling, we extend to a hierarchical structure (Fig 4, center):

$$p_\theta(\mathbf{x}, \mathbf{z}, \mathbf{z}_p|\mathbf{c}) = p(\mathbf{x}|\mathbf{z}, \mathbf{c})p(\mathbf{z}|\mathbf{z}_p, \mathbf{c})p_\theta(\mathbf{z}_p|\mathbf{c}). \tag{2}$$

While this hierarchical structure offers more flexibility, it provides minimal improvement without proper optimization. Simply iterating through the same visual input and refining projections without feedback can lead to model collapse. To address this limitation, we incorporate a grounding model $\mathbf{g} = g(\mathbf{z}_p, \mathbf{c})$ into the hierarchical projection (Fig 4, right):

$$p_\theta^g(\mathbf{x}, \mathbf{z}, \mathbf{z}_p|\mathbf{c}) = p(\mathbf{x}|\mathbf{z}, \mathbf{c})q(\mathbf{z}|g(\mathbf{z}_p, \mathbf{c}), \mathbf{c})p_\theta(\mathbf{z}_p|\mathbf{c}). \tag{3}$$

Here, $q$ is a guided distribution utilizing the grounding model $g$, which provides specialized feedback for vision-language alignment. This feedback mechanism is particularly effective for improving spatial relationships and object attributes, where grounding helps correct the base model's initial predictions. The discrepancy between base model predictions and grounded outputs serves as a valuable signal for enhancing vision-language alignment, especially in cases where grounding information conflicts with initial model predictions.

**Sampling** We implement a guided three-step sampling process to generate improved visual projections: *(i.1)* Prior Sampling, where we generate initial projections $\mathbf{z}_p \sim p_\theta(\mathbf{z}_p|\mathbf{c})$ from the base model. $\mathbf{z}_p$ acts as a draft used solely to obtain feedback. *(i.2)* Grounding, where we apply the grounding model to obtain feedback $\mathbf{g} \leftarrow g(\mathbf{z}_p, \mathbf{c})$; and *(i.3)* Guided Sampling, where we generate guided visual projections $\mathbf{z} \sim q(\mathbf{z}|g(\mathbf{z}_p, \mathbf{c}), \mathbf{c})$. This process repeats $K$ times for each visual input $\mathbf{c}$.

For each guided sample, we evaluate the guided distribution $q(\mathbf{z}|\mathbf{c}, \mathbf{g})$ with grounding feedback $\mathbf{g}$ and the prior distribution $p_\theta(\mathbf{z}|\mathbf{c})$ using the base model. This computation allows us to quantify grounding effects by comparing guided and prior distributions token-wise over the vocabulary,

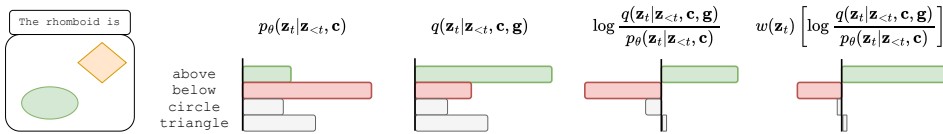

**Figure 6: Visualization of prior and guided distribution** for token $t$ over vocabulary $V = \{\texttt{above}, \texttt{below}, \texttt{circle}, \texttt{rhomboid}\}$. The base model $p_\theta$ incorrectly predicts "$\texttt{below}$" for the circle-rhomboid spatial relationship. With grounding feedback, $q$ correctly assigns higher likelihood to "$\texttt{above}$". Using log-ratio and re-weighting with $w(\mathbf{z}_t) \propto q(\mathbf{z}_t|\mathbf{z}_{<t}, \mathbf{c}, \mathbf{g})$ emphasizes grounding-relevant tokens while down-weighting tokens with similar likelihoods in both distributions.

revealing how visual context influences model predictions. For practical implementation, we convert visual grounding to textual form and include it in the prompt as context, rather than using direct visual representation. The complete prompt structure and examples are detailed in Appx G.

**Scoring** We evaluate sample quality by viewing alignment as a feedback-driven process inspired by policy optimization (Rafailov et al., 2024; Peters & Schaal, 2007; Go et al., 2023). We define a scoring function[1] that measures the *alignment gap* between the guided and prior distributions:

$$S(\mathbf{z}) \propto \log q(\mathbf{z}|\mathbf{c}, \mathbf{g}) - \log p_\theta(\mathbf{z}|\mathbf{c}), \quad \mathbf{z} \sim q(\mathbf{z}|\mathbf{c}, \mathbf{g}). \tag{4}$$

This score quantifies the effect of grounded visual projection on the model. When grounding provides no additional information, $q(\mathbf{z}|\mathbf{g}, \mathbf{c}) \approx p(\mathbf{z}|\mathbf{z}_p, \mathbf{c})$, and Eq. 3 reduces to 1. The score approximates the one sample $\mathbb{KL}$ divergence between $q$ and $p_\theta$. Low values indicate token trajectories well-known to the base model, while high values reveal surprising trajectories that offer learning opportunities. As shown in Fig. 6, the guided distribution $q$ helps correct misaligned predictions of the base model. We implement two scoring approaches. First, a log-ratio scoring:

$$S(q, p)_{\mathbf{z}} = \sum_{t=1}^{T} \sum_{v=1}^{V} w_{v,t} \left[ \log q_{v,t} - \log p_{\theta_{v,t}} \right] \tag{5}$$

where $w_{v,t} \propto q(\mathbf{z}_t|\mathbf{z}_{<t}, \mathbf{c}, \mathbf{g})$ over-emphasizes grounding-relevant tokens. Second, a weighted-difference scoring:

$$\Delta(q, p)_{\mathbf{z}} = \sum_{t=1}^{T} \sum_{v=1}^{V} w^q_{v,t} \log q_{v,t} - \sum_{t=1}^{T} \sum_{v=1}^{V} w^p_{v,t} \log p_{\theta_{v,t}} \tag{6}$$

The weighted-difference score (Settles, 2009) is inspired by the fact that grounding should reduces prediction uncertainty: $\mathbb{H}[\mathbf{z}|\mathbf{c}, \mathbf{g}] < \mathbb{H}[\mathbf{z}|\mathbf{c}]$. Both scoring methods provide similar signals for grounding and diversity (correlation analysis in Appx 24a). Importantly, generic surprise alone (pure exploration) does not enhance vision-language alignment. Our hypothesis is that informative grounding conditioning makes surprising instances statistically valuable for learning and alignment.

**Adaptation** Inspired by re-weighted regression (Peng et al., 2019) and off-policy policy optimization (Roux et al., 2025; Ahmadian et al., 2024; Gulcehre et al., 2023), we propose an iterative optimization where $q(\mathbf{z}|\mathbf{g}, \mathbf{c})$ serves as a behavioral policy providing high-quality demonstrations, while $p_\theta(\mathbf{z}|\mathbf{c})$ is our target model. We maximize:

$$\tilde{\mathcal{F}}(\mathbf{c}; \theta) = |k(\mathbf{c})|^{-1} \sum_{i=1}^{K} \left[ \mathbb{1}\{\mathbf{z}^i : S(q(\mathbf{z}^i|\mathbf{c}, \mathbf{g}), p_\theta(\mathbf{z}^i|\mathbf{c})) \geq S_{k(\mathbf{c})}\} \right]_{\text{sg}} \log p_\theta(\mathbf{z}^i|\mathbf{c}) \tag{7}$$

where $S_{k(\mathbf{c})}$ is the $k$-th highest score among $K$ samples generated for image $\mathbf{c}$ from the guided distribution, $\{\mathbf{z}^i\}_{i=1}^{K} \sim q(\mathbf{z}|\mathbf{c}, \mathbf{g})$. Gradients are stopped (sg) over the scores in the indicator function. This objective can be interpreted as both re-weighted maximum likelihood and greedy off-policy optimization (Appx F). While not necessarily optimal for likelihood or policy trajectories, this approach prioritizes vision-language alignment by selectively optimizing better-aligned samples. The final training loss averages this objective over a batch of visual inputs $\mathbf{c}$.

**Iterative Loop** Our approach uses a nested structure with two distinct loops. The *inner loop* is a data generation process: for each seed image, we sample, guide, and score multiple candidate captions to create a high-quality dataset of well-aligned image-text pairs. The *outer loop* is a model adaptation process: we use this curated dataset to fine-tune the base VLM's parameters. This two-stage design allows the model to first generate its own feedback-driven training data and then learn from it, creating a virtuous cycle of feedback-driven self-training (Fig. 8, 19). This design creates a natural balance between exploration, achieved through guided sampling, and exploitation, driven by model adaptation on the best aligned samples.

---

[1]if we assume that $q$ is the optimal alignment policy, we can write $q(\mathbf{z}|\mathbf{c}, \mathbf{g}) \propto p_\theta(\mathbf{z}|\mathbf{c}) \exp(S(\mathbf{z})/w)$

# 4 EXPERIMENTS

**Table 2: Hallucination Mitigation - F1** scores on POPE benchmark comparing LLaVA variants across adversarial, popular, random, and overall splits. Hallucination avoidance is influenced by model size, fine-tuning approach, encoder selection, and SVP adaptation. See E.9 for analysis of model scaling effects.

| Model | Size | $v_\theta$ | $t_\theta$ | POPE (F1 score ↑) adv | pop | random | all |
|---|---|---|---|---|---|---|---|
| LLaVA (Liu et al., 2024c) | 7b | CLIP | Vicuna | 72.0 | 75.3 | 80.7 | 76.0 |
| LLaVA-SFT⁺ (Sun et al., 2023) | 7b | CLIP | Vicuna | 80.1 | 82.4 | 85.5 | 82.7 |
| LLaVA-RLHF (Sun et al., 2023) | 7b | CLIP | Vicuna | 79.5 | 81.8 | 83.3 | 81.5 |
| LLaVA (Liu et al., 2024c) | 13b | CLIP | Vicuna | 74.4 | 78.2 | 78.8 | 77.1 |
| LLaVA-SFT⁺ (Sun et al., 2023) | 13b | CLIP | Vicuna | 81.1 | 82.6 | 84.8 | 82.8 |
| LLaVA-RLHF (Sun et al., 2023) | 13b | CLIP | Vicuna | 80.5 | 81.8 | 83.5 | 81.9 |
| LLaVA-NeXT-DPO (Liu et al., 2024b) | 7b | CLIP | Qwen2 | 83.43 | 83.78 | 84.73 | 83.98 |
| LLaVA-OV-DPO (Li et al., 2024) | 7b | SigLIP | Qwen2 | 85.12 | 86.24 | 87.37 | 86.24 |
| LLaVA-HA-DPO (Zhao et al., 2023) | 7b | CLIP | Vicuna | 82.54 | 87.89 | **90.25** | 86.90 |
| LLaVA-1.5 (Liu et al., 2024a) | 13b | CLIP | Vicuna | 84.53 | 86.31 | 87.17 | 86.00 |
| LLaVA-1.5 w/ SVP | 13b | CLIP | Vicuna | 84.66 | 86.84 | 87.44 | 86.31 |
| LLaVA-1.6 (Liu et al., 2024b) | 7b | CLIP | Mistral | 85.43 | 86.87 | 88.05 | 86.73 |
| LLaVA-1.6 w/ SVP | 7b | CLIP | Mistral | **85.93** | **89.04** | 90.02 | **88.33** |
| LLaVA-1.6 (Liu et al., 2024b) | 13b | CLIP | Vicuna | 85.17 | 86.36 | 87.20 | 86.24 |
| LLaVA-1.6 w/ SVP | 13b | CLIP | Vicuna | 85.15 | 87.50 | 89.23 | 87.30 |
| LLaVA-OV (Li et al., 2024) | 0.5b | SigLIP | Qwen2 | 82.28 | 83.19 | 83.89 | 83.12 |
| LLaVA-OV w/ SVP | 0.5b | SigLIP | Qwen2 | 83.45 | 84.70 | 85.46 | 84.53 |
| *Bigger VLMs* | | | | | | | |
| LLaVA-1.6 (Liu et al., 2024b) | 34b | CLIP | Yi-2 | - | - | - | 87.7 |
| InternVL (Chen et al., 2024b) | 19b | IViT | Vicuna | - | - | - | 87.6 |
| InternVL-1.2 (Chen et al., 2024b) | 40b | IViT | Yi-2 | - | - | - | 88.0 |
| InternVL-1.2⁺ (Chen et al., 2024b) | 40b | IViT | Yi-2 | - | - | - | 88.7 |
| VILA-1.5 (Lin et al., 2024) | 8b | SigLIP | LLaMA3 | - | - | - | 85.6 |
| VILA-1.5 (Lin et al., 2024) | 8b | SigLIP | Vicuna | - | - | - | 86.3 |
| VILA-1.5 (Lin et al., 2024) | 40b | IViT | Yi2 | - | - | - | 87.3 |
| VILA-1.5-AWQ (Lin et al., 2024) | 40b | IViT | Yi2 | - | - | - | 88.2 |

**Table 3: Hallucination Mitigation - Accuracy** across VLMs using fine-tuning, train-time, and test-time adaptation approaches. Size (Eff) indicates total parameters for multi-phase inference, e.g., Woodpecker (Wp) (Yin et al., 2023) requires multiple models for response processing. Higher is better.

| Model | Size (Eff) | $v_\theta$ | $t_\theta$ | POPE (Acc score ↑) adv | pop | random |
|---|---|---|---|---|---|---|
| *Fine-tuning* | | | | | | |
| InstructBLIP (Dai et al., 2023) | 7b | ViT | FlanT5 | 72.1 | 82.7 | 88.6 |
| LLaVA-SFT⁺ (Sun et al., 2023) | 7b | CLIP | Vicuna | 80.2 | 82.9 | 86.1 |
| mPLUG-Owl2 (Ye et al., 2024) | 8b | ViT | LLaMA2 | 84.1 | 86.2 | 88.3 |
| InstructBLIP (Dai et al., 2023) | 13b | ViT | Vicuna | 74.5 | 81.4 | 88.7 |
| LLaVA-SFT⁺ (Sun et al., 2023) | 13b | CLIP | Vicuna | 82.3 | 83.9 | 85.2 |
| *Test-time adaptation* | | | | | | |
| QwenVL w/ VCD (Leng et al., 2024) | 7b (14b) | CLIP | Vicuna | 84.3 | 87.1 | 88.6 |
| LLaVA w/ M3ID (Favero et al., 2024) | 7b (14b) | CLIP | Vicuna | 65.8 | 69.3 | 76.0 |
| Otter w/ Wp (Yin et al., 2023) | 7b (14b+) | CLIP | LLaMA | 83.0 | 84.3 | 86.7 |
| mPLUG-Owl w/ Wp (Yin et al., 2023) | 7b (14b+) | ViT | LLaMA | 81.0 | 84.1 | 86.3 |
| LLaVA w/ M3ID (Favero et al., 2024) | 13b (26b) | CLIP | Vicuna | 71.3 | 77.0 | 84.3 |
| *Train-time adaptation* | | | | | | |
| LLaVA-M3ID-DPO (Favero et al., 2024) | 7b | CLIP | Vicuna | 68.2 | 73.9 | 81.2 |
| LLaVA-RLHF (Sun et al., 2023) | 7b | CLIP | Vicuna | 80.7 | 83.3 | 84.8 |
| LLaVA-NeXT-DPO (Rafailov et al., 2024) | 7b | CLIP | Qwen2 | 85.2 | 85.6 | 86.6 |
| LLaVA-OV-DPO (Rafailov et al., 2024) | 7b | SigLIP | Qwen2 | 86.3 | 87.5 | 88.7 |
| LLaVA-HA-DPO (Zhao et al., 2023) | 7b | CLIP | Vicuna | 81.5 | 87.9 | **90.5** |
| SeVa (Zhu et al., 2024) | 7b | CLIP | Vicuna | 83.6 | 87.4 | 89.4 |
| LLaVA-M3ID-DPO (Favero et al., 2024) | 13b | CLIP | Vicuna | 73.2 | 79.1 | 85.2 |
| LLaVA-RLHF (Sun et al., 2023) | 13b | CLIP | Vicuna | 82.3 | 83.9 | 85.2 |
| InstructBLIP-HA-DPO (Zhao et al., 2023) | 13b | ViT | Vicuna | 80.7 | 85.8 | 89.8 |
| LLaVA-1.6 (Liu et al., 2024b) | 7b | CLIP | Mistral | 86.4 | 87.9 | 89.2 |
| LLaVA-1.6 w/ SVP | 7b | CLIP | Mistral | 86.2 | **89.6** | **90.6** |
| LLaVA-1.6 (Liu et al., 2024b) | 13b | CLIP | Vicuna | 86.4 | 87.7 | 88.5 |
| LLaVA-1.6 w/ SVP | 13b | CLIP | Vicuna | **86.7** | 88.4 | 89.2 |
| LLaVA-OV | 0.5b | SigLIP | Qwen2 | 84.3 | 85.2 | 86.0 |
| LLaVA-OV w/ SVP | 0.5b | SigLIP | Qwen2 | 85.0 | 86.3 | 87.2 |

## 4.1 EXPERIMENTAL DESIGN

**Base Model Selection** We selected the LLaVA model family (Liu et al., 2024c) to rigorously validate our method, SVP. LLaVA's straightforward architecture exhibits well-defined capability gaps (Table 4), providing a clear baseline to demonstrate SVP's effectiveness in building foundational visual-language skills. Moreover, its transparent and incrementally expanding open-source datasets [2] allow for a controlled analysis of performance gains, free from evaluation set overlaps or confounding variables like proprietary data or extensive reinforcement fine-tuning. To contextualize our results, we provide comprehensive comparisons against existing methods, with a focus on hallucination reduction (Tables 2 and 3).

**Seed Images** We utilize a pretrained GroundingDINO-tiny (GDINO-T) (Liu et al., 2023) as our expert guide for feedback. Our adaptation set consists of $C = 1000$ images randomly sampled from the COCO2014 training set (Lin et al., 2014). We selected the GDINO-T as it was not trained on the COCO dataset. *This design ensures the feedback stems from the guide's generalizable ability to ground the VLM-generated description within the visual context, rather than from memorized annotations.* Furthermore, as these COCO images were part of base LLaVA tuning data, our method isolates the grounding feedback as the sole source of new information for the model, removing confounding factors.

**Baselines** We conduct a comprehensive comparison against various baselines, including models fine-tuned with self-captioning without grounding and preference-based adaptation methods. Our evaluation encompasses a wide range of model scales (.5B, 7B, 8B, 13B, 19B, 40B parameters), architectures (LLaVA-1.5 (Liu et al., 2024a), LLaVA-1.6 (Liu et al., 2024b), LLaVA-OV (Li et al., 2024), VILA (Lin et al., 2024), InternVL (Chen et al., 2024b)), visual encoders (CLIP (Radford et al., 2021), SigLIP (Zhai et al., 2023), ViT (Dosovitskiy et al., 2020)), language encoders (Vicuna (Chiang et al., 2023), Mistral (Jiang et al., 2023), Qwen2 (Yang et al., 2024), Yi-2 (Young et al., 2024)), and scoring mechanisms $S(q, p)$ and $\Delta(q, p)$.

**Implementation Details** Here we detail the core setup for the experiments with LLaVA-1.5/1.6. We implement two SVP variants: SVP (C), using self-generated descriptions, and SVP (CVQ), which adds visual queries (already seen by the base models) to prevent over-specialization. For the sampling loop, we generate $K = 20$ samples for each of $C = 1000$ images, selecting the top-20% with our scoring mechanisms (Eq. 5, 6) to yield 4000/8000 training pairs, a size shown to be effective

---

[2] https://github.com/haotian-liu/LLaVA/blob/main/docs/Data.md

for adaptation (Sun et al., 2023; Zhu et al., 2024). For the adaptation loop, we fine-tune for one epoch on 8-A100 GPUs ($B = 20$) using LoRA (Hu et al., 2021), with $\alpha$ and $r$ scaled to model size ($\alpha = 16, r = 64$ for $\leq$7b; $\alpha = 256, r = 128$ for 13b). Following prior work (Li et al., 2024; Liu et al., 2024b), we run up to 3 SVP iterations. We use normalized *xyxy* boxes and filter degenerate samples ($< 0.5\%$) We feed the output of the grounding model to the VLM as context *without* any post-processing or filtering. Our evaluation uses sample-wise, zero-shot testing without prompt engineering or batching to ensure fair comparison across model variants. More details in Appendix K.

**Metrics** We use the CIDEr score (Vedantam et al., 2015) for captioning and referring tasks; accuracy for VQA and multitasking. F1, Accuracy and Recall for hallucination and object recall. We also consider standard metrics for language translation like BLEU (Papineni et al., 2002), METEOR (Banerjee & Lavie, 2005), and ROUGE (Lin, 2004) scores. We re-compute metrics for LLaVA baselines and variants (1.5, 1.6, OV) up to 13b parameters.

## 4.2 VISION-LANGUAGE BENCHMARKS

**Table 4: Benchmark Performance** across LLaVA variants (7B/13B) with same visual encoder (CLIP) and varying the text encoders (Mistral and Vicuna) evaluated using `lmms-eval` (`lite` split, full MMMU, POPE, and ScienceQA). Results show SVP (C) and SVP (CVQ) improve captioning, referring tasks, and object recall while reducing hallucinations, maintaining strong performance on multitask benchmarks. Higher is better.

| Model | $v_\theta$ | $t_\theta$ | VQA ScienceQA test | GQA test | NoCaps val | Captioning COCO2017 val | Flickr30k test | Referring RefCOCO val | Multitasking MMBench en_dev | MMMU val | Hallucinations POPE (F1) all | POPE (R) all |
|---|---|---|---|---|---|---|---|---|---|---|---|---|
| LLaVA-1.6-7b | CLIP | Mistral | **78.54** | **75.80** | 92.60 | 109.68 | 78.74 | 6.70 | **80.30** | 34.11 | 86.73 | 79.60 |
| w/ SVP (C) | CLIP | Mistral | 77.24 | 73.80 | 100.93 | 112.95 | 83.49 | 18.15 | 77.27 | 36.44 | **88.33** | **84.20** |
| w/ SVP (CVQ) | CLIP | Mistral | 78.40 | 75.10 | **103.95** | **115.02** | **85.31** | **24.74** | 78.03 | **37.44** | 88.25 | 84.41 |
| | | | ↓ 0.54 % | | | ↑ 8.48 % | | ↑ 18.04 | ↑ 3.43 % | | ↑ 3.94 % | |
| LLaVA-1.6-13b | CLIP | Vicuna | 70.30 | **74.60** | 83.89 | 104.21 | 69.86 | **29.71** | **83.33** | 35.22 | 86.24 | 78.13 |
| w/ SVP (C) | CLIP | Vicuna | **74.34** | 74.40 | 87.09 | 111.09 | 71.43 | 28.93 | 81.06 | **36.33** | 87.44 | 81.20 |
| w/ SVP (CVQ) | CLIP | Vicuna | 68.49 | 73.20 | **100.26** | **122.03** | **85.32** | 27.20 | 78.03 | 35.66 | 87.68 | 82.53 |
| | | | ↑ 2.65 % | | | ↑ 19.58 % | | ↓ 0.78 | ↑ 0.12 % | | ↑ 3.65 % | |

**Datasets** We evaluate SVP across six tasks using ten standard VLM benchmarks: COCO2017 (Lin et al., 2014), NoCaps (Agrawal et al., 2019), and Flickr30k (Plummer et al., 2015) for captioning; RefCOCO variants (Kazemzadeh et al., 2014) for referring expression generation; ScienceQA (Saikh et al., 2022) and GQA (Hudson & Manning, 2019) for VQA; MMBench (Liu et al., 2025) and MMMU (Yue et al., 2024) for multitasking; and POPE (Li et al., 2023b) for hallucination assessment. Following lmms-eval (Zhang et al., 2024b), we use both full and lite evaluation sets for captioning and VQA tasks to demonstrate result stability across sample sizes. For MMMU, POPE, and all RefCOCO variants, we use the complete evaluation sets.

**General Results** Across the 10 datasets and 6 tasks evaluated (Fig. 2 and Table 4), our method demonstrates significant improvements in captioning, referring expression generation, hallucination control, and object recall. We maintain comparable or improved performance on multitasking benchmarks and VQA tasks. The most substantial gains appear in captioning, with nearly 20% improvement, while performance remains stable even in challenging tasks like visual question answering.

The impact of SVP is especially dramatic for models with initial weaknesses in specific tasks. For instance, when applied to LLaVA with Mistral, which originally shows poor referring capabilities, SVP improves referring expression generation performance by a factor of three (Fig. 2a). The preservation of VQA performance is particularly significant, as it indicates that our method *enhances vision-language alignment without compromising existing capabilities* or requiring task-specific knowledge injection. This balanced improvement highlights SVP's ability to strengthen fundamental cross-modal understanding while maintaining the model's broader base capabilities.

**Captioning Tasks** We conducted extensive captioning experiments using both 7B and 13B model architectures across three standard datasets: COCO2017, Flickr30k, and NoCaps (Fig. 2b).

Our comprehensive evaluation, detailed in Tables 11 and 12, spans four datasets and employs four widely-accepted metrics for assessing language generation and alignment quality. The evaluation encompasses over 80,000 samples, providing robust statistical evidence for our findings.

SVP demonstrates consistent superior performance across all datasets and metrics compared to existing methods. This comprehensive improvement underscores the effective-

ness of our integrated sampling and feedback approach in enhancing image captioning capabilities. More fundamentally, these results validate our core hypothesis: strengthening vision-language alignment serves as a foundational principle for advancing VLM capabilities.

**Referring Tasks** We evaluate model performance on referring expression tasks, which require the VLM to generate descriptions for specific image regions (Fig. 7 and Appx 16). Our analysis compares four model variants: a baseline model, a model tuned without grounding (w/o **g**), a model incorporating visual grounding (w/ SVP (C)), and our full model with both grounding and visual queries (w/ SVP (CVQ)). The results demonstrate that SVP substantially improves performance across all datasets and tasks. Most notably, SVP significantly enhances the base model's ability to understand and describe spatial relationships, particularly in cases where initial performance is poor. In fact, our enhanced models achieve performance levels approaching those of much larger 13B parameter models (Table 4). A key insight emerges from these results: these improvements occur without direct access to grounding information (bounding boxes) during the adaptation phase. The grounding conditioning **g** is utilized only during the inner-loop sampling to construct $q(\mathbf{z}|\mathbf{c}, \mathbf{g})$

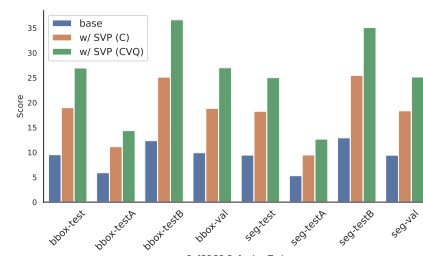

**Figure 7: Referring Expression Generation** on RefCOCO comparing base LLaVA-1.6-7b versus SVP (C) and SVP (CVQ) variants. CIDEr scores shown for detection (`bbox`) and segmentation (`seg`) on test/validation sets. SVP models outperform baseline without using bounding boxes. Appx 16 for RefCOCO+ and RefCOCOg results.

(Fig. 5.$(iii)$), after which we adapt model parameters $\theta$ using only the refined visual projections $\mathbf{z}$. This success in improving referring abilities without explicit grounding supervision suggests that enhanced modality alignment naturally leads to better spatial understanding in VLMs.

**Table 5: Component Ablation.** Performance comparison of LLaVA-1.6-7b variants after one adaptation iteration: base model, fine-tuning without feedback, sampling with grounding (no scoring), grounding with scoring, and full SVP (grounding, scoring, visual queries). Results provide evidence of the importance of the SVP's components for model performance.

| Model | Grounding | Scoring | VQ | RefCOCO | Flickr30k | MMMU | POPE |
|-------|-----------|---------|-----|---------|-----------|------|------|
| LLaVA | - | - | - | 6.70 | 78.74 | 34.11 | 86.73 |
| w/o SVP | ✗ | ✗ | ✗ | 3.01 | 79.03 | 35.55 | 87.21 |
| w/ SVP | ✓ | ✗ | ✗ | 9.98 | 78.67 | 35.77 | 86.92 |
| w/ SVP | ✓ | ✓ | ✗ | 18.15 | 83.49 | 36.44 | **88.33** |
| w/ SVP | ✓ | ✓ | ✓ | **24.74** | **85.31** | **37.44** | 88.25 |

**Table 6: Preference Ablation**. Comparison between SVP and DPO (Rafailov et al., 2024) for LLaVA-7b-OV with Qwen2 language model (higher is better). While DPO requires a learned reward model or human preference pairs, SVP uses only a small grounding model for feedback ($C = 2000$, $K = 10$, top-10%). Results show that DPO, while effective for general preference alignment, does not achieve the visual-language alignment gains of SVP.

| Model | Samples | SciQA | NoCaps | RefCOCO | MMBench | POPE |
|-------|---------|-------|--------|---------|---------|------|
| w/ DPO | $\geq 9.4k$ | 79.25 | 112.51 | 13.60 | 85.60 | **86.24** |
| w/ SVP (C) | $\approx 2k$ | **83.89** | **120.23** | **15.75** | **86.36** | 85.78 |

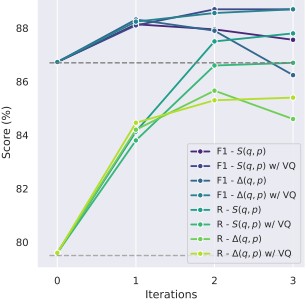

**Figure 8: Iterations.** Hallucinations (F1) and object recall (R) results with $C = 1000$ for $I = 3$ iterations with $\Delta(q, p)$ and $S(q, p)$ scores.

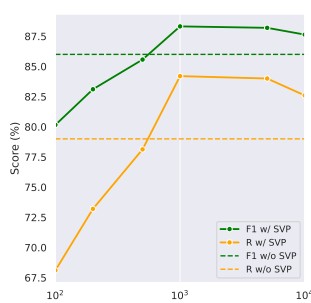

**Figure 9: Sample Size.** Hallucinations (F1) and object recall (R) results with $I = 1$ a single iteration increasing the sample size $C \in (0.1, 0.2, 0.5, 1, 5, 10)k$.

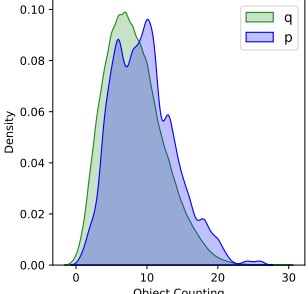

**Figure 10:** Distribution of groundable objects in captions from base model $p_\theta(\mathbf{z}|\mathbf{c})$ and grounded model $q(\mathbf{z}|\mathbf{c}, \mathbf{g})$. SVP guided models generate fewer hallucinated objects.

**Hallucination and Object Recall** We evaluate our model's hallucination rate (Tables 2 and 3) and object recall (Figs. 2c and 8), where object recall measures the model's ability to capture visual

elements in its textual output. Our comparison includes HA-DPO (Zhao et al., 2023), the leading DPO (Rafailov et al., 2024) variant for hallucination reduction, and CSR (Zhou et al., 2024b), an iterative self-rewarding VLM mechanism. For CSR, we evaluate both single-iteration performance and the best result across iterations $I \in [1 : 5]$. SVP demonstrates substantial improvements across most model variants on the POPE dataset. With the 7B model, SVP raises the F1 score from 86.7% to 88.3%, achieving performance comparable to models five times larger (E.9).

Similarly, the 13B model shows improvement from 86.2% to 87.5%. Most impressively, when running SVP for three iterations with our scoring mechanism (Eq. 5), object recall improves dramatically from 79% to over 87% (Fig. 8). These results provide strong evidence that enhancing modality alignment through self-captioning and grounding feedback effectively reduces hallucinations without requiring specialized fine-tuning. This validates our core hypothesis while demonstrating SVP's ability to significantly improve the model's factual accuracy and reliability. Notice that naively finetuning a strong VLM on a larger multimodal dataset does not improve hallucination or recall, as shown in Table 13, providing evidence that methods like SVP are necessary to align vision and language.

**Ablations** We conduct comprehensive ablation studies to analyze SVP's components and behavior. First, we examine the individual contributions of grounding, scoring, and visual queries (Table 5). We then investigate the impact of key hyperparameters: the number of iterations $I$ (Fig. 8, Appx 19) and sample size $C$ (Fig. 9). For scoring mechanisms, we evaluate both $\Delta(q, p)$ and $S(q, p)$ on the full captioning benchmark (Tables 11 and 12). We also compare SVP against DPO using Qwen2 (Yang et al., 2024) as the language model on a subset of our benchmark (Table 6). Additionally, we explore iSVP, a variant designed for inference-time adaptation without parameter tuning (Table 9, Fig. 13, Fig. 14). Finally, we quantify the set of groundable objects for captions generated by guided versus prior distributions (Fig. 10).

**Table 7:** Hallucination (F1) and Recall (R) using Qwen2-VL7b and Qwen2.5-VL-7b on the POPE dataset. Naively scaling the multimodal finetuning dataset does not solve visual-language alignment in powerful VLMs.

| Name | Multimodal Samples | F1 (all - 9k) | F1 (adv - 3k) | F1 (pop - 3k) | F1 (random - 3k) | Recall (all - 9k) |
|------|--------------------|---------------|---------------|---------------|------------------|-------------------|
| Qwen2-VL-7b | 100M | 87.8 | **85.8** | 87.9 | 89.8 | 83.4 |
| Qwen2.5-VL-7b | 100M++ | 86.2 | 85.1 | 86.1 | 87.1 | 77.9 |
| LLaVA-1.6-7b w/ SVP | 800K | **88.3** | **85.9** | **89.1** | **90.1** | **84.4** |

**Comparison with Qwen-VL** Table 7 compares Qwen2-VL and Qwen2.5-VL with LLaVA-1.6-7b w/ SVP on POPE. Despite using roughly two orders of magnitude fewer multimodal samples (800K vs. 100M+), LLaVA-1.6-7b w/ SVP achieves the best hallucination performance, with the highest F1 and recall across all subsets. This indicates that simply scaling multimodal finetuning data is insufficient to fix vision-language misalignment in strong VLMs, whereas targeted adaptation such as SVP is more effective.

As shown in Table 8, applying SVP to Qwen2.5-VL-7b similarly yields substantial gains in both F1 and recall without additional large-scale pretraining, providing evidence that SVP is a general mechanism to improve vision-language alignment.

## 5 CONCLUSION AND LIMITATIONS

We present SVP, a novel method that leverages self-captioning and grounding feedback to enhance VLMs without requiring additional annotations. Our approach significantly improves captioning quality, referring expression generation, hallucination control, and object recall while maintaining strong performance on VQA and multitasking benchmarks. These results demonstrate SVP's potential to unlock latent VLM capabilities, advancing toward more robust real-world applications, like 3D designs and video generation.

**Table 8:** Hallucination (F1) and Recall (R) using Qwen2.5-VL-7b w/o and w/ SVP adaptation. SVP can improve vision-language alignment in powerful VLM models.

| Name | F1 | Recall |
|------|------|--------|
| Qwen2.5-VL-7b | 86.2 | 77.9 |
| Qwen2.5-VL-7b w/ SVP | **89.1** | **83.7** |

**Limitations** However, SVP has limitations: it requires VLMs capable of in-context learning, relies multiple samples per input, and feedback depends on the availability of a pre-trained grounding model. The method may not benefit tasks without spatial components or those requiring specialized knowledge, such as VQA. Additionally, without injecting new information, its applicability to knowledge-intensive tasks remains uncertain without external data.

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

## REPRODUCIBILITY STATEMENT

We provide a detailed account of our methodology and experimental setup. The core SVP framework is described in the Method Section, with explicit pseudo-code for our algorithm and its scoring variants available in Appendix D. Our complete experimental design, including the selection of base models, datasets, and metrics, is detailed in the Experiments Section and Appendix K. Datasets details are in Table 17. A comprehensive table of all hyperparameters used for sampling and training is provided in Table 18. Furthermore, the exact prompts used to generate our data are included in Appendix G. The baselines and experiments are implemented on top of the `LLaVA` [3] and `LLaVA-NeXT` [4] codebases. We use `lmms-eval` [5] for evaluation.

## LARGE LANGUAGE MODELS USAGE

In preparing this manuscript, we utilized Large Language Models (LLMs) solely for the purpose of polishing the text. Their use was strictly confined to improving grammar, clarity, and overall English fluency. The core intellectual contributions of this paper - including the initial ideation, the design of the methodology, the execution of experiments, and the analysis of results - were conceived and developed entirely by the human authors.

---

[3] `https://github.com/haotian-liu/LLaVA`
[4] `https://github.com/LLaVA-VL/LLaVA-NeXT`
[5] `https://github.com/EvolvingLMMs-Lab/lmms-eval`

CONTENTS

# A  RELATED WORK

**Improving Vision-Language Models**    Researchers have investigated explicit grounding in VLMs, primarily to address hallucinations (Wan et al., 2024; Favero et al., 2024), with less focus on developing general paradigms for improving vision-language alignment. A common strategy involves incorporating grounding annotations into training data (Peng et al., 2024) for vision-centric VLMs (Beyer et al., 2024; Yuan et al., 2021; You et al., 2023; Zhang et al., 2025).

However, this annotation process is costly, time-consuming, and prone to errors. For instance, directly generating coordinate tokens as output is sample-inefficient, requiring billions of annotations even for small VLMs to develop a competitive detector (Yuan et al., 2021). While explicit supervision during fine-tuning can enhance alignment between visual and linguistic representations (Liu et al., 2024a; Sun et al., 2023), these train-time methods necessitate large amounts of high-quality visual-text data and are resource-intensive to scale with human annotations.

Train-time techniques like Reinforcement Learning from Human Feedback (RLHF (Christiano et al., 2017; Ouyang et al., 2022)) and Direct Preference Optimization (DPO (Rafailov et al., 2024)), primarily used for aligning LLMs with human preferences, can be adapted to align VLM text outputs with visual inputs (Zhou et al., 2024a; Sun et al., 2023; Wang et al., 2024). These approaches incorporate feedback and preferences during post-training but are limited by the need for reward signals (Sun et al., 2023), curated preference pairs (Zhu et al., 2024; Zhou et al., 2024a), and AI feedback (Wang et al., 2024).

Test-time methods (Wan et al., 2024), such as Visual Contrastive Decoding (Leng et al., 2024) and Multi-Modal Mutual-Information Decoding (Favero et al., 2024), aim to improve grounding at inference by leveraging differences between vision-conditional and unconditional models, without altering the model architecture or training. Woodpecker (Yin et al., 2023) proposes a five-step inference procedure to mitigate hallucination. While somewhat effective, these methods often require memory-intensive and computationally expensive inference, as well as model-specific heuristics, which limits their generalization and usability.

**Grounding in Vision-Language Models**    Visual grounding can be conceptualized as the dual of text-image alignment. When viewed as a mechanism to elicit and organize information within Vision-Language Models (VLMs), it represents a form of alignment between visual and textual modalities, encompassing both representation and generation aspects.

The concept of grounding has deep roots in cognitive sciences (Kiefer & Barsalou, 2013; Barsalou, 2008; Anderson, 2010; Glenberg & Kaschak, 2002). In the context of computer vision, visual grounding can be seen as an extension of the classic closed-set detection problem (Girshick et al., 2014; Carion et al., 2020; Redmon, 2016; Zhang et al., 2022).

Traditional object detection tasks involve regressing bounding box coordinates and assigning class labels to regions within an input image. While leveraging curated benchmark datasets (Lin et al., 2014) has led to rapid improvements in precision and speed, this approach has been constrained by predefined class sets. Scaling to a larger number of classes and adapting to varying detection granularities have proven challenging (Gupta et al., 2019; Dave et al., 2021).

Visual grounding inverts this paradigm by using the set of classes as input and employing a vision-language model to assign bounding boxes to each element in the input. This concept can be further generalized to accommodate captions, descriptions, and various forms of textual input. Contrastive models such as GLIP (Li et al., 2022b) and GroundingDINO (Liu et al., 2023) offer flexible, generalized detection models that enhance spatial understanding (Yin et al., 2023) and serve as foundations for a wide range of tasks. Moreover, auto-regressive VLMs have been developed to perform grounding and referring tasks (Yuan et al., 2021; You et al., 2023; Peng et al., 2024; Tong et al., 2024), further expanding the capabilities of these models in bridging visual and linguistic information.

**Self-Training in Vision and Language Models**    Self-training and self-play autonomous learners have been a long standing goal of the AI field (Schmidhuber, 1987; 1999). In the context of Vision-Language Models (VLMs), self-training can be conceptualized as a form of self-play (Silver et al., 2016; 2017), where the model enhances its performance through sampling and external feedback mechanisms (Anthony et al., 2017). The advent of Large Language Models (LLMs) (Brown et al., 2020; Achiam et al., 2023) has necessitated novel approaches to self-training, given the challenges in defining explicit feedback for natural language trajectories.

Reinforcement Learning from Human Feedback (RLHF) (Ouyang et al., 2022) and Reinforcement Learning from AI Feedback (RLAIF) (Bai et al., 2022) have emerged as prominent mechanisms. These methods score samples from the base model and select preferred outputs based on specific criteria, such as human preferences in chat interactions. Both approaches learn preference or reward models from human or AI feedback, and these concepts have been successfully adapted to VLMs (Sun et al., 2023; Favero et al., 2024).

Further developments in this field include using rewards for ranking (Dong et al., 2023) and implicitly specifying preferences through positive and negative pairs (Rafailov et al., 2024). Alignment can also be achieved through AI distillation (Sudalairaj et al., 2024; Chiang et al., 2023) and self-refinement techniques (Kang et al., 2024b;a; Wang et al., 2022; Sun et al., 2024).

A recent class of algorithms for self-training involves iterative processes (Zelikman et al., 2022; 2024; Anthony et al., 2017; Gulcehre et al., 2023) that leverage feedback to enhance downstream tasks and reasoning chains (Wei et al., 2022) in LLMs. Moreover, feedback can be incorporated at inference time (Madaan et al., 2024) and even utilize the model's own capabilities as evaluator (Yao et al., 2022; Shinn et al., 2024). These methods can be seen as instantiating meta-learning algorithms.

Meta-learning (Schaul & Schmidhuber, 2010; Hospedales et al., 2020; Finn et al., 2017), often described as learning to learn, plays a crucial role in the development of self-improving AI systems. This approach aims to create models that can adapt quickly to new tasks by leveraging knowledge from previously learned tasks (Schmidhuber, 1987; 1999). In the context of VLMs and LLMs, meta-learning techniques have been explored to enhance model adaptability and generalization across diverse domains. For instance, few-shot in-context learning methods (Brown et al., 2020; Xie et al., 2021; Akyürek et al., 2022) demonstrate how large models can rapidly adapt to new tasks with minimal task-specific examples.

# B  Sampling-based Visual Projection Workflow

**Instruction:** Please describe the content of this image as detailed as possible.

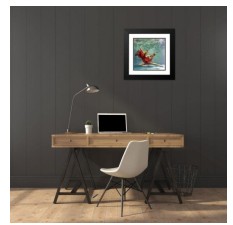 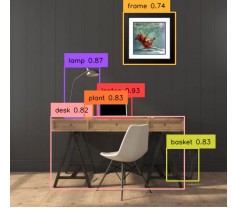

*VLM response:*
The image depicts a modern home office setup. At the center of the scene is a wooden desk with a sleek, minimalist design. The desk features a flat surface with a built-in drawer and a small shelf on the left side. On the right side of the desk, there is a laptop open and ready for use.
Above the desk, mounted on the wall, is a framed artwork of a ballerina in a dynamic pose, adding a touch of elegance and movement to the space. The artwork is encased in a black frame that contrasts with the wall's dark gray color.
To the left of the desk, there is a floor lamp with a modern design, providing additional lighting. The lamp has a tall, slender base and a curved arm with a white shade.
On the desk, there is a small plant in a white pot, adding a touch of greenery and life to the space. The plant is placed on the left side of the desk, near the laptop. Underneath the desk, there is a black wire basket, which is likely used for cable management and storage.

Note: missing chair, missing background.

**Grounded Objects**:   desk (0.82).  laptop (0.93).  artwork (0.64).  frame (0.74).  lamp (0.87).  plant (0.83).  pot (0.68).  basket (0.83).

**Instruction:** Please describe the content of this image as detailed as possible.
**Context:** Use the following context. The list of numbers are bounding boxes for the objects. Check if object and bounding boxes connections are correct. Check if there are additional objects in the image and describe them.
format: cxcywh.
boxes:  lamp [0.33,0.47,0.16,0.23], laptop [0.48,0.54,0.16,0.09], desk [0.49,0.75,0.61,0.33], plant [0.36,0.56,0.04,0.06], frame [0.64,0.22,0.24,0.24], basket [0.80,0.80,0.15,0.15].

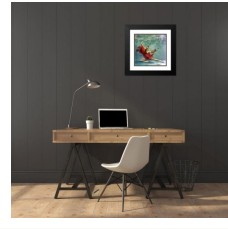 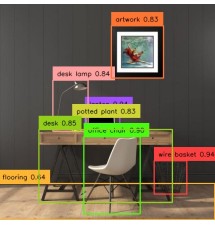

*VLM response:*
The image depicts a modern workspace setup against a dark gray wall with horizontal wood paneling. On the wall, there is a framed artwork featuring a figure in a red dress with a blue background, possibly a painting or a print. Below the artwork, there is a desk with a light wood finish. The desk has a rectangular shape with a flat top and is supported by four black metal legs.
On the desk, there is a laptop with a black screen and a silver body, positioned towards the center. To the left of the laptop, there is a white desk lamp with a curved neck and a silver base, turned off. Next to the lamp, there is a small white potted plant with green leaves.
In front of the desk, there is a white office chair with a modern design, featuring a curved backrest and armrests. The chair is positioned facing the desk, with the seat slightly pulled out.
To the right of the desk, there is a black wire basket with a handle, placed on the floor. The flooring appears to be a light-colored wooden plank. The overall lighting in the room is soft and ambient, suggesting a calm and focused work environment.
**Grounded Objects:**   artwork (0.83).  desk (0.85).  laptop (0.94).  desk lamp (0.84).  potted plant (0.83).  office chair (0.90).  wire basket (0.94).  flooring (0.64).

**Figure 11: Sampling VLMs with and without Grounding Feedback.** Incorporating grounding feedback helps VLMs to focus on factual information and better describe the details in the input image. We use GroundingDINO (Liu et al., 2023), an open-set grounding model, to obtain the conditioning information. When the predicted bounding boxes overlap above a certain threshold, we select the box with the highest score, following a standard non-maximum-suppression approach. By leveraging this grounding feedback, the model is better able to specify the entities and relationships between the objects in the image, leading to an improved parsing of the visual information. This results in more accurate and detailed descriptions, such as identifying a `desk lamp` instead of a `floor lamp`, mentioning an `office chair`, describing the `flooring` in the background, and differentiating between an `artwork` and a simple `frame`, or a `potted plant` and a generic `plant`. More visualizations in J.

## C In-Context Visual Projection and Text-to-Image Alignment

**Table 9:** Text-to-Image alignment scores computed using base VLMs and VLMs with in-context grounding (w/ `iSVP`). Image Text Matching (ITM (Li et al., 2023a)) and ImageReward (Xu et al., 2024) are traditionally used to evaluate AI-generated images from real text prompts. Here we apply these scores to assess AI-generated captions for real images. Although Text-to-Image alignment is not fully indicative of Image-to-Text alignment in this context, these scores provide an additional qualitative measure of caption-image correspondence. Higher is better.

| Model | Size | ITMScore (BLIP2) ↑ | ImageReward ↑ |
|---|---|---|---|
| LLaVA-1.6 | 7b | 0.83 | 0.47 |
| LLaVA-1.6 w/ `iSVP` | 7b | **0.89** | **0.49** |
| LLaVA-1.6 | 13b | 0.82 | 0.44 |
| LLaVA-1.6 w/ `iSVP` | 13b | **0.87** | **0.46** |

A central question in vision-language modeling is how the alignment between visual and textual data impacts text-to-image generation. Visual grounding serves as a key mechanism for structuring this cross-modal information, creating a unified representational space that enables effective multi-modal reasoning.

To explore this relationship, we evaluated text-to-image alignment using `iSVP`, a technique for inference-time adaptation that requires no parameter tuning. In our experiment, we provided a pre-trained grounding model as an inference-time tool to a base model (LLaVA-1.6). We then prompted both the base model and the grounded version (LLaVA-1.6 w/ `iSVP`) to generate captions for 1000 images on the fly, without any fine-tuning. We sample one caption per image, i.e. $K = 1$.

As shown in Table 9, Fig. 13, and Fig. 14, the captions generated by the `iSVP`-enhanced model were significantly better aligned with the images, according to standard metrics for Text-to-Image alignment like ITS and ImageReward. These results provide strong evidence that the `SVP` framework is also an effective and general mechanism for improving vision-language alignment at inference time.

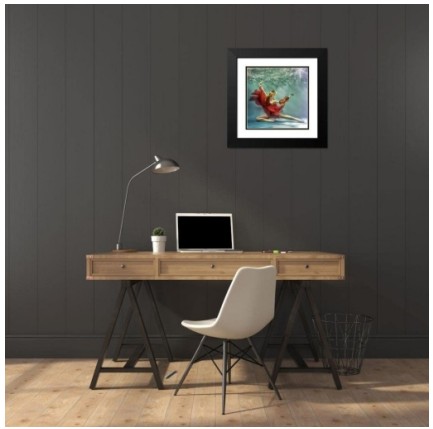

**(a)** Input image

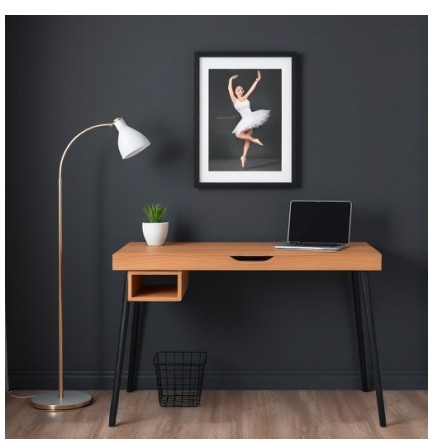

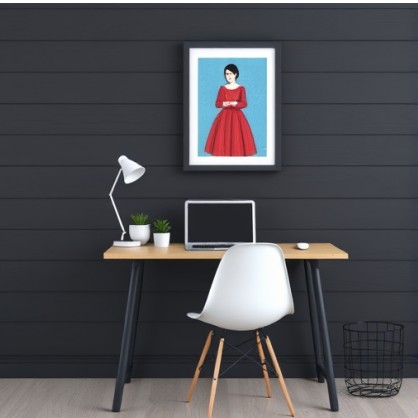

**(b)** Text-to-Image generation using base VLM response - $\mathbf{z} \sim p_\theta(\mathbf{z}|\mathbf{c})$. See left side 11.

**(c)** Text-to-Image generation using grounded VLM response - $\mathbf{z} \sim q(\mathbf{z}|\mathbf{c}, \mathbf{g})$. See right side 11.

**Figure 12:** FLUX-schnell (Labs, 2024) text to image generation using the original VLM response (left) and the response leveraging grounding (right) as input. We generated a single image without multiple attempts or selective filtering. The comparison clearly illustrates that the grounding-enhanced response produces more accurate and reliable generation outcomes.

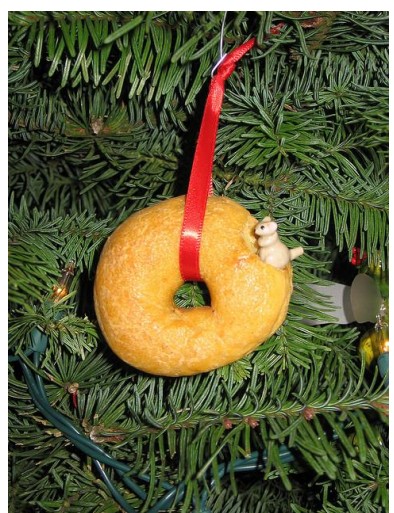

**(a)** Input image from `coco2017_cap_val_lite`. Image id: `000000466567`. Target Captions (provided as ground truth): ["A tree with a donut as an ornament", "A plastic tree with a doughnut hanging by a strip of red ribbon.", "A Christmas ornament is a donut with a squirrel on it.", "A doughnut hanging from a Christmas tree as a decoration.", "a donut being used as an ornament for a chistmas tree"]

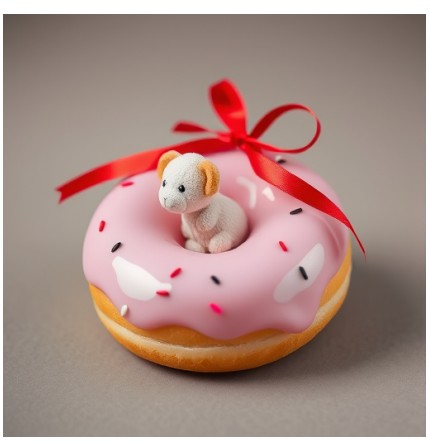

**(b)** Text-to-Image generation using base VLM response - $\mathbf{z} \sim p_\theta(\mathbf{z}|\mathbf{c})$: `"A donut with a red ribbon and a small toy animal on it"` for image (a).

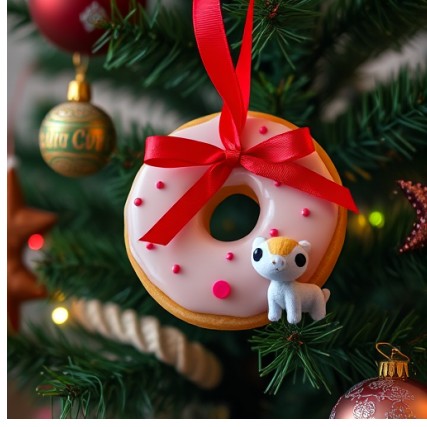

**(c)** Text-to-Image generation using grounded VLM response - $\mathbf{z} \sim q(\mathbf{z}|\mathbf{c},\mathbf{g})$: `"A donut with a red ribbon and a small toy animal on a Christmas tree"` for image (a).

**Figure 13:** FLUX-schnell (Labs, 2024) text to image generation using the base VLM response (left) and the response using `iSVP` (right) as input. We generated a single image without multiple attempts or selective filtering.

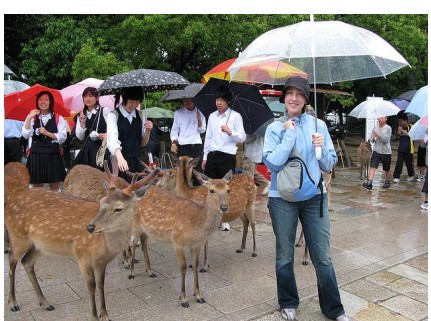

**(a)** Input image from `coco2017_cap_val_lite`. Image id: `000000253742`. Target Captions (provided as ground truth): ["A woman standing next to a herd of animals.", "a woman holding an umbrella at the park", "A woman standing in the rain with an umbrella with a herd of deer behind her.", "On a rainy day at the zoo umbrellas are frequently seen.", "Several people holding umbrellas and standing next to deer."]

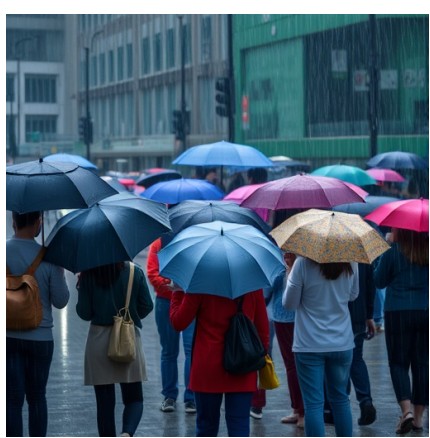

**(b)** Text-to-Image generation using base VLM response - $\mathbf{z} \sim p_\theta(\mathbf{z}|\mathbf{c})$: `"A group of people holding umbrellas and standing in the rain"` for image (a).

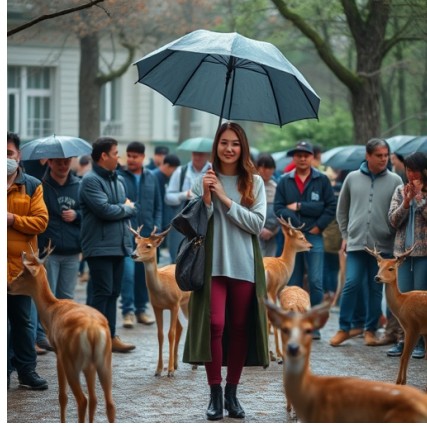

**(c)** Text-to-Image generation using grounded VLM response - $\mathbf{z} \sim q(\mathbf{z}|\mathbf{c}, \mathbf{g})$: `"A woman holding an umbrella stands among a group of people and deer"` for image (a).

**Figure 14:** FLUX-schnell (Labs, 2024) text to image generation using the base VLM response (left) and the response using `iSVP` (right) as input. We generated a single image without multiple attempts or selective filtering.

# D  SVP ALGORITHMS

---

**Algorithm 2** Sampling-based Visual Projection (SVP) w/ log-ratio scoring $S(q, p)$

---

**Require:**
1: Base VLM $p_\theta(\mathbf{z}_p|\mathbf{c})$
2: Grounding model $g(\mathbf{z}, \mathbf{c})$
3: Scoring function $S(q, p)$
4: Seed images $\mathcal{C} = \{\mathbf{c}_c\}_{c=1}^C$
5: Samples per image $K$, top-k ratio $k$
6: Learning rate $\alpha$, iterations $I$, vocabulary size $V$, grounded sequence length $T$
**Ensure:** Updated model parameters $\theta_1 \leftarrow \theta$
7: **for** iteration $i = 1$ to $I$ **do**
8:     $\mathcal{D} \leftarrow \{\}$                                                    ▷ Initialize dataset
9:     **for** each image $\mathbf{c} \in \mathcal{C}$ **do**
10:         $\mathbf{Z}_q \leftarrow \{\}$                                              ▷ Sample buffer
11:         **for** $j = 1$ to $K$ **do**
12:             $\mathbf{z}_p^j \sim p_{\theta_i}(\mathbf{z}|\mathbf{c})$              ▷ Sample from prior
13:             $\mathbf{g}_j \leftarrow g(\mathbf{z}_p^j, \mathbf{c}_v)$              ▷ Grounding feedback
14:             $\mathbf{z}_q^j \sim q(\mathbf{z}|\mathbf{c}, \mathbf{g}_j)$          ▷ Sample with grounding
15:             $\mathbf{Z}_q \leftarrow \mathbf{Z}_q \cup \{\mathbf{z}_q^j\}$
16:         **end for**
17:         **for** $\mathbf{z}_q \in \mathbf{Z}_q$ **do**
18:             $S(q, p)_{\mathbf{z}_q} \leftarrow \sum_{t=1}^T \sum_{v=1}^V w_{v,t}[\log q_{v,t} - \log p_{v,t}]$
19:             $s_q \leftarrow S(q, p_{\theta_i})_{\mathbf{z}_q}$                     ▷ Score samples
20:         **end for**
21:         $S_k \leftarrow k$-th highest score in $\{s_q\}$
22:         $\mathbf{Z}^* \leftarrow \{\mathbf{z}_q : s_q \geq S_k\}$                 ▷ Select top-k
23:         $\mathcal{D} \leftarrow \mathcal{D} \cup \{(\mathbf{c}, \mathbf{z}) : \mathbf{z} \in \mathbf{Z}^*\}$
24:     **end for**
25:     **for** minibatch $B \subset \mathcal{D}$ **do**
26:         $\mathcal{L}(\theta) \leftarrow -\frac{1}{|B|} \frac{1}{|k(\mathbf{c})|} \sum_{(\mathbf{c},\mathbf{z}) \in B} \log p_\theta(\mathbf{z}|\mathbf{c})$
27:         $\theta_i \leftarrow \theta - \alpha \nabla_\theta \mathcal{L}$            ▷ Update parameters
28:     **end for**
29:     $\theta_{i+1} = \theta_i$
30: **end for**
        **return** $p_{\theta_I}(\mathbf{z}|\mathbf{c})$

---

**Algorithm 3** Sampling-based Visual Projection (SVP) w/ weighted difference scoring $\Delta(q,p)$

---

**Require:**
 1: Base VLM $p_\theta(\mathbf{z}_p|\mathbf{c})$
 2: Grounding model $g(\mathbf{z}, \mathbf{c})$
 3: Scoring function $\Delta(q, p)$
 4: Seed images $\mathcal{C} = \{\mathbf{c}_c\}_{c=1}^{C}$
 5: Samples per image $K$, top-k ratio $k$
 6: Learning rate $\alpha$, iterations $I$, vocabulary size $V$, grounded sequence length $T$
**Ensure:** Updated model parameters $\theta_1 \leftarrow \theta$
 7: **for** iteration $i = 1$ to $I$ **do**
 8:     $\mathcal{D} \leftarrow \{\}$             ▷ Initialize dataset
 9:     **for** each image $\mathbf{c} \in \mathcal{C}$ **do**
10:        $\mathbf{Z}_q \leftarrow \{\}$           ▷ Sample buffer
11:        **for** $j = 1$ to $K$ **do**
12:           $\mathbf{z}_p^j \sim p_{\theta_i}(\mathbf{z}|\mathbf{c})$       ▷ Sample from prior
13:           $\mathbf{g}_j \leftarrow g(\mathbf{z}_p^j, \mathbf{c}_v)$      ▷ Grounding feedback
14:           $\mathbf{z}_q^j \sim q(\mathbf{z}|\mathbf{c}, \mathbf{g}_j)$     ▷ Sample with grounding
15:           $\mathbf{Z}_q \leftarrow \mathbf{Z}_q \cup \{\mathbf{z}_q^j\}$
16:        **end for**
17:        **for** $z_q \in Z_q$ **do**
18:           $\Delta(q,p)_{\mathbf{z}_q} = \sum_{t=1}^{T} \sum_{v=1}^{V} w_{v,t}^q \log q_{v,t} - \sum_{t=1}^{T} \sum_{v=1}^{V} w_{v,t}^p \log p_{\theta_{v,t}}$
19:           $s_q \leftarrow \Delta(q,p)_{\mathbf{z}_q}$       ▷ Score samples
20:        **end for**
21:        $S_k \leftarrow k$-th highest score in $\{s_q\}$
22:        $\mathbf{Z}^* \leftarrow \{\mathbf{z}_q : s_q \geq S_k\}$     ▷ Select top-k
23:        $\mathcal{D} \leftarrow \mathcal{D} \cup \{(\mathbf{c}, \mathbf{z}) : \mathbf{z} \in \mathbf{Z}^*\}$
24:     **end for**
25:     **for** minibatch $B \subset \mathcal{D}$ **do**
26:        $\mathcal{L} \leftarrow -\frac{1}{|B|} \frac{1}{|k(\mathbf{c})|} \sum_{(\mathbf{c},\mathbf{z}) \in B} \log p_\theta(\mathbf{z}|\mathbf{c})$
27:        $\theta \leftarrow \theta - \alpha \nabla_\theta \mathcal{L}$      ▷ Update parameters
28:     **end for**
29:     $\theta_{i+1} = \theta_i$
30: **end for**
        **return** $p_{\theta_I}(\mathbf{z}|\mathbf{c})$

---

# E   ADDITIONAL EXPERIMENTS

## E.1   IMAGE TEXTUALIZATION ABLATION

**Table 10:** Accuracy comparison for SVP and Image-Textualization (IT) on the POPE dataset splits. SVP is competitive with IT and more data efficient, requiring less feedback.

| Name | Samples | All | Adversarial | Popular | Random |
|------|---------|-----|-------------|---------|--------|
| LLaVA w/ SVP | 4k | **88.3** | **85.9** | 89.1 | **90.1** |
| LLaVA w/ IT (Pi et al., 2024) | 10k | 86.4 | 81.3 | 90.6 | 87.4 |
| GPT4-V w/ IT (Pi et al., 2024) | 50k | 87.4 | 83.3 | **90.8** | 88.2 |

Table 10 compares our SVP approach with Image Textualization (IT) on the POPE benchmark, which evaluates object hallucination via yes/no existence queries across multiple splits (All, Adversarial, Popular, Random).

LLaVA trained with SVP achieves the highest overall accuracy of 88.3% using only 4k preference samples, outperforming LLaVA with IT (86.4% with 10k samples) and GPT-4V with IT (87.4% with 50k samples). SVP also attains the best performance on the more challenging Adversarial and Random splits (85.9 and 90.1%), while remaining competitive on the Popular split despite using substantially less feedback than IT-based baselines.

These results indicate that supervision-free preferences derived directly from the base MLLM can replace large-scale IT-generated descriptions, yielding strong robustness to hallucination with markedly improved data efficiency.

## E.2 IoU Ablation

In this work we have seen how SVP can be leveraged to improve vision-language alignment without relying on additional human annotation. However, because our analysis is focused on the textual output in a VLM, it is unclear if and how the geometric alignment in the grounding model is amortized in the VLM using SVP.

Her we provide an experiment on a subset of RefCOCO (`val_lite`) with 500 samples. We sample a base LLaVA-1.6-7B model and the same model adapted with w/ SVP using $k = 5$ to better evaluate the robustness of our method to the number of images/sample. Then we use the grounding model to ground generated caption from both models, and compute the IoU@0.5 with the RefCOCO ground truth bounding box.

Figure 15 shows performance metrics for LLaVA (blue bars) and LLaVA w/ SVP (brown bars) on RefCOCO referring tasks. The x-axis displays three key metrics: IoU@0.5 (intersection over union at 50%), IoU<0.05 (a failure rate metric, not grounded object), and CIDEr (a consensus-based image description metric). The results highlights the positive impact of SVP, showing a 30% increase

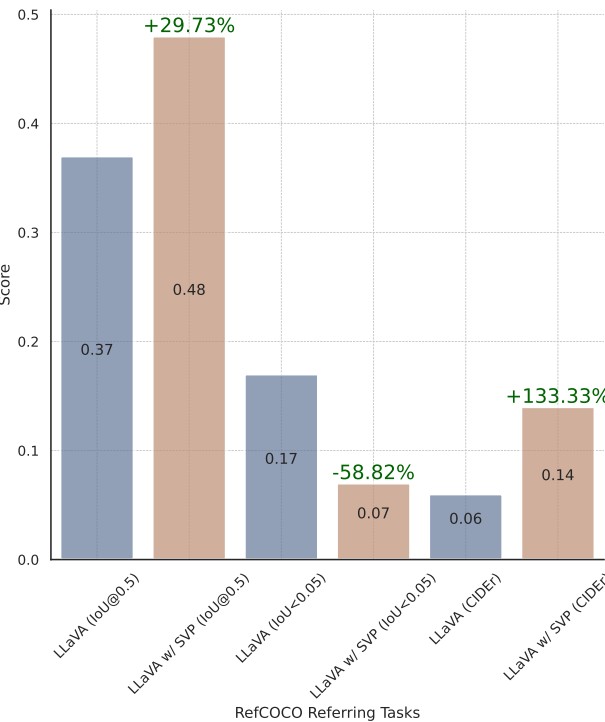

**Figure 15:** Performance comparison of the baseline LLaVA model against LLaVA with Sampling-based Visual Projection (SVP) across RefCOCO referring tasks, demonstrating significant improvements in grounding accuracy (IoU@0.5) and captioning quality (CIDEr), alongside a reduction in failure cases (IoU<0.05)

in IoU@0.5 scores (rising from 0.37 to 0.48) and a substantial 130% increase in CIDEr scores (rising from 0.06 to 0.14). Furthermore, the failure rate IoU<0.05) drops significantly by 58.82% (from 0.17 to 0.07), indicating that the SVP integration effectively reduces instances where the model fails to ground the target object. These results clearly show that the feedback-driven self-training mechanism driving SVP not only improves visual-language alignment, but effectively extract geometric knowledge from the grounding model. See Figure 16 for visualizations.

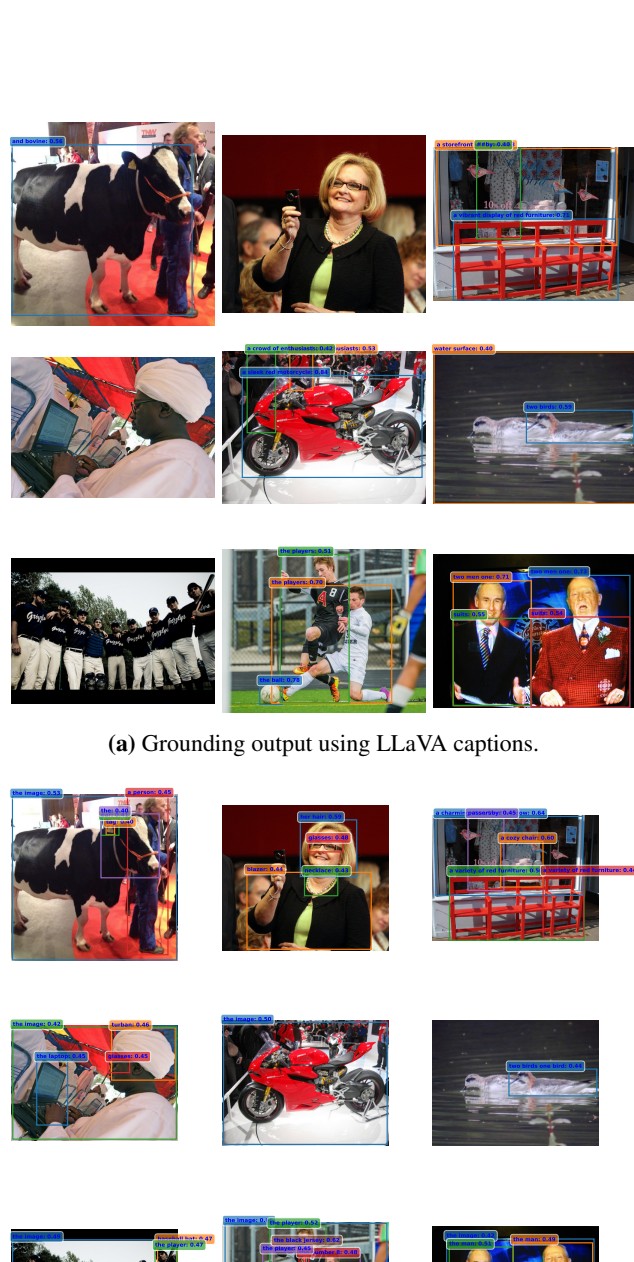

**(a)** Grounding output using LLaVA captions.

**(b)** Grounding output using LLaVA w/ SVP captions.

**Figure 16:** Qualitative comparison of grounding performance using different captioning strategies. (a) Baseline LLaVA generates captions that often lack groundable details or fail to trigger detections entirely (missing detections in 3/9 examples). (b) LLaVA w/ SVP produces more descriptive, groundable captions, resulting in successful detections across all examples and improved downstream IoU performance. The green bounding boxes illustrate the grounding model's ability to localize the caption's content.

## E.3   GROUNDING MODEL ABLATION

To assess the impact of visual grounding quality on model performance, we conducted an ablation study using the RefCOCO (`val_lite`) dataset comprising 500 images. We evaluated four configurations: a baseline without grounding (w/o GDINO), a noisy variant where 30% of GDINO-tiny predictions are randomly dropped (GDINO-noisy), our standard setup (GDINO-tiny), and a high-capacity version (GDINO-base). Performance was measured across three difficulty levels defined by Intersection over Union (IoU) thresholds: easy (IoU@0.5), medium (IoU@0.75), and hard (IoU@0.95). As shown in the results, increasing the capacity of the grounding model yields consistent performance gains across all strictness levels. The GDINO-base model achieved the highest accuracy in every category, peaking at over 0.50 for IoU@0.5 and maintaining a lead even at the strictest IoU@0.95 threshold. Notably, providing even noisy grounding signals (GDINO-noisy) offered a tangible improvement over the ungrounded baseline, confirming that explicit visual localization cues are critical for this task.

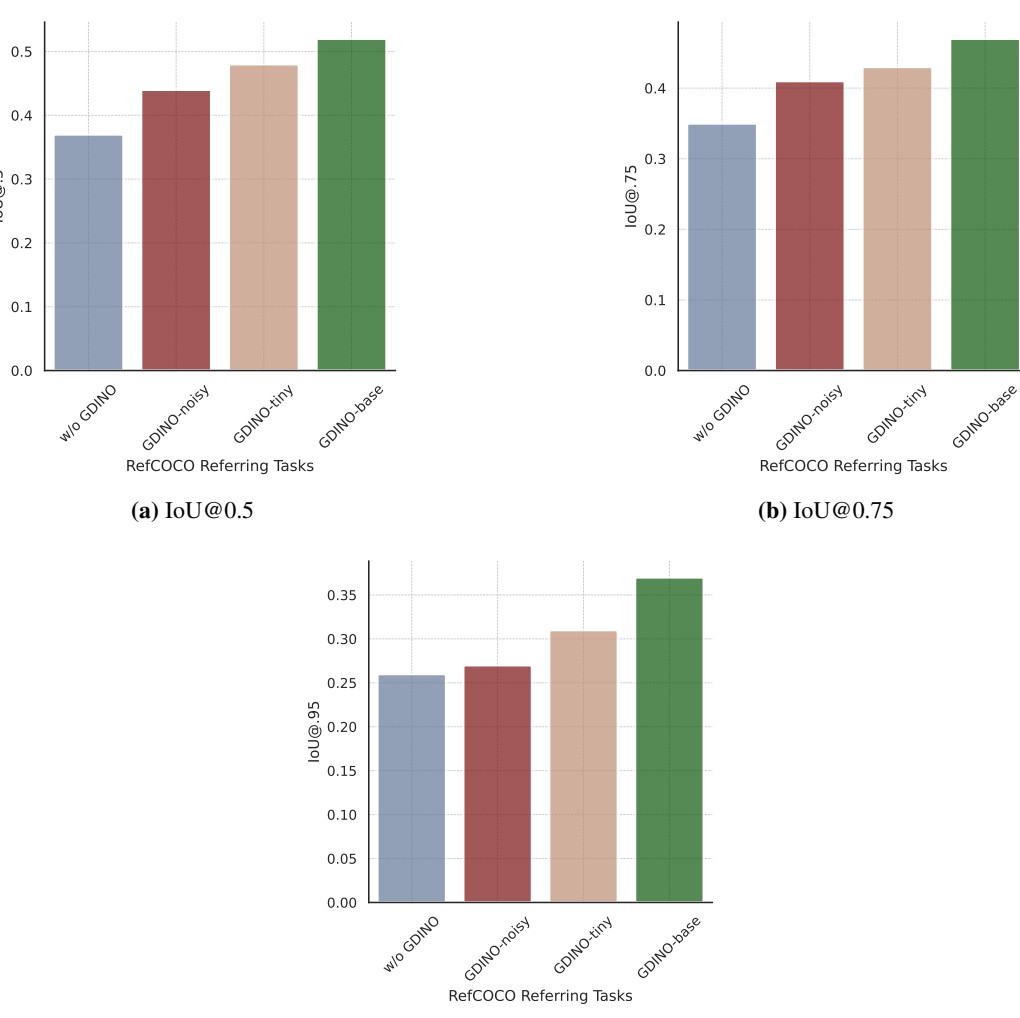

**(a)** IoU@0.5

**(b)** IoU@0.75

**(c)** IoU@0.95

**Figure 17:** Performance ablation of different grounding models on RefCOCO validation tasks. We compare four setups: w/o GDINO (no grounding), GDINO-noisy (GDINO-tiny predictions with 30% random dropout), GDINO-tiny (standard setup), and GDINO-base (high capacity). Bar charts display IoU accuracy across three difficulty thresholds: (a) Easy (IoU@0.5), (b) Medium (IoU@0.75), and (c) Hard (IoU@0.95). Higher capacity models consistently improve performance across all strictness levels, demonstrating the importance of robust visual grounding signals.

### E.4 NUMBER OF GENERATED SAMPLES PER IMAGE $K$ ABLATION

To assess the impact of the sampling strategy on model performance, we conducted an ablation study on the number of samples per image, denoted as $K$, generated during the sampling loop. We evaluated the LLaVA-1.6-7B model across a diverse set of benchmarks, including RefCOCO (`val_lite`) and Flickr30k (`test_lite`) for captioning capabilities, the MMMU val set for multitasking performance, and POPE for hallucination control. We varied $K$ across the set $0, 1, 5, 10, 20$, where $K = 0$ represents a baseline with no grounding and $K = 1$ introduces grounding without scoring. As shown in Figure 18, performance metrics exhibit a consistent upward trend as the number of samples increases. We observed a sharp improvement in generation quality as $K$ rose to 10, beyond which the performance gains saturated, plateauing around $K = 20$. Based on these results, we selected $K = 20$ for the majority of our subsequent experiments to maximize performance while maintaining computational feasibility.

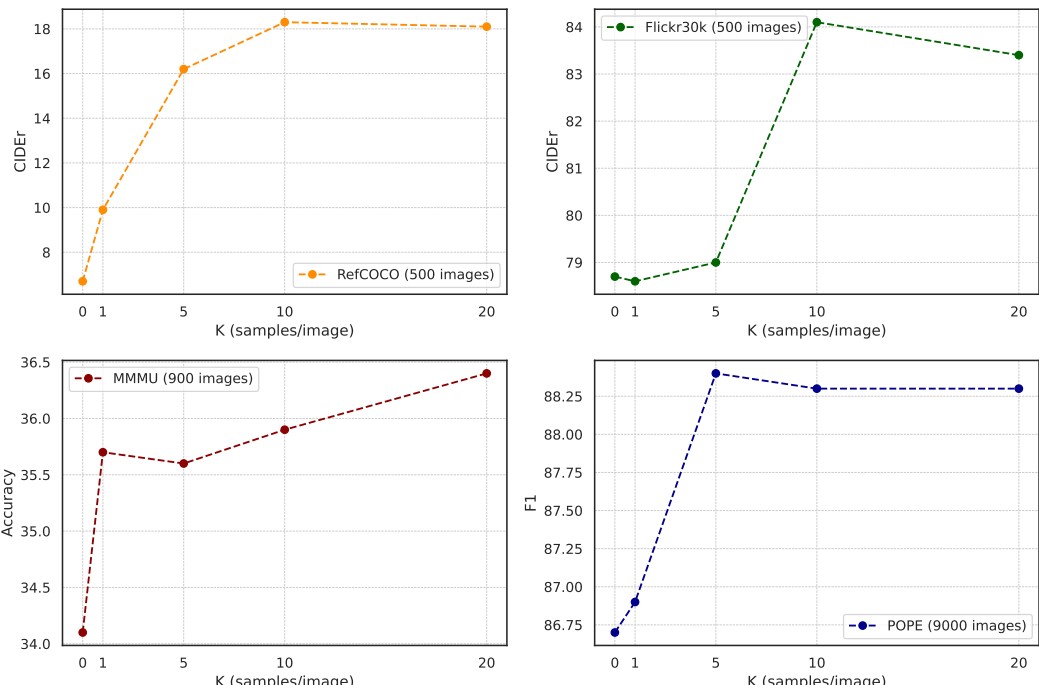

**Figure 18:** Ablation study on the effect of sample count $K$ during the sampling loop. $K = 0$ represents the baseline without grounding, while $K = 1$ utilizes grounding without scoring. We report CIDEr scores for captioning tasks (RefCOCO, Flickr30k), Accuracy for multitasking (MMMU), and F1 scores for hallucination evaluation (POPE). Results indicate that increasing $K$ yields significant gains up to $K = 10$, with performance saturating near $K = 20$. Consequently, $K = 20$ is adopted for subsequent experiments.

## E.5 CAPTIONING

**Table 11: Captioning Performance** on COCO2014, NoCaps, COCO2017, and Flickr30k datasets (80k samples) using `lmms-eval`. Results compare LLaVA-1.6-7B/13B models with weighted-difference ($\Delta(q,p)$) and log-ratio ($S(q,p)$) scoring mechanisms. Performance measured by METEOR (M), ROUGE-L (R), and CIDEr (C); higher scores better. See K for dataset details.

| Model | Score | COCO2014_val | | | COCO2017_val | | | NoCaps_test | | | Flickr30k_test | | |
|---|---|---|---|---|---|---|---|---|---|---|---|---|---|
| | | M | R | C | M | R | C | M | R | C | M | R | C |
| LLaVA-1.6-7b | - | 26.14 | 54.25 | 107.65 | 26.00 | 54.12 | 109.32 | 27.03 | 56.98 | 96.08 | 23.63 | 51.61 | 73.17 |
| w/ SVP (C) | $\Delta(q,p)$ | 28.74 | 56.69 | 111.98 | 28.74 | 56.69 | **114.77** | 29.37 | 59.52 | **104.79** | 25.62 | 53.25 | 75.98 |
| w/ SVP (CVQ) | $\Delta(q,p)$ | **29.26** | 56.62 | 111.38 | 29.24 | 56.67 | 114.72 | 30.07 | **59.69** | 104.58 | **26.34** | **53.58** | **77.68** |
| w/ SVP (C) | $S(q,p)$ | 28.64 | **56.74** | **112.45** | 28.57 | **56.71** | 114.69 | 29.29 | 59.62 | 104.75 | 25.54 | 53.40 | 76.53 |
| w/ SVP (CVQ) | $S(q,p)$ | 29.22 | 56.25 | 109.57 | **29.25** | 56.34 | 113.08 | **30.08** | 59.55 | 104.01 | 26.26 | 53.23 | 76.73 |
| LLaVA-1.6-13b | - | 24.67 | 52.03 | 99.39 | 24.72 | 52.23 | 102.04 | 25.44 | 54.93 | 88.13 | 22.21 | 48.78 | 66.68 |
| w/ SVP (C) | $\Delta(q,p)$ | 25.31 | 54.28 | 104.83 | 25.30 | 54.40 | 107.20 | 26.16 | 57.21 | 93.11 | 22.54 | 50.82 | 67.77 |
| w/ SVP (CVQ) | $\Delta(q,p)$ | 28.38 | **56.71** | **113.30** | **28.49** | **57.03** | **117.23** | 28.94 | **59.19** | **102.32** | **25.69** | **53.61** | **78.11** |
| w/ SVP (C) | $S(q,p)$ | 25.32 | 54.22 | 104.84 | 25.37 | 54.37 | 107.52 | 26.14 | 57.14 | 93.11 | 22.71 | 51.00 | 68.56 |
| w/ SVP (CVQ) | $S(q,p)$ | **28.39** | 56.54 | 112.65 | 28.35 | 56.67 | 116.09 | **28.96** | 59.14 | 101.93 | 25.59 | 53.25 | 77.00 |

**Table 12: Captioning Performance** on COCO2014, NoCaps, COCO2017, and Flickr30k datasets (80k samples) using `lmms-eval`. Comparing LLaVA-1.6-7B/13B models with weighted-difference ($\Delta(q,p)$) and log-ratio ($S(q,p)$) scoring. Evaluated using BLEU-1 to BLEU-4 (B1-B4); higher scores better.

| Model | Score | COCO2014_val | | | | COCO2017_val | | | | NoCaps_test | | | | Flickr30k_test | | | |
|---|---|---|---|---|---|---|---|---|---|---|---|---|---|---|---|---|---|
| | | B4 | B3 | B2 | B1 | B4 | B3 | B2 | B1 | B4 | B3 | B2 | B1 | B4 | B3 | B2 | B1 |
| LLaVA-1.6-7b | - | 31.04 | 41.51 | 54.40 | 68.81 | 30.82 | 41.24 | 54.14 | 68.54 | 38.43 | 50.03 | 62.89 | 75.43 | 28.57 | 39.90 | 54.54 | 71.41 |
| w/ SVP (C) | $\Delta(q,p)$ | 32.29 | 44.25 | 59.33 | 76.16 | 32.61 | 44.50 | 59.44 | 76.09 | 41.05 | 54.12 | **68.82** | 83.18 | 28.94 | 40.62 | 55.85 | 73.53 |
| w/ SVP (CVQ) | $\Delta(q,p)$ | 31.69 | 43.50 | 58.46 | 75.52 | 32.01 | 43.72 | 58.53 | 75.53 | 40.88 | 53.78 | 68.49 | **83.42** | 29.22 | 40.71 | 55.63 | 73.27 |
| w/ SVP (C) | $S(q,p)$ | **32.75** | **44.76** | **59.86** | **76.71** | 32.82 | **44.74** | **59.78** | 76.54 | 41.15 | **54.17** | 68.77 | 82.93 | **29.59** | **41.38** | **56.68** | 74.36 |
| w/ SVP (CVQ) | $S(q,p)$ | 30.95 | 42.67 | 57.60 | 74.76 | 31.46 | 43.02 | 57.78 | 74.86 | 40.29 | 53.27 | 68.16 | 83.17 | 28.76 | 40.09 | 54.90 | 72.56 |
| LLaVA-1.6-13b | - | 27.33 | 36.76 | 48.51 | 61.98 | 27.64 | 37.06 | 48.84 | 62.33 | 34.06 | 44.86 | 56.93 | 68.78 | 24.28 | 34.50 | 48.31 | 65.26 |
| w/ SVP (C) | $\Delta(q,p)$ | 29.97 | 39.65 | 51.34 | 63.79 | 29.96 | 39.65 | 51.37 | 63.76 | 37.28 | 48.33 | 59.97 | 70.31 | 27.15 | 37.88 | 51.83 | 67.78 |
| w/ SVP (CVQ) | $\Delta(q,p)$ | **33.65** | **45.40** | **59.99** | 76.45 | **34.28** | **45.90** | **60.43** | **76.71** | **40.77** | **53.66** | **68.09** | **82.25** | **29.91** | **41.92** | **57.53** | **75.55** |
| w/ SVP (C) | $S(q,p)$ | 29.97 | 39.78 | 51.67 | 64.45 | 30.25 | 39.97 | 51.83 | 64.56 | 37.54 | 48.61 | 60.40 | 71.12 | 27.60 | 38.64 | 52.60 | 68.83 |
| w/ SVP (CVQ) | $S(q,p)$ | 33.45 | 45.26 | 59.90 | **76.47** | 34.00 | 45.59 | 60.10 | 76.50 | 40.35 | 53.24 | 67.81 | 82.17 | 29.40 | 41.39 | 57.03 | 75.18 |

## E.6 HALLUCINATION RATE

**Table 13:** Hallucination (F1) and Recall (R) using Qwen2-VL7b and Qwen2.5-VL-7b. Naively scaling the multimodal finetuning dataset does not solve visual-language alignment in more powerful VLMs, and can even be detrimental.

| Name | Multimodal Samples | F1 (all - 9k) | F1 (adv - 3k) | F1 (pop - 3k) | F1 (random - 3k) | Recall (all - 9k) |
|---|---|---|---|---|---|---|
| Qwen2-VL-7b (Yang et al., 2024) | 100M | 87.8 | **85.8** | 87.9 | 89.8 | 83.4 |
| Qwen2.5-VL-7b (Yang et al., 2024) | 100M | 86.2 | 85.1 | 86.1 | 87.1 | 77.9 |
| LLaVA-1.6-7b w/ SVP | 800K | **88.3** | **85.9** | **89.1** | **90.1** | **84.4** |

**Table 14:** Hallucination Mitigation performance on POPE benchmark. Comparison of LLaVA model variants' F1 scores across adversarial, popular, random, and overall splits. Results demonstrate impacts of model size, fine-tuning strategy, encoder choices, and SVP adaptation on hallucination avoidance.

| Model | Size | $v_\theta$ | $t_\theta$ | POPE ($F1$ score ↑) | | | |
|---|---|---|---|---|---|---|---|
| | | | | adv | pop | random | all |
| LLaVA (Liu et al., 2024c) | 7b | CLIP | Vicuna | 72.0 | 75.3 | 80.7 | 76.0 |
| LLaVA-SFT$^+$ (Sun et al., 2023) | 7b | CLIP | Vicuna | 80.1 | 82.4 | 85.5 | 82.7 |
| LLaVA-RLHF (Sun et al., 2023) | 7b | CLIP | Vicuna | 79.5 | 81.8 | 83.3 | 81.5 |
| LLaVA (Liu et al., 2024c) | 13b | CLIP | Vicuna | 74.4 | 78.2 | 78.8 | 77.1 |
| LLaVA-SFT$^+$ (Sun et al., 2023) | 13b | CLIP | Vicuna | 81.1 | 82.6 | 84.8 | 82.8 |
| LLaVA-RLHF (Sun et al., 2023) | 13b | CLIP | Vicuna | 80.5 | 81.8 | 83.5 | 81.9 |
| LLaVA-NeXT-DPO (Liu et al., 2024b) | 7b | CLIP | Qwen2 | 83.43 | 83.78 | 84.73 | 83.98 |
| LLaVA-OV-DPO (Li et al., 2024) | 7b | SigLIP | Qwen2 | 85.12 | 86.24 | 87.37 | 86.24 |
| LLaVA-HA-DPO (Zhao et al., 2023) | 7b | CLIP | Vicuna | 82.54 | 87.89 | **90.25** | 86.90 |
| LLaVA-1.5 (Liu et al., 2024a) | 13b | CLIP | Vicuna | 84.53 | 86.31 | 87.17 | 86.00 |
| LLaVA-1.5 w/ SVP | 13b | CLIP | Vicuna | 84.66 | 86.84 | 87.44 | 86.31 |
| LLaVA-1.6 (Liu et al., 2024b) | 7b | CLIP | Mistral | 85.43 | 86.87 | 88.05 | 86.73 |
| LLaVA-1.6 w/ SVP | 7b | CLIP | Mistral | **85.93** | **89.04** | 90.02 | **88.33** |
| LLaVA-1.6 (Liu et al., 2024b) | 13b | CLIP | Vicuna | 85.17 | 86.36 | 87.20 | 86.24 |
| LLaVA-1.6 w/ SVP | 13b | CLIP | Vicuna | 85.15 | 87.50 | 89.23 | 87.30 |
| LLaVA-OV (Li et al., 2024) | 0.5b | SigLIP | Qwen2 | 82.28 | 83.19 | 83.89 | 83.12 |
| LLaVA-OV w/ SVP | 0.5b | SigLIP | Qwen2 | 83.45 | 84.70 | 85.46 | 84.53 |
| LLaVA-1.6 (Liu et al., 2024b) | 34b | CLIP | Yi-2 | - | - | - | 87.7 |
| InternVL (Chen et al., 2024b) | 19b | IViT | Vicuna | - | - | - | 87.6 |
| InternVL-1.2 (Chen et al., 2024b) | 40b | IViT | Yi-2 | - | - | - | 88.0 |
| InternVL-1.2$^+$ (Chen et al., 2024b) | 40b | IViT | Yi-2 | - | - | - | 88.7 |
| VILA-1.5 (Lin et al., 2024) | 8b | SigLIP | LLaMA3 | - | - | - | 85.6 |
| VILA-1.5 (Lin et al., 2024) | 8b | SigLIP | Vicuna | - | - | - | 86.3 |
| VILA-1.5 (Lin et al., 2024) | 40b | IViT | Yi2 | - | - | - | 87.3 |
| VILA-1.5-AWQ (Lin et al., 2024) | 40b | IViT | Yi2 | - | - | - | 88.2 |

**Table 15:** Evaluating hallucination rates in different VLMs adapted with fine-tuning, train-time adaptation, and test-time adaptation. Higher is better. Eff-Size: effective model size for multi-phase inference pipelines. Woodpecker (Yin et al., 2023) requires multiple models to process the response.

| Model | Size | Eff-Size | $v_\theta$ | $t_\theta$ | POPE (*Acc* score ↑) | | |
|---|---|---|---|---|---|---|---|
| | | | | | adv | pop | random |
| *Fine-tuning* | | | | | | | |
| InstructBLIP (Dai et al., 2023) | 7b | 7b | ViT | FlanT5 | 72.1 | 82.7 | 88.6 |
| LLaVA-SFT$^+$ (Sun et al., 2023) | 7b | 7b | CLIP | Vicuna | 80.2 | 82.9 | 86.1 |
| mPLUG-Owl2 (Ye et al., 2024) | 8b | 8b | ViT | LLaMA2 | 84.1 | 86.2 | 88.3 |
| InstructBLIP (Dai et al., 2023) | 13b | 13b | ViT | Vicuna | 74.5 | 81.4 | 88.7 |
| LLaVA-SFT$^+$ (Sun et al., 2023) | 13b | 13b | CLIP | Vicuna | 82.3 | 83.9 | 85.2 |
| *Test-time adaptation* | | | | | | | |
| QwenVL w/ VCD (Leng et al., 2024) | 7b | 14b | CLIP | Vicuna | 84.3 | 87.1 | 88.6 |
| LLaVA w/ M3ID (Favero et al., 2024) | 7b | 14b | CLIP | Vicuna | 65.8 | 69.3 | 76.0 |
| Otter w/ Woodpecker (Yin et al., 2023) | 7b | $\geq$ 14b | CLIP | LLaMA | 83.0 | 84.3 | 86.7 |
| mPLUG-Owl w/ Woodpecker (Yin et al., 2023) | 7b | $\geq$ 14b | ViT | LLaMA | 81.0 | 84.1 | 86.3 |
| LLaVA w/ M3ID (Favero et al., 2024) | 13b | 26b | CLIP | Vicuna | 71.3 | 77.0 | 84.3 |
| *Train-time adaptation* | | | | | | | |
| LLaVA-M3ID-DPO (Favero et al., 2024) | 7b | 7b | CLIP | Vicuna | 68.2 | 73.9 | 81.2 |
| LLaVA-RLHF (Sun et al., 2023) | 7b | 7b | CLIP | Vicuna | 80.7 | 83.3 | 84.8 |
| LLaVA-NeXT-DPO (Rafailov et al., 2024) | 7b | 7b | CLIP | Qwen2 | 85.2 | 85.6 | 86.6 |
| LLaVA-OV-DPO (Rafailov et al., 2024) | 7b | 7b | SigLIP | Qwen2 | 86.3 | 87.5 | 88.7 |
| LLaVA-HA-DPO (Zhao et al., 2023) | 7b | 7b | CLIP | Vicuna | 81.5 | 87.9 | **90.5** |
| SeVa (Zhu et al., 2024) | 7b | 7b | CLIP | Vicuna | 83.6 | 87.4 | 89.4 |
| LLaVA-M3ID-DPO (Favero et al., 2024) | 13b | 13b | CLIP | Vicuna | 73.2 | 79.1 | 85.2 |
| LLaVA-RLHF (Sun et al., 2023) | 13b | 13b | CLIP | Vicuna | 82.3 | 83.9 | 85.2 |
| InstructBLIP-HA-DPO (Zhao et al., 2023) | 13b | 13b | ViT | Vicuna | 80.7 | 85.8 | 89.8 |
| LLaVA-1.6 (Liu et al., 2024b) | 7b | 7b | CLIP | Mistral | 86.4 | 87.9 | 89.2 |
| LLaVA-1.6 w/ SVP | 7b | 7b | CLIP | Mistral | 86.2 | **89.6** | **90.6** |
| LLaVA-1.6 (Liu et al., 2024b) | 13b | 13b | CLIP | Vicuna | 86.4 | 87.7 | 88.5 |
| LLaVA-1.6 w/ SVP | 13b | 13b | CLIP | Vicuna | **86.7** | 88.4 | 89.2 |
| LLaVA-OV | 0.5b | 0.5b | SigLIP | Qwen2 | 84.3 | 85.2 | 86.0 |
| LLaVA-OV w/ SVP | 0.5b | 0.5b | SigLIP | Qwen2 | 85.0 | 86.3 | 87.2 |

## E.7 REFERRING TASKS

**Table 16:** Evaluation of referring expression generation on various `RefCOCO`, `RefCOCO+`, and `RefCOCOg` datasets using LLaVA-1.6-7b. The experiment compares the performance of different models, including a base model, a model without visual grounding (w/o **g**), a model with Visual Projections (w/ `SVP` (C)), and a model with `SVP` and Visual Query (w/ `SVP` (CVQ) ). The performance is measured using the CIDEr score on bounding box (`bbox`) and segmentation (`seg`) referring task on the test and validation sets for each dataset. The results show that `SVP` models significantly outperform the base and w/o **g** models, indicating the importance of visual grounding for referring tasks. Notice that the adapted models do not have access to the bounding boxes during fine-tuning.

| | base | w/o **g** | $\Delta(q,p)$ w/ `SVP` (C) | $\Delta(q,p)$ w/ `SVP` (CVQ) | $S(q,p)$ w/ `SVP` (C) | $S(q,p)$ w/ `SVP` (CVQ) |
|---|---|---|---|---|---|---|
| *RefCOCO* | | | | | | |
| `bbox-test`  | 9.53  | 3.57  | 18.99 | 26.96 | 20.74 | 25.52 |
| `bbox-testA` | 5.91  | 1.59  | 11.14 | 14.37 | 12.33 | 14.00 |
| `bbox-testB` | 12.35 | 6.27  | 25.13 | 36.65 | 27.64 | 34.71 |
| `bbox-val`   | 9.93  | 3.95  | 18.84 | 27.01 | 21.07 | 25.76 |
| `seg-test`   | 9.46  | 3.70  | 18.27 | 25.02 | 19.68 | 23.89 |
| `seg-testA`  | 5.32  | 1.37  | 9.48  | 12.67 | 10.95 | 11.70 |
| `seg-testB`  | 12.92 | 6.44  | 25.49 | 35.08 | 26.61 | 33.28 |
| `seg-val`    | 9.44  | 4.02  | 18.35 | 25.15 | 19.60 | 23.95 |
| *RefCOCO+* | | | | | | |
| `bbox-testA` | 6.68  | 2.16  | 12.25 | 16.93 | 14.05 | 16.44 |
| `bbox-testB` | 10.98 | 6.21  | 23.31 | 33.02 | 25.46 | 30.98 |
| `bbox-val`   | 9.57  | 3.68  | 18.00 | 26.67 | 20.70 | 25.35 |
| `seg-testA`  | 5.98  | 1.86  | 10.74 | 13.97 | 12.30 | 13.56 |
| `seg-testB`  | 11.75 | 6.45  | 23.67 | 31.25 | 24.59 | 29.70 |
| `seg-val`    | 9.19  | 3.90  | 17.15 | 24.31 | 19.13 | 23.81 |
| *RefCOCOg* | | | | | | |
| `bbox-test`  | 20.27 | 13.68 | 47.74 | 59.74 | 50.89 | 56.79 |
| `bbox-val`   | 19.70 | 12.16 | 47.69 | 59.65 | 50.73 | 56.81 |
| `seg-test`   | 18.76 | 12.90 | 45.23 | 54.39 | 47.51 | 51.18 |
| `seg-val`    | 18.77 | 12.55 | 45.45 | 54.01 | 46.93 | 50.77 |

## E.8 ITERATION ABLATION

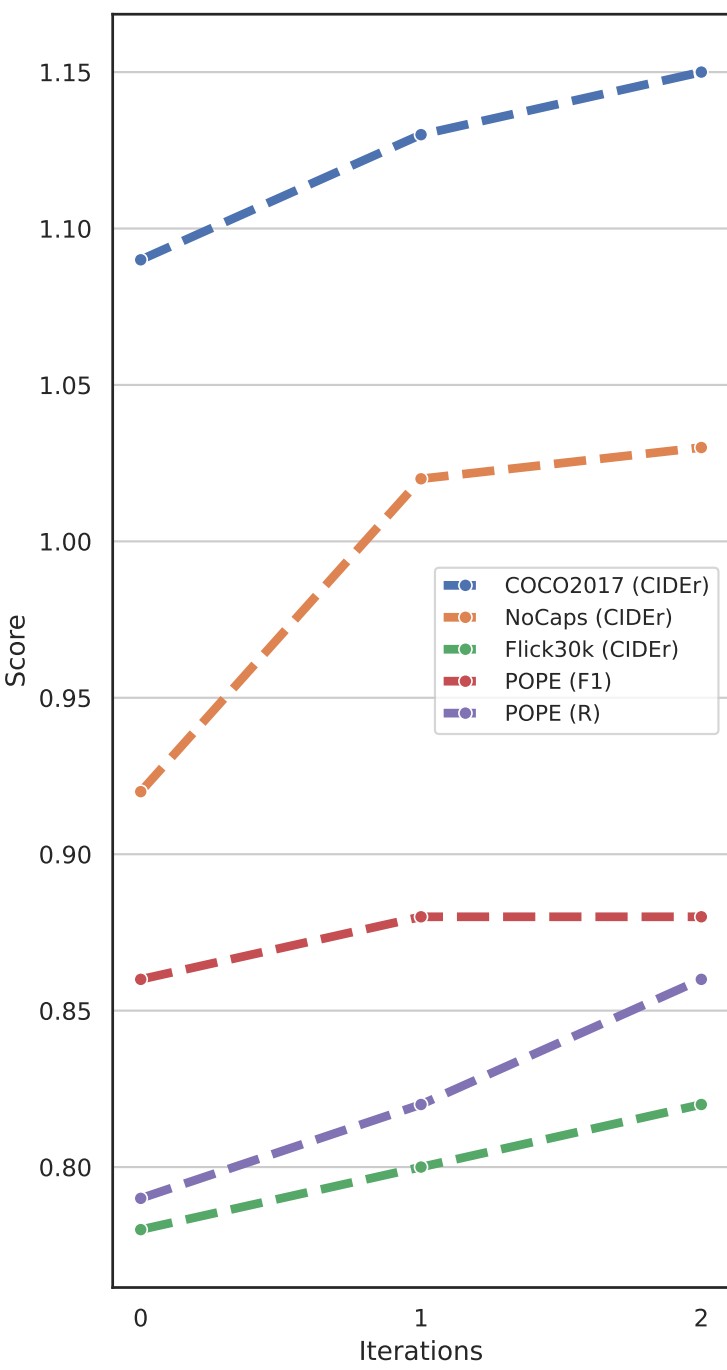

**Figure 19:** SVP effectively boosts captioning performance and reduces hallucinations on benchmark tasks using LLaVA-1.6-7b as base model. The second iteration of SVP adaptation leads to significant improvements compared to the initial round, underscoring the value of this technique for enhancing visual-language model capabilities. However, the gains tend to plateau after the second iteration, suggesting diminishing returns from further fine-tuning.

## E.9  MODEL SIZE ABLATION

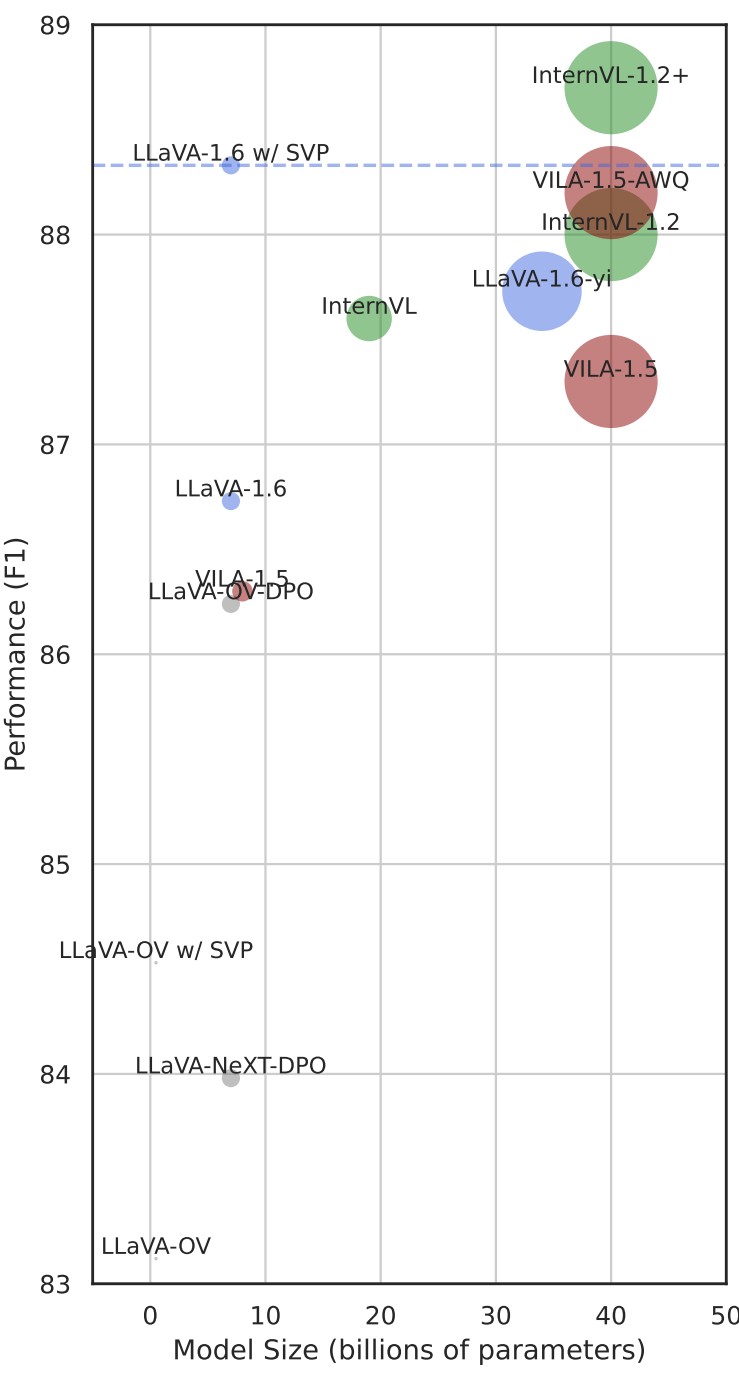

**Figure 20:** Model size comparison using the F1 metric on the POPE dataset. `SVP` improves the base model and achieves better or comparable performance with models five times larger.

## E.10  OBJECT GROUNDING ABLATION

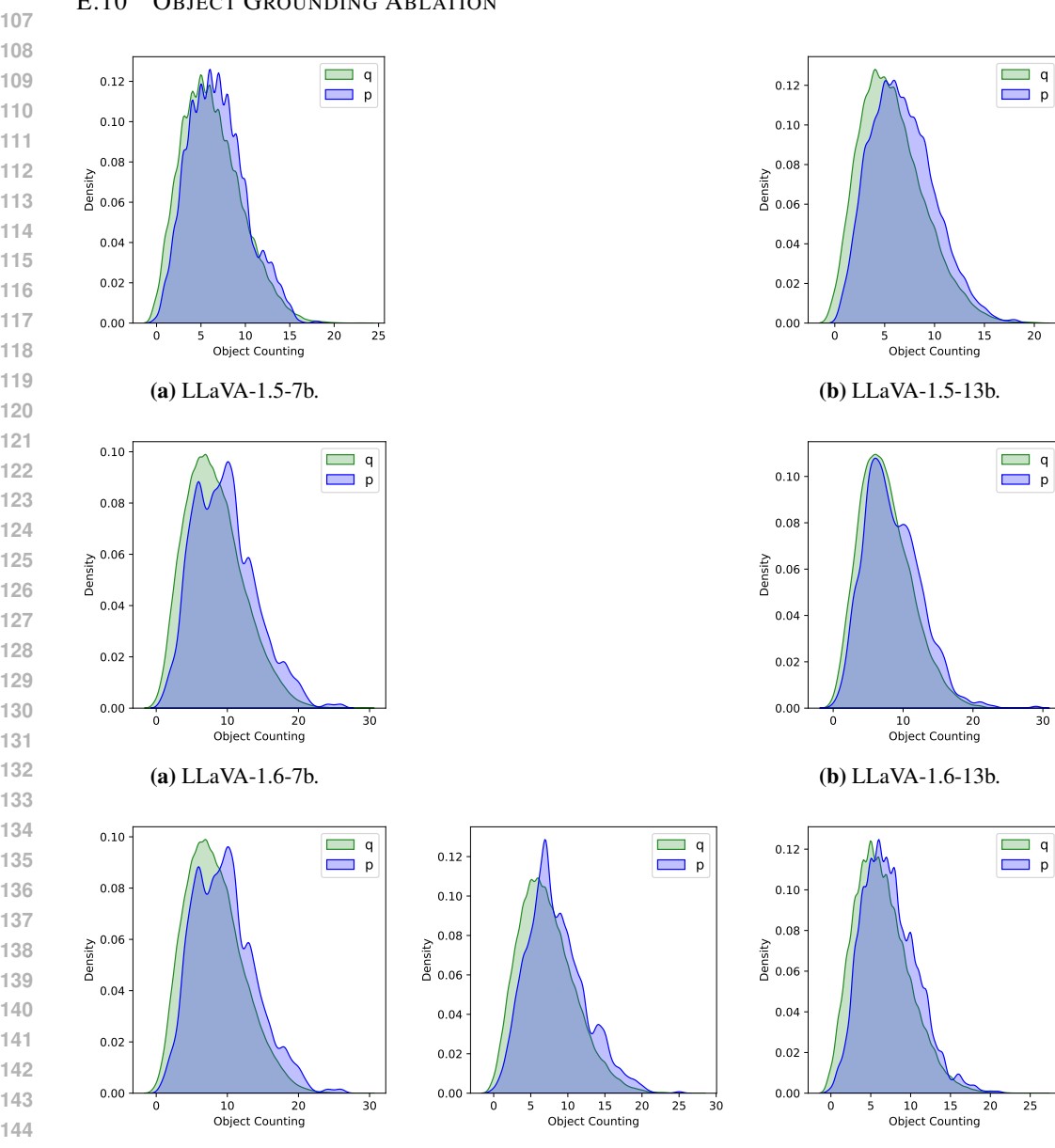

**(a)** LLaVA-1.5-7b.

**(b)** LLaVA-1.5-13b.

**(a)** LLaVA-1.6-7b.

**(b)** LLaVA-1.6-13b.

**(a)** LLaVA-1.6-7b iteration 1.

**(b)** LLaVA-1.6-7b iteration 2.

**(c)** LLaVA-1.6-7b iteration 3.

**Figure 23:** Distribution of groundable objects in generated caption sampling the base model $p_\theta(\mathbf{z}|\mathbf{c})$ and the grounded model $q(\mathbf{z}|\mathbf{c}, \mathbf{g})$. Models adapted with SVP generate less groundable objects and have better object recall.

## E.11   SCORE ABLATION

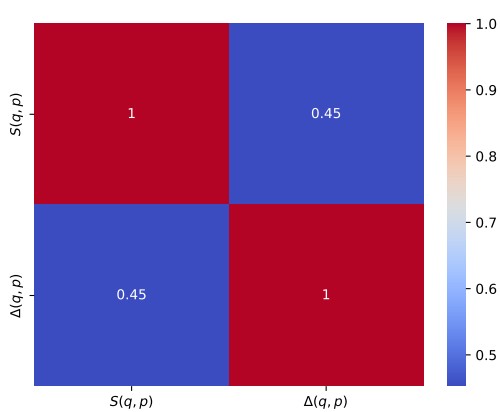

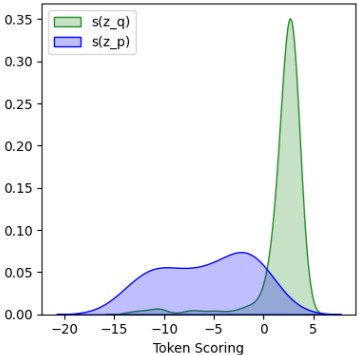

**(a)** **Top1 Ranking Correlation** for weighted-difference $\Delta(q, p)$ and log-ratio $S(q, p)$ score using LLaVA-1.6-7b as base model.

**(b)** **Empirical Distribution** of sequence scores. Log-space representation of $S(q, p_\theta)$ for sequence scoring. We see the scoring mechanism's effectiveness to differentiate between posterior samples $\mathbf{z}_q$ (with grounding) and prior samples $\mathbf{z}_p$ (without grounding).

# F TRAINING OBJECTIVE DERIVATION

We derive our visual-language alignment objective following two approaches: re-weighted maximum likelihood and greedy off-policy optimization. Assuming a deterministic output distribution $p(\mathbf{x}|\mathbf{z}, \mathbf{c}) = d(\mathbf{z}, \mathbf{c})$, we start with re-weighted maximum likelihood as a negated KL maximization:

$$\mathcal{F}_{\text{MLE}}(\mathbf{c}; \theta) = -\mathbb{KL}\left[q(\mathbf{z}|\mathbf{c}, \mathbf{g}), p_\theta(\mathbf{z}|\mathbf{c})\right] = \int q(\mathbf{z}|\mathbf{c}, \mathbf{g}) \log p_\theta(\mathbf{z}|\mathbf{c}) d\mathbf{z} - \int q(\mathbf{z}|\mathbf{c}, \mathbf{g}) \log q(\mathbf{z}|\mathbf{c}, \mathbf{g}) d\mathbf{z} \tag{8}$$

Taking the gradient with respect to $\theta$:

$$\nabla_\theta \mathcal{F}_{\text{MLE}}(\mathbf{c}; \theta) = \nabla_\theta \int q(\mathbf{z}|\mathbf{c}, \mathbf{g}) \log p_\theta(\mathbf{z}|\mathbf{c}) d\mathbf{z} = \int q(\mathbf{z}|\mathbf{c}, \mathbf{g}) \nabla_\theta \log p_\theta(\mathbf{z}|\mathbf{c}) d\mathbf{z} \tag{9}$$

Approximating the expectation with $K$ samples from $\mathbf{z} \sim q(\mathbf{z}|\mathbf{c}, \mathbf{g})$ and filtering using our scoring mechanism:

$$\nabla_\theta \mathcal{F}_{\text{MLE}}^{k(\mathbf{c})}(\mathbf{c}; \theta) \approx \frac{1}{|k(\mathbf{c})|} \sum_{i=1}^{K} \left[ \mathbb{1}\{\mathbf{z}^i : S(q(\mathbf{z}^i|\mathbf{c}, \mathbf{g}), p_\theta(\mathbf{z}^i|\mathbf{c})) \geq S_{k(\mathbf{c})}\} \right]_{\text{sg}} \nabla_\theta \log p_\theta(\mathbf{z}^i|\mathbf{c}) \tag{10}$$

For the policy optimization approach, we begin with the standard on-policy REINFORCE estimator using our scoring mechanism $f(\mathbf{z})$ as reward:

$$\mathcal{F}_{\text{RL-ON}}(\mathbf{c}; \theta) = \mathbb{E}_{p_\theta(\mathbf{z}|\mathbf{c})}\left[f(\mathbf{z})\right] = \int p_\theta(\mathbf{z}|\mathbf{c}) f(\mathbf{z}) d\mathbf{z} \tag{11}$$

The gradient for $\theta$ yields:

$$\nabla_\theta \mathcal{F}_{\text{RL-ON}}(\mathbf{c}; \theta) = \nabla_\theta \int p_\theta(\mathbf{z}|\mathbf{c}) f(\mathbf{z}) d\mathbf{z} = \int \nabla_\theta p_\theta(\mathbf{z}|\mathbf{c}) f(\mathbf{z}) d\mathbf{z} = \int p_\theta(\mathbf{z}|\mathbf{c}) \nabla_\theta \log p_\theta(\mathbf{z}|\mathbf{c}) f(\mathbf{z}) d\mathbf{z}. \tag{12}$$

To incorporate our guiding distribution $q$, we use importance sampling:

$$\nabla_\theta \mathcal{F}_{\text{RL-ON}}^{q}(\mathbf{c}; \theta) = \int q(\mathbf{z}|\mathbf{c}, \mathbf{g}) \frac{p_\theta(\mathbf{z}|\mathbf{c})}{q(\mathbf{z}|\mathbf{c}, \mathbf{g})} \nabla_\theta \log p_\theta(\mathbf{z}|\mathbf{c}) f(\mathbf{z}) d\mathbf{z} \tag{13}$$

This is an unbiased estimator for the on-policy gradient leveraging the "off-policy" or behavioral/guiding distribution $q$. If now we approximating the expectation for $q$ with $K$ samples and filter using the score contained in $f(\mathbf{z})$, we can write:

$$\nabla_\theta \mathcal{F}_{\text{RL-OFF}}^{k(\mathbf{c})}(\mathbf{c}; \theta) \approx \frac{1}{|k(\mathbf{c})|} \sum_{i=1}^{K} \left[ \mathbb{1}\{\mathbf{z}^i : S(q(\mathbf{z}^i|\mathbf{c}, \mathbf{g}), p_\theta(\mathbf{z}^i|\mathbf{c})) \geq S_{k(\mathbf{c})}\} \right]_{\text{sg}} \frac{p_\theta(\mathbf{z}^i|\mathbf{c})}{q(\mathbf{z}^i|\mathbf{c}, \mathbf{g})} \nabla_\theta \log p_\theta(\mathbf{z}^i|\mathbf{c}), \tag{14}$$

where we leverage the fact that $f(\mathbf{z}^i) = \mathbb{1}\{\mathbf{z}^i : S(q(\mathbf{z}^i|\mathbf{c}, \mathbf{g}), p_\theta(\mathbf{z}^i|\mathbf{c})) \geq S_{k(\mathbf{c})}\}$. Gradients are stopped over the scores in the indicator function.

By construction we are only retaining samples with low importance ratio $p_\theta/q$. We are introducing bias focusing on samples that will improve vision-language alignment, and reducing the importance sampling estimator variance. Simplifying the previous gradient considering the importance ratio constant, we obtain the objective we maximize:

$$\nabla_\theta \tilde{\mathcal{F}}_{\text{RL-OFF}}^{k(\mathbf{c})}(\mathbf{c}; \theta) \approx \frac{1}{|k(\mathbf{c})|} \sum_{i=1}^{K} \left[ \mathbb{1}\{\mathbf{z}^i : S(q(\mathbf{z}^i|\mathbf{c}, \mathbf{g}), p_\theta(\mathbf{z}^i|\mathbf{c})) \geq S_{k(\mathbf{c})}\} \right]_{\text{sg}} \nabla_\theta \log p_\theta(\mathbf{z}^i|\mathbf{c}) \tag{15}$$

Both approaches yield equivalent gradients after approximations: $\nabla_\theta \mathcal{F}_{\text{MLE}}^{k(\mathbf{c})}(\mathbf{c}; \theta) = \nabla_\theta \tilde{\mathcal{F}}_{\text{RL-OFF}}^{k(\mathbf{c})}(\mathbf{c}; \theta)$. This equivalence provides a strong theoretical foundation for our method. We optimize this objective at each SVP iteration by averaging over a batch of visual inputs: $\mathcal{L}(\theta) = -1/|B| \sum_{c=1}^{C} \mathcal{F}(\mathbf{c}; \theta)$.

## G PROMPTING

**System Prompt - Sampling**

You are an AI visual-language assistant that can analyze images and helps writing detailed descriptions of images.
`<instruction>`
Describe the scene and the objects in the image in details. Describe the object attributes and positions. Output only the descriptions of objects that are in the image. Use separate sentence for each object.
Include details like object counts, position of the objects, relative position between the objects. Start your description with "In the image, ".
`</instruction>`

**System Prompt - Grounded Sampling**

You are an AI visual-language assistant that can analyze images and helps writing detailed descriptions of images.
In addition, specific objects and object locations within the image are given, along with detailed coordinates inside `<context></context>`. These coordinates are in the form of bounding boxes, represented as `(x1, y1, x2, y2)` with floating numbers ranging from 0 to 1. These values correspond to the top left x, top left y, bottom right x, and bottom right y.
`<instruction>`
Using the provided objects and bounding boxes inside `<context></context>`, describe the image.
Describe the scene and the objects in the image in details. Describe the object attributes and positions. Output only the descriptions of objects that are in the image. Use separate sentence for each object.
Include details like object counts, position of the objects, relative position between the objects. *Do not mention the bounding box coordinates. Utilize this data to explain the scene using natural language.*
Start your description with "In the image, ".
`</instruction>`

**Base Prompt**

Please generate a detailed and comprehensive description for the content of this image. Be precise.

**Grounded Prompt**

`<context>`
`{grounding}`
`</context>`
Please generate a detailed and comprehensive description for the content of this image. Be precise.

# H ITERATIVE SELF-TRAINING IN GENERATIVE MODELS

Our method is inspired by recent advances in self-training in language modelling (Zelikman et al., 2022; Dong et al., 2023; Gulcehre et al., 2023), where the focus has been in improving chain-of-though reasoning leveraging feedback. The Self-Taught Reasoner (STaR (Zelikman et al., 2022)) uses ground truth labels and rationalization as feedback (Zelikman et al., 2022) and fine-tunes the model on reasoning steps that generate the correct answer (Hoffman et al., 2024). In 25 we provide a comparison between SVP and rationalization in STaR, where the model tries to find the correct reasoning path given external feedback in the form or a ground truth response.

```
P: Where do you put your
grapes just before
checking out?
Answer Choices:
(a) mouth
(b) grocery cart
(c) super market
(d) fruit basket
(e) fruit market
C: grocery cart (CORRECT)
R: The answer should be
the place where grocery
items are placed before
checking out.
Grocery cart makes the most
sense for holding grocery
items.
```

```
P: <IMAGE> Please describe
the content of this image.
C:
lamp   [0.33,0.47,0.16,0.23],
laptop [0.48,0.54,0.16,0.09],
desk   [0.49,0.75,0.61,0.33],
plant  [0.36,0.56,0.04,0.06],
frame  [0.64,0.22,0.24,0.24],
basket [0.80,0.80,0.15,0.15],
R: The image depicts
a modern work setup.
On the wall,
there is a framed artwork.
Below the frame,
there is a desk.
On the desk,
there is a laptop.
```

**(a)** Rationalization in STaR. We sample from $q(\mathbf{z}|\mathbf{c}, \mathbf{y})$, where $\mathbf{y}$ is the ground truth label provided as context $C$. The response $\mathbf{z}$ is then leveraged to build $p(\mathbf{y}|\mathbf{c}, \mathbf{z})$ or a deterministic decoding $d(\mathbf{c}, \mathbf{z})$.

**(b)** Grounded sampling in SVP. We sample from $q(\mathbf{z}|\mathbf{c}, \mathbf{g})$, where $\mathbf{g}$ is the grounding information provided as context $C$. The response $\mathbf{z}$ is then leveraged to build $p(\mathbf{x}|\mathbf{c}, \mathbf{z})$ or a deterministic decoding $d(\mathbf{c}, \mathbf{z})$.

**Figure 25:** Comparison between rationalization in STaR (Zelikman et al., 2022) and posterior sampling in SVP.

## I DPO Derivation

The DPO loss comparing policy $\pi_\theta$ to reference $\pi_{\text{ref}}$ is:
$$\mathcal{L}_{\text{DPO}}(\pi_\theta, \pi_{\text{ref}}) = -\mathbb{E}_{(\mathbf{x}, \mathbf{y}_w, \mathbf{y}_l) \sim \mathcal{D}}[\log \sigma(\beta \, \delta r_\theta)], \tag{16}$$
where $\mathbf{x}$ is the input prompt, $\mathbf{y}_w$ and $\mathbf{y}_l$ are preferred and dis-preferred responses, $\sigma(z)$ is the sigmoid function, and $\beta = 1$ for simplicity. $\delta r_\theta$ represents the log-probability ratio difference between winning and losing samples:
$$\delta r_\theta = \log \frac{\pi_\theta(\mathbf{y}_w|\mathbf{x})}{\pi_\theta(\mathbf{y}_l|\mathbf{x})} = \log \frac{\pi_\theta(\mathbf{y}_w|\mathbf{x})}{\pi_{\text{ref}}(\mathbf{y}_w|\mathbf{x})} - \log \frac{\pi_\theta(\mathbf{y}_l|\mathbf{x})}{\pi_{\text{ref}}(\mathbf{y}_l|\mathbf{x})} = r_w(\theta) - r_l(\theta) \tag{17}$$

**Preference Feedback and Optimal Policy**  In determining the shape of DPO's implicit reward $r_w(\theta)$, we can draw insights from the standard PPO formulation used in RLHF. The RLHF framework integrates reinforcement learning with human preferences through three key components: (i) a reward model $s_\psi$ (typically parametric) that encodes human preference labels; (ii) a generative policy model $\pi_\theta$ that can be sampled and improved through reward feedback; (iii) a reference model $\pi_{\text{ref}}$ that provides stability during learning. This framework is expressed mathematically as:
$$\mathcal{F}_{\text{PPO}} = \mathbb{E}_{\pi_\theta(\mathbf{z}|\mathbf{c})} \left[ s_{\bar{\psi}}(\mathbf{z}, \mathbf{c}) - \gamma \log \frac{\pi_\theta(\mathbf{z}|\mathbf{c})}{\pi_{\text{ref}}(\mathbf{z}|\mathbf{c})} \right], \tag{18}$$
where $\mathbf{z}$ represents a textual continuation for a given prompt $\mathbf{c}$ (either visual or textual). For this regularized policy optimization problem, it can be demonstrated that the optimal policy takes the form:
$$\pi_\theta^*(\mathbf{z}|\mathbf{c}) \propto \pi_{\text{ref}}(\mathbf{z}|\mathbf{c}) \exp\left( \frac{s_{\bar{\psi}}(\mathbf{z}, \mathbf{c})}{\gamma} \right). \tag{19}$$

When we isolate the reward term $s_{\bar{\psi}}$, we find that this formulation aligns with the DPO framework, differing only by constant terms.

**Gradient Derivation**  Applying the chain rule to find $\nabla_\theta \mathcal{L}_{\text{DPO}}$:
$$-\nabla_\theta \mathcal{L}_{\text{DPO}} = \mathbb{E}\left[ \frac{\sigma'(\delta r_\theta)}{\sigma(\delta r_\theta)} \nabla_\theta \delta r_\theta \right] \tag{20}$$
Using $\sigma'(z) = \sigma(z)(1 - \sigma(z))$:
$$\nabla_\theta \mathcal{L}_{\text{DPO}} = -\mathbb{E}\left[ (1 - \sigma(\delta r_\theta)) \cdot \nabla_\theta \delta r_\theta \right] \tag{21}$$
The gradient of $\delta r_\theta$ simplifies to:
$$\nabla_\theta \delta r_\theta = \nabla_\theta \log \pi_\theta(\mathbf{y}_w|\mathbf{x}) - \nabla_\theta \log \pi_\theta(\mathbf{y}_l|\mathbf{x}) \tag{22}$$
The final DPO gradient is:
$$\nabla_\theta \mathcal{L}_{\text{DPO}} = -\beta \, \mathbb{E}\left[ (1 - \sigma(\delta r_\theta))[\nabla_\theta \log \pi_\theta(\mathbf{y}_w|\mathbf{x}) - \nabla_\theta \log \pi_\theta(\mathbf{y}_l|\mathbf{x})] \right] \tag{23}$$
This formulation optimizes preferences using a re-weighted maximum likelihood without requiring a separate reward model or RL training.

**Preference Optimization and Vision-Language Alignment**  The application of SVP for vision-language alignment shares conceptual similarities with DPO, though with notable distinctions. A key difference lies in our scoring mechanisms' approach to negative samples. While our scores can effectively guide ranking and top-K selection, a small difference between guided and prior distributions doesn't necessarily indicate an undesirable sample. Low scores may simply reflect cases where grounding information offers minimal improvement, such as when the base model's response is already grounded in the visual input, or the response does not require the visual information. Consequently, in SVP, the gradient information from negative samples provides limited value.

DPO employs a gradient re-weighting scheme based on how much a model's preference predictions deviate from ground truth. This acts as a soft sample selection mechanism, giving greater weight to more informative training examples. Our approach parallels this concept, though we implement a hard sample selection strategy that emphasizes examples offering greater vision-language alignment information.

## J  QUALITATIVE EXAMPLES

Here we provide additional visualizations and examples to illustrate our method. The images used for captioning and visual queries were selected from the COCO2014 training set (Lin et al., 2014), while referring examples were sourced from the RefCOCO dataset (Kazemzadeh et al., 2014), specifically the val_lite set (Zhang et al., 2024b). The examples are presented with minimal post-processing, limited to the removal of special characters and correction of obvious grammatical errors. Both models were prompted using the protocols outlined in G. All hyper-parameters remained consistent across both models. The examples compare output of LLaVA-1.6 models sampling from the base model (w/o SVP) and sampling from the adapted model (w/ SVP). These comparisons provide qualitative insights into the performance differences between the base and adapted models.

### J.1  CAPTIONING TASKS

**Instruction:** Describe the content of this image as detailed as possible.

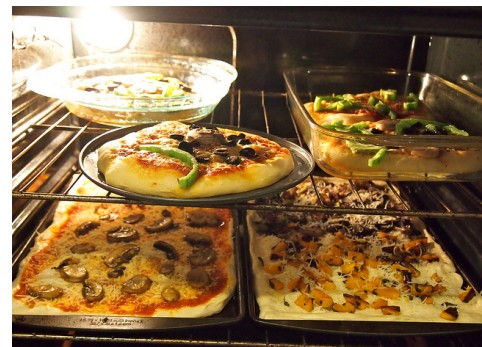

***VLM response w/o SVP:***
In the image, there are four pizzas in an oven. The oven is filled with racks, each holding a pizza. The pizzas are of different types, with various toppings visible. The oven is illuminated, casting a warm glow on the pizzas. The pizzas are arranged in two rows, with two pizzas in each row. The pizzas are cooking, as indicated by their position in the oven. The oven is a standard kitchen appliance, with a door that is open to reveal the pizzas inside. The pizzas are the main focus of the image, with their colors and textures contrasting against the dark interior of the oven. The image captures a moment of culinary preparation, with the pizzas in various stages of cooking.

***VLM response w/ SVP:***
In the image, there is a well-lit oven with a black interior. Inside the oven, there are three pizzas on the top rack and two pizzas on the bottom rack. On the top rack, the first pizza is on the left side of the oven. It has a golden brown crust and is topped with green peppers and mushrooms. The second pizza is on the right side of the oven. It also has a golden brown crust and is topped with green peppers and black olives. On the bottom rack, the first pizza is on the left side of the oven. It has a golden brown crust and is topped with mushrooms. The second pizza is on the right side of the oven. The light inside the oven illuminates the pizzas, highlighting their toppings and crusts. The pizzas appear to be freshly baked, as their crusts are golden brown and their toppings are evenly distributed. The overall scene suggests a home or restaurant kitchen where pizzas are being prepared.

**Instruction:** Describe the content of this image as detailed as possible.

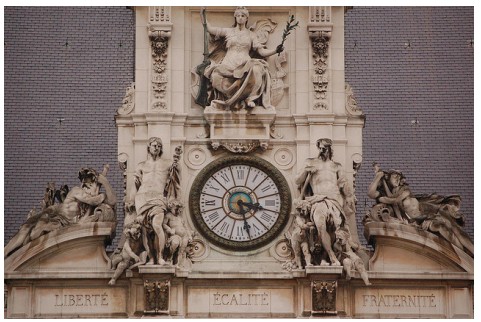

***VLM response w/o SVP:***
In the image, there is a large clock mounted on a building. The clock face is white with black numbers and hands. The building itself is adorned with intricate carvings and statues. There are three statues visible in the image. One statue is of a woman holding a shield, another is of a man holding a spear, and the third is of a man holding a scroll. The carvings and statues are detailed and appear to be made of stone. The building has a gray roof, and the clock is positioned in the center of the image. The statues are positioned on either side of the clock, with the woman on the left, the man with the spear in the middle, and the man with the scroll on the right. The overall scene is a blend of architectural and artistic elements.

***VLM response w/ SVP:***
In the image, there is a grand clock tower that is ornate and intricate. The tower is made of stone and is topped with a statue of a woman holding a sword. This statue is located at the top of the tower, near the center. The tower also features a large clock face on the front, which is situated below the statue. The clock face is white with black numbers and hands. Below the clock face, there are two smaller statues of people. These statues are positioned on either side of the clock face, with the one on the left being slightly larger than the one on the right. The tower is set against a gray brick background, which contrasts with the white stone of the tower. The overall scene suggests a sense of grandeur and history.

**Instruction:** Describe the content of this image as detailed as possible.

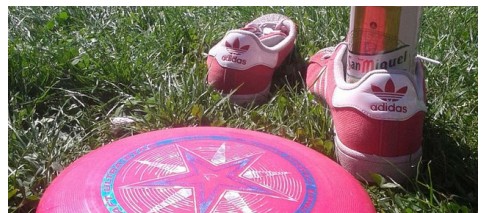

***VLM response w/o SVP:***
In the image, there are three objects: a pink frisbee, a pair of red sneakers, and a beer bottle. The pink frisbee is located in the foreground, with its star-shaped design prominently displayed. The red sneakers are positioned in the background, with one shoe slightly in front of the other. The beer bottle is also in the background, standing upright and slightly to the right of the sneakers. The objects are arranged in a way that suggests a casual, outdoor setting, possibly a park or a beach. The frisbee and sneakers are the main focus of the image, while the beer bottle adds a touch of leisure to the scene.

***VLM response w/ SVP:***
In the image, there is a casual outdoor setting with a pair of red and white Adidas shoes. The shoes are placed on a grassy area, with one shoe closer to the camera and the other slightly behind it. The Adidas logo is prominently displayed on both pairs of shoes. A pink frisbee with a blue star design is also present in the scene. It is positioned in the foreground, with the shoes in the background. The frisbee appears to be resting on the grass, and its bright color stands out against the natural backdrop.The overall scene suggests a leisurely outdoor activity, possibly a game of frisbee, with the Adidas shoes indicating a sporty or active lifestyle. The grassy area and the casual arrangement of the shoes and frisbee give the image a relaxed and informal vibe.

**Instruction:** Describe the content of this image as detailed as possible.

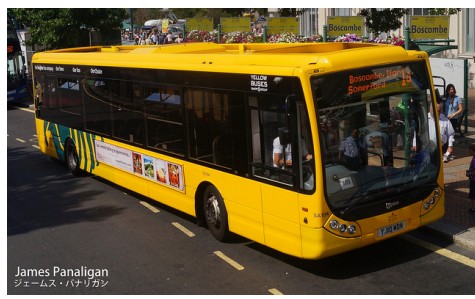

***VLM response w/o SVP:***
In the image, there is a large yellow bus parked on the side of a street. The bus is positioned on the right side of the image and is the most prominent object in the scene. It is a double-decker bus, which is characterized by its two levels of seating. The bus is parked next to a sidewalk, which is visible on the right side of the image. There are several people in the image, some of whom are standing near the bus. One person is standing on the sidewalk, while others are standing on the street. There is also a car parked on the left side of the image, which is partially visible. The bus is parked in front of a building, which is located on the left side of the image. The building is partially visible, with only the top part of it visible in the image. The bus is also parked near a tree, which is located on the left side of the image. The tree is partially visible, with only the top part of it visible in the image. The overall scene suggests a typical day in a city, with people going about their daily routines and public transportation being a common sight.

***VLM response w/ SVP:***
In the image, a large yellow bus dominates the scene, traveling down a city street. The bus is vibrant and appears to be in motion, heading towards the right side of the frame. Its size and bright color make it the main focus of the image. In the background, another bus can be seen, albeit much smaller and less prominent. It's positioned on the left side of the image, providing a sense of depth and perspective to the scene. On the right side of the image, there's a large advertisement. It's not clear what the advertisement is for, but its size and placement suggest it's a significant feature of the urban landscape. The setting appears to be a typical city street. Trees can be seen in the background, adding a touch of nature to the urban environment. There are also several people scattered throughout the image, going about their day, which contributes to the everyday, relaxed atmosphere of the scene. The overall composition of the image suggests a bustling city life, with public transportation playing a significant role. The buses, people, and advertisement all come together to paint a picture of a vibrant, active cityscape.

**Instruction:** Describe the content of this image as detailed as possible.

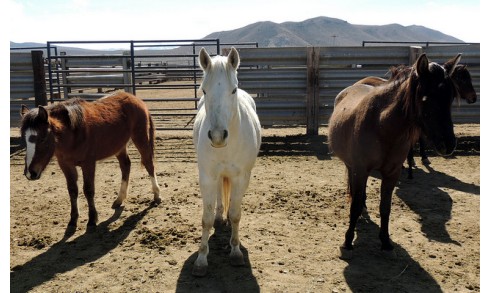

*VLM response w/o SVP:*
In the image, there are three horses standing in a dirt enclosure. The horse on the left is brown with a white blaze on its face. It is standing with its head turned to the side, looking towards the camera. The horse in the middle is white with a brown spot on its head. It is standing with its head turned to the side, looking towards the camera. The horse on the right is brown with a white blaze on its face. It is standing with its head turned to the side, looking towards the camera. The enclosure is made of metal fencing and there is a mountain visible in the background.

*VLM response w/ SVP:*
In the image, there are three horses standing in a dirt enclosure. The horse on the left is brown with a white stripe running down its nose. Its head is slightly turned to the left. The horse in the middle is white and stands with its head facing forward. The horse on the right is black and its head is turned to the right. The three horses are standing next to each other, separated by a metal fence. In the background, there is a clear blue sky and a mountain range. The overall scene is a serene outdoor setting.

**Instruction:** Describe the content of this image as detailed as possible.

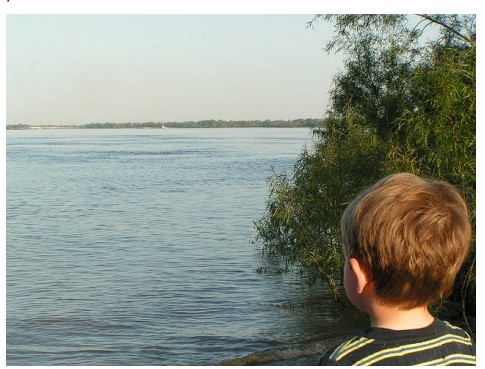

*VLM response w/o SVP:*
In the image, there is a young boy standing on the shore of a body of water. The boy is wearing a striped shirt and has blonde hair. He is looking out towards the water, which is calm and reflects the light. In the distance, there is a boat on the water, and the horizon is visible. The boy is standing in front of a tree with green leaves, which is located on the shore. The overall scene suggests a peaceful day by the water.

*VLM response w/ SVP:*
In the image, a young boy is standing by a body of water. He is wearing a black striped shirt. His back is turned to the camera, and he is looking out at the water. The boy appears to be standing on a grassy area near the water's edge. The water in the image is calm and appears to be a lake or a large river. It is a clear day, and the sky is visible in the background. There are no other people or objects in the image. The boy is the only person present. The relative position of the boy to the water is such that he is standing on the right side of the image. He is standing closer to the foreground than the water, which is further back in the image. The boy is also standing to the right of the water.

**Instruction:** Describe the content of this image as detailed as possible.

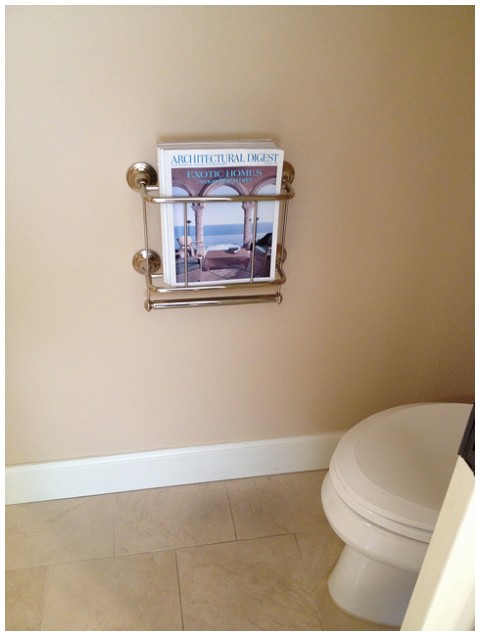

***VLM response w/o SVP:***
In the image, there is a bathroom setting. The main object is a white toilet, which is positioned in the lower right corner of the image. The toilet is situated on a beige tiled floor. Above the toilet, there is a metal rack mounted on the wall. The rack holds a single magazine, which is titled "Architectural Digest". The magazine is displayed prominently on the rack, indicating that it is the main focus of the rack's purpose. The overall scene suggests a well-organized and aesthetically pleasing bathroom space, with the magazine rack providing a functional and decorative element. The white toilet contrasts with the beige tiles, creating a clean and modern look. The magazine on the rack adds a touch of sophistication and suggests an interest in architecture or design.

***VLM response w/ SVP:***
In the image, there is a bathroom setting with a white toilet located towards the right side of the frame. Above the toilet, there is a magazine rack mounted on the wall. The rack holds two magazines, one titled "ARCHITECTURAL DIGEST" and the other "EXOTIC HOMES". The "ARCHITECTURAL DIGEST" magazine is positioned slightly higher than the "EXOTIC HOMES" magazine, and both are centrally located within the rack. The "EXOTIC HOMES" magazine is situated directly below the "ARCHITECTURAL DIGEST" magazine. The bathroom walls are beige, and the overall scene suggests a quiet and relaxed atmosphere.

**Instruction:** Describe the content of this image as detailed as possible.

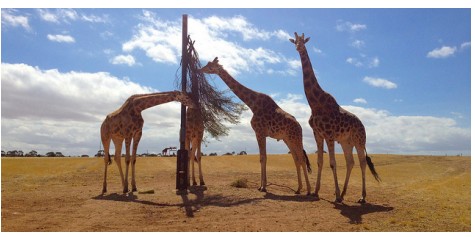

***VLM response w/o SVP:***
In the image, there are three giraffes standing in a field. The giraffe on the left is bending its neck down towards the ground, possibly to reach for food or water. The giraffe in the middle is standing upright, with its neck extended upwards. The giraffe on the right is also standing upright, but its neck is bent down towards the ground. All three giraffes are facing the same direction, suggesting they might be looking at something in the distance. The field they are standing in appears to be dry and dusty, with no visible vegetation. There are no other objects or animals in the image.

***VLM response w/ SVP:***
In the image, there are three giraffes standing in a field. The giraffes are tall and majestic, with their long necks reaching upwards. The giraffe on the left is slightly bending its neck, possibly to reach for something. The giraffe in the middle is standing upright, and the giraffe on the right is also bending its neck. In the center of the image, there is a tall pole. The pole has a feeder attached to it, which is likely providing food for the giraffes. The pole is surrounded by the field, which appears to be dry and brown. The giraffes are spread out across the image, with the giraffe on the left being closer to the pole than the other two. The giraffe on the right is the furthest away from the pole. The giraffes are the main focus of the image, with their long necks and distinctive patterns standing out against the backdrop of the field.

## J.2 REFERRING TASKS

**Instruction:** Provide a short description for this region.

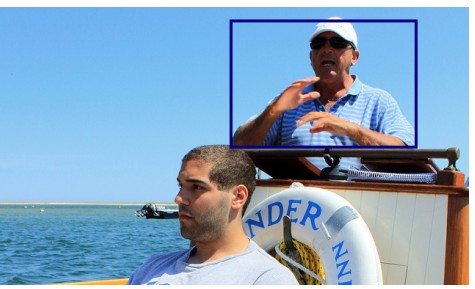

***VLM response w/o SVP:***
The image shows a man sitting in the foreground, looking to the side with a serious expression. Behind him, there is another man standing on a boat, gesturing with his hands. The boat has a lifebuoy with the word "UNDER" on it, and the man standing is wearing a white cap and sunglasses. The background features a clear blue sky and calm water, with a few boats visible in the distance. The overall setting suggests a leisurely day out on the water.

***VLM response w/ SVP:***
A man wearing a white hat and sunglasses.

**Instruction:** Provide a short description for this region.

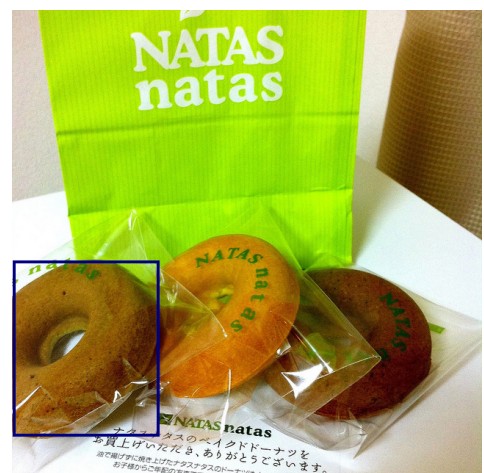

***VLM response w/o SVP:***
The image shows a bag of donuts with the brand name "NATAS" printed on it. The donuts are visible through the clear plastic packaging, and there are three donuts in total. The donuts appear to be of a similar type, possibly a ring donut, and they are placed on a white surface. The background is blurred, but there is a hint of a beige or light brown object to the right of the bag. The text on the bag is in Japanese, and the donuts have a hole in the center, which is typical for ring donuts.

***VLM response w/ SVP:***
A donut in a bag.

**Instruction:** Provide a short description for this region.

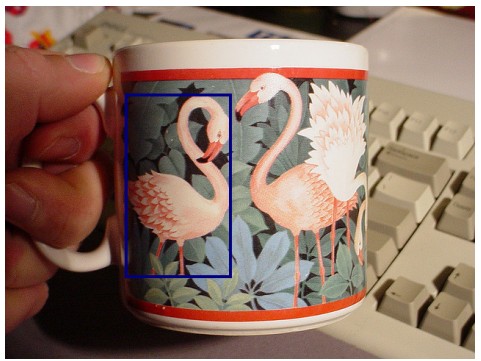

***VLM response w/o SVP:***
The image shows a person holding a mug with a flamingo design. The mug features two flamingos, one pink and one white, surrounded by green foliage and blue flowers. The design is colorful and playful, with the flamingos depicted in a stylized manner. The mug has a white handle and is placed in front of a computer keyboard, suggesting that the person might be using the computer while enjoying their drink. The background is blurred, but it appears to be an indoor setting with a desk or table.

***VLM response w/ SVP:***
A white coffee mug with a flamingo design.

**Instruction:** Provide a short description for this region.

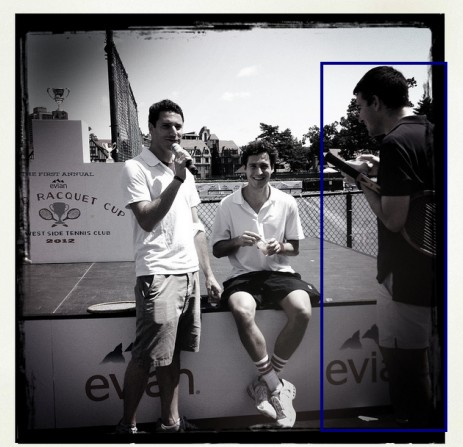

AUG   12

***VLM response w/o SVP:***
The image is a black and white photograph capturing a moment on a tennis court. In the foreground, there are two men standing on the court. The man on the left is holding a tennis racket and appears to be speaking into a microphone, suggesting he might be a commentator or announcer. He is wearing a white shirt and shorts, which is typical attire for a tennis player or official. The man on the right is seated on the edge of the court, holding a tennis racket and smiling, possibly a player or a coach. He is wearing a white shirt and black shorts. In the background, there is a fence separating the court from the spectator area, and beyond the fence, there are trees and buildings, indicating that the court is part of a larger sports complex or club. The sky is clear, suggesting it might be a sunny day. The image has a vintage or retro feel due to the black and white filter and the style of the clothing and the court's design. The text "THE FIRST ANNUAL WEST SIDE TENNIS CLUB" and "EVIAN RACQUET CUP 2012" are visible, indicating that this is an event sponsored by Evian, a brand known for its water products. The date "AUG 12" is also visible, which is likely the date of the event.The overall atmosphere of the image is casual and relaxed, with the focus on the tennis court and the people involved in the event.

***VLM response w/ SVP:***
A man holding a tennis racket.

**Instruction:** Provide a short description for this region.

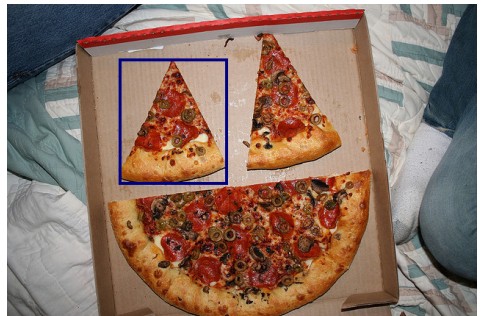

*VLM response w/o SVP:*
The pizza is in a box.

*VLM response w/ SVP:*
A slice of pizza.

**Instruction:** Provide a short description for this region.

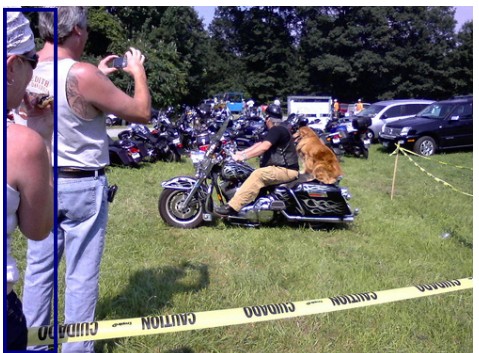

*VLM response w/o SVP:*
This is a photo of a man riding a motorcycle with a dog sitting on the back. The man is wearing a black helmet and a black vest, and the dog appears to be a golden retriever. The man is also wearing a black tank top and blue jeans. The motorcycle is black and has a large engine. The setting appears to be an outdoor event or gathering, as there are other motorcycles parked in the background and a yellow police tape is visible in the foreground. The man is taking a photo of the scene with his cell phone.

*VLM response w/ SVP:*
A man taking a picture of a dog sitting on a motorcycle.

## J.3  VISUAL QUERIES

**Instruction**: What activity could potentially be enjoyed near the area shown in the image?

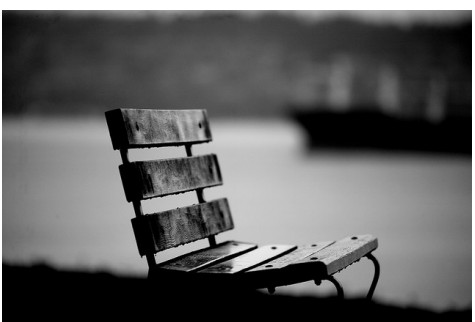

*VLM response:*
The image shows a wooden bench situated near a body of water, possibly a lake or a river. Given the serene and peaceful setting, one could potentially enjoy various activities such as: Sitting on the bench to relax, read a book, or simply enjoy the view of the water and the surrounding landscape. Taking a leisurely walk or jog along the water's edge, enjoying the tranquility and the natural beauty of the area. Engaging in photography, capturing the picturesque scenery and the reflections on the water's surface. Bird watching, as the area might be home to various bird species. Fishing, if the water body is a lake or a river with fish populations. Picnicking with friends or family, enjoying a meal outdoors in the fresh air and the calming presence of the water. These activities are suitable for the setting depicted in the image, offering a chance to unwind and connect with nature.

**Instruction**: What are some factors contributing to the difficulty of this activity?

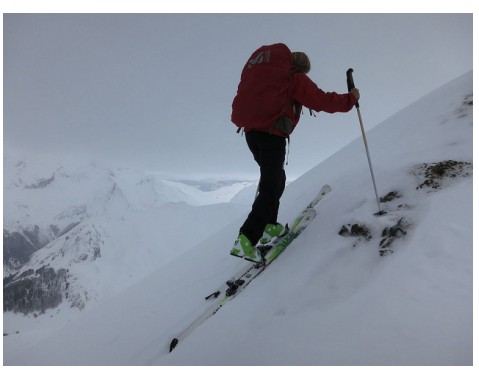

*VLM response:*
The image shows a person skiing on a steep, snow-covered slope. The difficulty of this activity can be attributed to several factors: Terrain: The steepness of the slope increases the challenge of maintaining balance and control while skiing. Steeper slopes require more advanced skiing techniques and greater physical strength. Snow Conditions: The type of snow can greatly affect the difficulty of skiing. In the image, the snow appears to be well-packed, which can make it harder to carve turns and maintain speed. Visibility: The image shows a cloudy day, which can reduce visibility and make it more difficult to navigate the slope. Equipment: The skier is wearing a backpack, which can add extra weight and affect balance and maneuverability. Weather: The cloudy day suggests that the weather conditions might be challenging, with the possibility of poor visibility, cold temperatures, or even snowfall. Skill Level: The skier's skill level plays a significant role in the difficulty of the activity. More advanced skiers are better equipped to handle steep slopes and challenging

# K DETAILS

## K.1 BASELINES

We conduct a comprehensive comparison against various baselines, including models fine-tuned with self-captioning without grounding and preference-based adaptation methods. Our evaluation encompasses a wide range of model scales (.5B, 7B, 8B, 13B, 19B, 40B parameters), architectures (LLaVA-1.5 (Liu et al., 2024a), LLaVA-1.6 (Liu et al., 2024b), LLaVA-OV (Li et al., 2024), VILA (Lin et al., 2024), InternVL (Chen et al., 2024b)), visual encoders (CLIP (Radford et al., 2021), SigLIP (Zhai et al., 2023), ViT (Dosovitskiy et al., 2020)), language encoders (Vicuna (Chiang et al., 2023), Mistral (Jiang et al., 2023), Qwen2 (Yang et al., 2024), Yi-2 (Young et al., 2024)), and scoring mechanisms $S(q, p)$ and $\Delta(q, p)$.

## K.2 IMPLEMENTATION DETAILS

We implement two SVP variants: SVP (C) using only grounded self-generated captions, and SVP (CVQ) which additionally incorporates visual queries from the model's training history to prevent over-specialization on descriptive tasks. For the sampling loop, we generate $K$ (20 for LLaVA-1.5 and LLaVA-1.6, 10 for LLaVA-OV) samples per image from both base and grounded VLMs, selecting the `top-k` (20% for LLaVA-1.5 and LLaVA-1.6, and 10% for LLaVA-OV) using our scoring mechanisms (Eq. 5, 6).

For example, with LLaVA-1.6 and $C = 1000$ images, we collect 4000 samples for SVP (C) and double this for SVP (CVQ) by including visual queries, yielding 8000 total training pairs. While smaller than typical supervised datasets, this proves sufficient for effective model adaptation (Sun et al., 2023; Zhu et al., 2024). We use normalized `xyxy` bounding boxes and filter out degenerate samples ($< 0.5\%$ for LLaVA-1.5/1.6, 5% for LLaVA-OV), with $w_{v,t} = q_{v,t}$.

For the adaptation loop, we fine-tune using LoRA (Hu et al., 2021) ($\alpha = 16$, $r = 64$ for $\leq$ 7b models; $\alpha = 256$, $r = 128$ for 13b models) for one epoch on 8-A100 GPUs with batch size $B = 20$. Following (Li et al., 2024; Liu et al., 2024b), we run up to 3 iterations of SVP. Our evaluation uses sample-wise, zero-shot testing without prompt engineering or batching to ensure fair comparison across model variants.

## K.3 METRICS

We use the CIDEr score (Vedantam et al., 2015) for captioning and referring tasks; accuracy for VQA and multitasking. F1, Accuracy and Recall for hallucination and object recall. We also consider standard metrics for language translation like BLEU (Papineni et al., 2002), METEOR (Banerjee & Lavie, 2005), and ROUGE (Lin, 2004) scores. We re-compute metrics for LLaVA baselines and variants (1.5, 1.6, OV) up to 13b parameters.

## K.4 DATASETS

**Table 17:** Datasets utilized in SVP. We use COCO2014 images as conditioning for building visual projections, using self-captioning and grounding feedback. VP: visual projection. VQA: visual question answering. REG: referring expression generation. We use `lmms-eval` (Zhang et al., 2024b) for all the evaluations. The `lite` splits as proposed in (Zhang et al., 2024b).

| Dataset | Task | Split | $N$ |
|---|---|---|---|
| *Sampling* | | | |
| COCO2014 (Lin et al., 2014) | VP | `train` | 100:10000 |
| | | | |
| *Evaluation* | | | |
| ScienceQA (Saikh et al., 2022) | VQA | `test` | 4241 |
| GQA (Hudson & Manning, 2019) | VQA | `lite` | 500 |
| COCO2017 (Lin et al., 2014) | Captioning | `val_lite` | 500 |
| Flickr30k (Plummer et al., 2015) | Captioning | `test_lite` | 500 |
| NoCaps (Agrawal et al., 2019) | Captioning | `val_lite` | 500 |
| COCO2014 (Lin et al., 2014) | Captioning | `val` | 40504 |
| COCO2017 (Lin et al., 2014) | Captioning | `val` | 5000 |
| Flickr30k (Plummer et al., 2015) | Captioning | `test` | 31783 |
| NoCaps (Agrawal et al., 2019) | Captioning | `val` | 4500 |
| RefCOCO (Kazemzadeh et al., 2014) | REG | `val_lite` | 500 |
| RefCOCO (Kazemzadeh et al., 2014) | REG | `val` | 8811 |
| RefCOCO (Kazemzadeh et al., 2014) | REG | `test` | 5000 |
| RefCOCO (Kazemzadeh et al., 2014) | REG | `testA` | 1975 |
| RefCOCO (Kazemzadeh et al., 2014) | REG | `testB` | 1810 |
| RefCOCO+ (Kazemzadeh et al., 2014) | REG | `val` | 3805 |
| RefCOCO+ (Kazemzadeh et al., 2014) | REG | `testA` | 1975 |
| RefCOCO+ (Kazemzadeh et al., 2014) | REG | `testB` | 1798 |
| RefCOCOg (Kazemzadeh et al., 2014) | REG | `val` | 7573 |
| RefCOCOg (Kazemzadeh et al., 2014) | REG | `test` | 5023 |
| MMBench (Liu et al., 2025) | Multitasking | `en_dev_lite` | 500 |
| MMMU (Yue et al., 2024) | Multitasking | `val` | 900 |
| POPE (Li et al., 2023b) | Hallucinations | `adv` | 3000 |
| POPE (Li et al., 2023b) | Hallucinations | `pop` | 3000 |
| POPE (Li et al., 2023b) | Hallucinations | `random` | 3000 |

## K.5 HYPERPARAMETERS

**Table 18:** Hyper-parameters for the main experiments.

|  | LLaVA-1.5-13b | LLaVA-1.6-7b | LLaVA-1.6-13b | LLaVA-OV-0.5b | LLaVA-OV-7b |
|---|---|---|---|---|---|
| *Sampling* |  |  |  |  |  |
| images | 1000 | 1000 | 1000 | 2000 | 2000 |
| iterations | 1 | 1 | 1 | 1 | 1 |
| prompt-version | llava_v1 | mistral_instruct | llava_v1 | qwen_1_5 | qwen_1_5 |
| sample-batch | 20 | 20 | 20 | 10 | 10 |
| samples/image | 20 | 20 | 20 | 10 | 10 |
| top $k$ | 0.2 | 0.2 | 0.2 | 0.1 | 0.1 |
| *Training* |  |  |  |  |  |
| accelerators | A100 | A100 | A100 | A100 | A100 |
| deepspeed | w/ ZeRO-2 | w/ ZeRO-3 | w/ ZeRO-3 | w/ ZeRO-3 | w/ ZeRO-3 |
| epochs | 1 | 1 | 1 | 3 | 3 |
| grad-acc | 1 | 1 | 1 | 2 | 2 |
| learning-rate | $2e^{-4}$ | $2e^{-4}$ | $2e^{-4}$ | $1e^{-5}$ | $1e^{-5}$ |
| lora | w/ | w/ | w/ | w/ and w/o | w/ and w/o |
| lora-$\alpha$ | 256 | 16 | 256 | 16 | 16 |
| lora-r | 128 | 64 | 128 | 64 | 64 |
| lr-schedule | cos | cos | cos | cos | cos |
| max-tokens | 2048 | 2048 | 2048 | 1024 | 1024 |
| mix-precision | w/ | w/ | w/ | w/ | w/ |
| optimizer | AdamW | AdamW | AdamW | AdamW | AdamW |
| samples | 4000:8000 | 4000:8000 | 4000:8000 | 2000 | 2000 |
| text-encoder | Vicuna-13b-v1.5 | Mistral-7b-Instruct-v0.2 | Vicuna-13b-v1.5 | Qwen2-0.5b | Qwen2-7b |
| train-batch | 16 | 16 | 16 | 4 | 4 |
| vision-encoder | CLIP-L/14 | CLIP-L/14 | CLIP-L/14 | SigLIP-SO/14 | SigLIP-SO/14 |
| warm-up-rate | 0.03 | 0.03 | 0.03 | 0.03 | 0.03 |

