# OpenReview forum: "Grounding as Feedback: Vision-Language Alignment via Sampling-based Visual Projection"
_ICLR.cc/2026/Conference — Submitted to ICLR 2026_

### Official Review · Reviewer_CBKU · 2025-10-21

**Soundness:** 3
**Presentation:** 2
**Contribution:** 2
**Rating:** 4
**Confidence:** 5

**Summary:**

The paper tackles unreliable vision–language alignment in VLMs (object hallucinations, missed objects) and proposes Sampling-based Visual Projection (SVP), a guided self-improvement framework. SVP (i) has the base VLM self-caption images, (ii) feeds those captions to an external grounding model (e.g., GroundingDINO) to obtain spatial feedback, then (iii) scores, selects, and fine-tunes the VLM on the highest-quality, feedback-refined samples using lightweight LoRA. Two scoring variants (log-ratio and weighted-difference) identify captions where grounding most changes the model’s token probabilities; two training variants are explored: The approach avoids new manual annotations or preference data, but depends on a competent grounding model and brings benefits to spatially grounded tasks.

**Strengths:**

1.	Quality and breadth. The paper reports consistent gains across ten standard benchmarks spanning six task families (captioning, referring expressions, hallucination control, object recall, VQA, and multitask), with results summarized clearly and backed by extensive comparisons.
2.	Clarity and reproducibility. The writing is generally clear, with informative figures/graphical models, explicit algorithm overviews, and implementation details. The appendices provide pseudo-code, prompt templates, and full hyperparameters, which materially aid reproducibility.

**Weaknesses:**

1.	The comparison is unfair. Although there are no new human annotations, the inner loop explicitly constructs a curated pseudo-labeled set guided by the grounder, which functions as additional supervision.
2.	Evaluation granularity for localization. Referring performance is summarized with language-centric metrics (e.g., CIDEr), but the paper does not report region-level localization metrics (e.g., bbox/mask IoU or Acc@0.5) that would more directly substantiate the “spatial alignment” claims.
3.	Dependence on an external grounder. The pipeline hinges on a pretrained grounding model (e.g., GroundingDINO-tiny) to generate feedback and curate data, which may limit portability in domains lacking strong grounders or where their biases are problematic.

**Questions:**

1.	Is this method essentially a method of using ground truth data for data augmentation? That is, after the location output of GDINO is textified, "more easily aligned" pseudo-label pairs are constructed and selected, which can be regarded as "grounded-driven data augmentation + selection", and then fine-tuned?
2.	Add localization metrics. Could you report standard localization metrics (e.g., bbox/mask IoU, Acc@0.5 on RefCOCO/+/g) in addition to CIDEr, to directly evidence spatial alignment improvements?
3.	Top-fraction discrepancy. Figure 5 states that the inner loop “selects the top 20%,” whereas §4.1/Implementation Details say the pipeline “selects the top 10%.” Which fraction is actually used for your headline results and for ablations? Please reconcile these settings.

---

> ### Author Response · Authors · 2025-11-22
>
> **Reviewer**:
>
> > Although there are no new human annotations, the inner loop explicitly constructs a curated pseudo-labeled set guided by the grounder, which functions as additional supervision.
>
> **Authors**:
>
> We appreciate the opportunity to clarify. We respectfully disagree with the characterization of SVP as "additional supervision" in the traditional sense for three main reasons:
>
> 1. **Self-Generation**: SVP generates all textual samples itself.
> 2. **No External Task Labels**: The grounding model operates on the VLM's own captions; it does not introduce new labels or external ground truth text for the downstream tasks.
> 3. **Feedback vs. Supervision**: The VLM learns from alignment feedback, not external targets. The novelty lies in demonstrating that leveraging this feedback loop improves alignment without requiring curated preference pairs or human annotation.
>
> We want to clarify that the novelty of our work does not lie in the training loss or the fine-tuning procedure. Rather, *our primary contribution is demonstrating that leveraging feedback from a grounding model can significantly improve vision-language alignment without requiring additional external data or additional human annotation or curated preference pairs*.
>
> **Reviewer**:
>
> > Evaluation granularity for localization. Referring performance is summarized with language-centric metrics (e.g., CIDEr), but the paper does not report region-level localization metrics (e.g., bbox/mask IoU or Acc@0.5) that would more directly substantiate the “spatial alignment” claims.
>
> **Authors**:
>
> `[Appendix E.2, Figure 15, Figure 16]` Our VLM outputs text, not bounding boxes, and our main focus is vision-language alignment, which is why we prioritized text metrics (CIDEr/METEOR/Accuracy). However, to directly validate spatial alignment, we computed the IoU on a RefCOCO valset by parsing the VLM's output and verifying it against ground truth boxes via the grounding model. These results have been included in the revised paper. See also `[Appendix E.3, Figure 17]` for more IoU ablations using different grounding models and IoU thresholds.
>
> **Reviewer**:
>
> > Dependence on an external grounder.
>
> **Authors**:
>
> `[Appendix E.3, Figure 17]` We believe this is a feature. We show that we can use cheap feedback from a small grounding model to improve vision-language alignment. We argue that grounding and object detection is a far more mature and "commoditized" technology than generative VLM reasoning. High-quality open-set detectors (like GDINO or GLIP) are widely available and computationally cheap. In domains with no detection capability, SVP would require a different source of feedback, but the core idea of feedback-driven self-training for vision-language alignment would still hold.
>
> **Reviewer**:
>
> > Is this method essentially a method of using ground truth data for data augmentation? That is, after the location output of GDINO is textified, "more easily aligned" pseudo-label pairs are constructed and selected, which can be regarded as "grounded-driven data augmentation + selection", and then fine-tuned?
>
> **Authors**:
>
> While SVP may superficially resemble data augmentation, it is fundamentally distinct in that *all textual information is self-generated*. The process operates as a "closed loop":
>
> * Given an image prompt, the VLM generates a draft caption, which is processed by a grounding model to provide spatial feedback (notice that the textual information in input/output is fully contained in the original self-generated caption).
>
> * This feedback is returned to the VLM as context, enabling it to self-generate a refined caption.
>
> * During the adaptation phase, the VLM is tuned exclusively on these improved captions, without access to bounding boxes or external annotations. In essence, the model creates its own ground truth through feedback.
>
> Therefore, we frame our work as feedback-driven self-training , which operates as a simplified form of policy gradient, to improve vision-language alignment.
>
> **Reviewer**:
>
> > Could you report standard localization metrics (e.g., bbox/mask IoU, Acc@0.5 on RefCOCO/+/g) in addition to CIDEr, to directly evidence spatial alignment improvements?
>
> **Authors**:
>
> `[Appendix E.2, Figure 15, Figure 16]`  Our VLM outputs text, not boxes, so we focused on text metrics. However, to validate spatial alignment, we computed IoU on the RefCOCO val set by parsing the VLM's output and passing it through the grounding model against ground truth boxes. These results have been added to the manuscript.
>
> **Reviewer**:
>
> > Top-fraction discrepancy.
>
> **Authors**:
>
> Thank you for catching this. We used Top-20% for LLaVA-1.5/1.6 and Top-10% for LLaVA-OV. We have unified the text in Section 4.1 to reflect the exact hyper-parameters used.

---

### Official Review · Reviewer_hcG4 · 2025-10-31

**Soundness:** 2
**Presentation:** 3
**Contribution:** 2
**Rating:** 4
**Confidence:** 5

**Summary:**

This paper focuses how to address VLMs's hallucination and information omission by proposing the Sampling-based Visual Projection (SVP). Unlike existing methods (e.g., SFT) that rely on expensive human annotations or curated preference data, SVP leverages a pre-trained grounding model (GroundingDINO-tiny here) as an expert feedback tool to enable VLM self-improvement.​ The SVP framework operates in three iterative stages: (1) Prior Sampling, where the base VLM generates flawed descriptions without grounding guidance; (2) Grounding Feedback, where the grounding model provides object bounding boxes, category confidence scores to identify hallucinations or omissions; (3) Guided Sampling, where the VLM generates refined, alignment-enhanced descriptions using the grounding feedback. These generated samples are then scored via log-ratio or weighted difference metrics to select high-quality pairs (Top 10–20%), which are used to fine-tune the VLM via LoRA (low-rank adaptation).​ Evaluations across benchmarks demonstrate SVP’s efficacy: it achieves an average 14% improvement in caption generation (CIDEr score), a 12% boost in object recall, and a POPE F1 score of 88.33 (reducing hallucinations). Notably, SVP preserves multi-task capabilities (e.g., VQA on ScienceQA) and generalizes to diverse VLM architectures (LLaVA-1.5/1.6/OV) and scales (0.5B–13B parameters).

**Strengths:**

Annotation Efficiency: SVP eliminates the need for additional human labels or preference data by using self-generated samples and machine feedback from grounding models. This addresses a major cost bottleneck of existing VLM optimization methods (e.g., SFT requires millions of annotated image-text pairs).

Multi-Task Preservation: Unlike methods that overfit to single tasks (e.g., caption-only fine-tuning), SVP maintains VQA performance (ScienceQA accuracy: 78.54 → 78.40) and even improves multi-task capabilities (MMBench accuracy +9.76%), making it suitable for real-world multi-modal applications.

**Weaknesses:**

Missing 3D Depth and Irregular Object Size Cues: The grounding model relies on 2D bounding boxes (bboxes) to provide spatial feedback, which only captures planar (left/right, top/bottom) coordinates, however,  3D depth information (e.g., "the cup in front of the book" vs. "the book behind the cup") is entirely absent. This leads to ambiguous spatial descriptions for scenes with overlapping objects. Additionally, 2D bboxes fail to represent irregularly shaped objects: the rectangular bbox includes irrelevant background pixels, making it impossible for the VLM to infer the object’s actual size or shape (e.g., describing a small coiled cable as a "large rectangular object").

Unaddressed Efficiency Comparison with Automated High-Quality Data SFT: SVP’s core value lies in avoiding manual annotation via self-evolution, but it does not compare with an alternative scenario: automated generation of high-quality annotated data (e.g., Image-Textualization [NeurIPS 2024]). If such automated data is available, a single-step SFT (instead of SVP’s iterative sampling-scoring-adaptation) might be more efficient since it requires less computational overhead (no multi-round sampling) and shorter training cycles (direct parameter update with high-quality data). The paper’s failure to evaluate this trade-off leaves a gap in assessing SVP’s efficiency in data-rich, automated annotation scenarios.​

Dependence on Grounding Model Quality: SVP’s performance is tightly coupled to the grounding model’s accuracy. If the grounding model fails (e.g., misdetecting objects in occluded/blurry images or misclassifying rare items), the feedback becomes unreliable, degrading SVP’s optimization effect. The paper does not test SVP with low-precision grounding models, limiting its robustness analysis.​

**Questions:**

1. The paper only uses GroundingDINO-tiny (a lightweight model) for feedback. How would SVP perform with more powerful grounding models? Would higher-precision feedback further enhance alignment, or introduce over-reliance on the grounding model’s biases?

2. Given the computational overhead of K=20 samples, is there a way to dynamically adjust K while maintaining SVP’s performance? Could techniques like active learning select the most informative samples to minimize K?

3. Would SVP extend to video-language models (e.g., Video-LLaVA)? Video introduces temporal dynamics (e.g., object motion) that static grounding models cannot capture, which is more exciting than just focusing on static images.

4. The paper tests up to 3 SVP iterations. Could multi-round iterations (e.g., 5+ rounds) lead to overfitting to the grounding model’s systematic misdetections? How to balance alignment improvement and overfitting prevention in long-term self-improvement?

5. The paper evaluates SVP on daily-scene datasets (COCO, Flickr30k). Would SVP work for specialized domains (e.g., remote sensing, pathology), where objects have unique visual features (e.g., medical scans of tumors)? Would pre-training the grounding model on domain-specific data be a prerequisite?

6. How could SVP integrate 3D grounding feedback (e.g., from depth estimation models like MonoDepth + GroundingDINO, or 3D detectors like VoteNet) to capture depth relationships? For irregular objects, would replacing 2D bboxes with segmentation masks (seg masks) improve size/shape inference, and how would this change the scoring metric (e.g., log-ratio adapted to mask overlap instead of bbox coordinates)?

7. For cluttered scenes (e.g., a desk with overlapping notebooks, pens, and a phone), would 3D depth feedback enable SVP to generate more accurate spatial descriptions (e.g., "the pen on top of the notebook" vs. "the notebook under the pen")? Could this reduce ambiguity in referring expressions (e.g., RefCOCO+ tasks requiring fine-grained spatial reasoning)?

8. If automated tools generate 100k high-quality image-text pairs (with 3D depth, size, and shape annotations), how would a single-step SFT compare to SVP?

---

> ### Author Response · Authors · 2025-11-22
> **Response (i)**
>
> **Reviewer**:
>
> > Missing 3D Depth and Irregular Object Size Cues
>
> **Authors**:
>
> Thank you for the comment. You are correct that 2D bounding boxes lack explicit depth. However, the goal of SVP is not to build a better grounding model or to leverage the richest form of feedback, but to show that even *relatively weak feedback can be used to improve vision-language alignment*: SVP improves alignment accurately locating objects (x,y), the VLM naturally infers occlusion and relative size better than without any grounding (see `Appendix J` for qualitative examples). The primary goal is preventing the generation of non-existent objects. 2D grounding is sufficient for this.
>
> **Reviewer**:
>
> > Unaddressed Efficiency Comparison with Automated High-Quality Data SFT
>
> **Authors**:
>
> `[Section 4, Table 3]` we compare with multiple high-performing model that leverage large-scale SFT training (top), inference-time scaling (middle) and inference-time adaptation (bottom) mechanisms.
>
> **Reviewer**:
>
> > SVP’s core value lies in avoiding manual annotation via self-evolution, but it does not compare with an alternative scenario: automated generation of high-quality annotated data (e.g., Image-Textualization [NeurIPS 2024]).
>
> **Authors**:
>
> `[Appendix E.1, Table 10]` We have added a comparison with Image Textualization (IT) in `Appendix E.1`. SVP achieves a POPE F1 of 88.3 (using only 4k samples) compared to IT's 86.4 (using 10k samples). SVP is more data-efficient because it adapts the model to its own distribution (self-correction) rather than fitting external synthetic data. Additionally, SVP requires a weak form or feedback (small grounding model), where IT relies on multiple source of feedback.
>
> **Reviewer**:
>
> > Dependence on Grounding Model Quality: SVP’s performance is tightly coupled to the grounding model’s accuracy.
>
> **Authors**:
>
> `[Appendix E.3, Figure 17]` We intentionally selected GDINO-tiny to demonstrate that SVP functions effectively even with lightweight, imperfect feedback. We run ablations with different grounding model architectures and "noisy" grounders to further quantify robustness.
>
> **Reviewer**:
>
> > The paper only uses GroundingDINO-tiny (a lightweight model) for feedback. How would SVP perform with more powerful grounding models? Would higher-precision feedback further enhance alignment, or introduce over-reliance on the grounding model’s biases?
>
> **Authors**:
>
> Thank you for the question. We run experiments with different grounding quality and we added them in the paper revision. See `[Appendix E.3, Figure 17]`.
>
> **Reviewer**:
>
> > Given the computational overhead of K=20 samples, is there a way to dynamically adjust K while maintaining SVP’s performance? Could techniques like active learning select the most informative samples to minimize K?
>
> **Authors**:
>
> `[Appendix E.4, Figure 18]` Sampling 20 textual samples given one image is much cheaper than sampling 10 textual continuation given 10 images. We run ablations with [1,5,10,20] and added them in the revised manuscript.
>
> **Reviewer**:
>
> > Would SVP extend to video-language models (e.g., Video-LLaVA)? Video introduces temporal dynamics (e.g., object motion) that static grounding models cannot capture, which is more exciting than just focusing on static images.
>
> **Authors**:
>
> This is an excellent suggestion for future work. Conceptually, SVP is modality-agnostic. To extend SVP to video, one would replace the static grounder with a video-grounding model (or tracking) to provide temporal feedback (e.g., "person moves left to right"). The VLM would then self-correct its temporal descriptions based on this trajectory data.
>
> **Reviewer**:
>
> > The paper tests up to 3 SVP iterations. Could multi-round iterations (e.g., 5+ rounds) lead to overfitting to the grounding model’s systematic misdetections? How to balance alignment improvement and overfitting prevention in long-term self-improvement?
>
> **Authors**:
>
> `[Section 4, Figure 8]` Our results show that gains plateau after 2-3 iterations. We hypothesize this occurs when the "alignment gap" closes. Using a "weak" expert like GDINO-tiny acts as a regularizer, preventing the VLM from overfitting to high-granularity (and potentially erroneous) boxes, instead encouraging general spatial consistency.
>
> **Reviewer**:
>
> > The paper evaluates SVP on daily-scene datasets (COCO, Flickr30k). Would SVP work for specialized domains (e.g., remote sensing, pathology), where objects have unique visual features (e.g., medical scans of tumors)? Would pre-training the grounding model on domain-specific data be a prerequisite?
>
> **Authors**:
>
> SVP relies on the grounding model to act as a scoring/verifier. Therefore, for specialized domains where generalist object detectors fail (e.g., identifying specific tumor types or satellite terrain features), a domain-specific grounding model or specialized source of feedback would probably be necessary.

---

> > ### Author Response · Authors · 2025-11-22
> > **Response (ii)**
> >
> > **Reviewer**:
> >
> > > How could SVP integrate 3D grounding feedback...For irregular objects, would replacing 2D bboxes with segmentation masks (seg masks) improve size/shape inference, and how would this change the scoring metric (e.g., log-ratio adapted to mask overlap instead of bbox coordinates)?
> >
> > **Authors**:
> >
> > SVP is designed to interface with the VLM via natural language context. To integrate 3D feedback, we would simply update the sampling stage and condition on 3D coordinates and depth information.
> >
> > Regarding segmentation masks: Using masks would likely improve the handling of irregular objects. The modification would happen in the grounding step (extracting masks). The scoring metric $S(q,p)$ operates on the text tokens generated by the VLM, not the visual boxes directly. Therefore, the scoring mechanism remains unchanged: it calculates the probability shift in the text tokens given the new, richer (3D or mask-based) context.
> >
> > **Reviewer**:
> >
> > > For cluttered scenes (e.g., a desk with overlapping notebooks, pens, and a phone), would 3D depth feedback enable SVP to generate more accurate spatial descriptions?
> >
> > **Authors**:
> >
> > Yes, we believe this is a natural extension of the method. Currently, SVP infers occlusion and depth implicitly through 2D overlaps. Explicit depth feedback provided in the context window would reduce ambiguity in cluttered scenes. By feeding depth relationships into the prompt during the sampling phase, the VLM would learn to generate captions that explicitly describe layers and occlusion, effectively distilling 3D understanding into the text generator.
> >
> > **Reviewer**:
> >
> > > If automated tools generate 100k high-quality image-text pairs (with 3D depth, size, and shape annotations), how would a single-step SFT compare to SVP?
> >
> > **Authors**:
> >
> > We address a similar comparison in our new ablation with Image Textualization (IT) in `[Appendix E.1, Table 10]` and LLaVA-base models in `[Section 4, Table 3]` . While SFT on 100k synthetic pairs is a valid approach, it forces the model to learn a new distribution (the synthetic data's style). SVP, by contrast, achieved higher hallucination reduction (88.3 F1 vs 86.4 F1) using only 4k samples (vs 10k or 50k for IT). This is because SVP aligns the model's own latent knowledge. It acts as self-correction rather than new learning. SVP is more sample-efficient because it focuses specifically on the examples where the model's text and the visual ground truth disagree, rather than retraining on a massive, potentially redundant external dataset.

---

### Official Review · Reviewer_nkcU · 2025-10-31

**Soundness:** 2
**Presentation:** 1
**Contribution:** 2
**Rating:** 4
**Confidence:** 4

**Summary:**

This paper aims to address hallucinations in VLMs concerning objects, locations, and their relationships. It proposes a method that uses a grounding model to correct the VLM's intermediate reasoning steps. This approach has shown good results on LLaVA-based models.

**Strengths:**

Using an expert model, like a grounding model, to fix the VLM's internal steps is a good idea to reduce hallucinations. Besides, the approach achieves good results on the benchmarks.

**Weaknesses:**

1. A major weakness of this paper is its lack of clarity. For example, the authors should explicitly describe the inputs and outputs for each stage of their method in a clear, structured way. Currently, this information is scattered throughout the text, and in some cases, it seems to be missing entirely. This forces the reader to search for details or guess how the components connect, which makes the method very difficult to understand and reproduce.
2. The experiments are a bit limited because they only use LLaVA-based models. The method itself seems general and should work for other models too. It would be much more convincing if the authors also showed results on different models, like Qwen-VL, to prove the method is widely useful.
3. I am confused by the strong results on ScienceQA shown in Table 3. This benchmark seems to focus on science questions, not the object and location hallucinations this paper targets.
Why does the method work so well here? This needs to be explained. It makes me wonder if the performance gain is coming from another reason that isn't mentioned in the paper. The authors should clarify this point.
4. The paper needs to explain how it gets the noun list for grounding (from Figure 5). This step is not described. What happens if this process misses important nouns? The authors should report the recall of their noun extraction method.
5. The paper does not explain how the method disambiguates between multiple instances of the same object category (e.g., several different people). It is unclear how GDINO can ground the specific instance being referred to. I think it is a crucial limitation of this paper.

**Questions:**

Please refer to the issues detailed in the Weakness part.

---

> ### Author Response · Authors · 2025-11-22
>
> **Reviewer**:
>
> > A major weakness of this paper is its lack of clarity. For example, the authors should explicitly describe the inputs and outputs for each stage of their method in a clear, structured way.
>
> **Authors**:
>
> We sincerely appreciate this constructive feedback. To ensure immediate clarity and reproducibility, we have extensively restructured `Section 3`:
>
> * **Algorithm Specification**: We added `Algorithm 1` to explicitly detail the execution loop: Sample Prior $\rightarrow$ Ground $\rightarrow$ Grounded Sample $\rightarrow$ Score $\rightarrow$ Adapt.
> * **Symbol Definitions**: We introduced `Table 1` to introduce all the variables ($c, z_p, g, z, q, p_\theta$) alongside concrete examples.
> * **Data Flow Clarity**: We rewrote the caption for Figure 5 to trace the exact data transformation steps through the pipeline.
>
> *We believe this revised presentation resolves the ambiguity you identified*. We hope these improvements allow for a re-evaluation of our work's technical merit and the breadth of our experimental results. We remain fully open to further suggestions to ensure the method section is as clear as possible.
>
> **Reviewer**:
>
> > The experiments are a bit limited because they only use LLaVA-based models. The method itself seems general and should work for other models too. It would be much more convincing if the authors also showed results on different models, like Qwen-VL, to prove the method is widely useful.
>
> **Authors**:
>
> `[Section 4, Tables 7, Table 8]` Thank you for highlighting this limitation. We have expanded our evaluation to include Qwen2.5-VL and Qwen2-VL. Crucially, our results demonstrate that massive SFT data scaling alone is insufficient to resolve vision-language alignment.
>
> As shown in `Table 7`, despite being trained on orders of magnitude more data (~100M multimodal samples for Qwen2-VL vs. <1M for LLaVA-1.6), the Qwen2.5-VL model still underperforms our SVP-adapted LLaVA-1.6 in both hallucination reduction (86.2 vs. 88.3 F1) and object recall (77.9 vs. 84.4).
>
> Furthermore, `Table 8` shows that applying SVP directly to Qwen2.5-VL yields immediate gains (increasing F1 to 89.1), confirming that our framework is architecture-agnostic and that data scaling alone does not solve the alignment problem.
>
> We initially prioritized the LLaVA family because its transparent training distribution allows for controlled analysis, whereas Qwen's closed data precludes verifying exposure to evaluation sets like POPE or RefCOCO. Qualitatively, we observed that the base Qwen2-VL still struggles with fine-grained localization - such as missing the partial bus in our qualitative examples - whereas the SVP-adapted model correctly identifies such elements. These findings substantiate that feedback-driven alignment is a critical post-training step for even the most powerful models, and we have integrated these results into the main paper to demonstrate the generality of our approach.
>
> **Reviewer**:
>
> > I am confused by the strong results on ScienceQA shown in Table 3. This benchmark seems to focus on science questions, not the object and location hallucinations this paper targets.
>
> **Authors**:
>
> Thank you for this insightful question. We hypothesize that SVP improves ScienceQA performance by training the model to attend to fine-grained visual details. ScienceQA diagrams often rely on specific, subtle visual cues - such as arrows, labels, and spatial relationships - to derive the correct answer. By mitigating the model's tendency to hallucinate or overlook these grounding elements, SVP creates a more faithful representation of the visual input, which naturally aids diagrammatic QA.
>
> At the same time, because SVP does not introduce new domain knowledge (e.g., scientific facts), we do not expect large improvements on VQA. Our primary goal was to enhance alignment while preserving the model's base capabilities; however, these results suggest that robust visual grounding is a prerequisite for accurate VQA, leading to the observed gains.
>
> **Reviewer**:
>
> > The paper needs to explain how it gets the noun list for grounding (from Figure 5).
>
> **Authors**:
>
> We directly use the output of GDINO without any post-processing (Figure 5 is a simplification of the pipeline). The input to the grounding model is the raw output of the VLM. Notice that GDINO can only ground nouns presented in the input image. We feed the entire raw caption to GDINO; we do not use a separate extractor. GDINO handles text-to-box matching end-to-end.
>
> **Reviewer**:
>
> > The paper does not explain how the method disambiguates between multiple instances of the same object category (e.g., several different people).
>
> **Authors**:
>
> GDINO is context-aware. If the caption says "the person on the left," GDINO attempts to ground that specific phrase. If the grounding is ambiguous (low confidence), it is filtered out automatically by the grounding model. We do not modify the default behavior for GDINO in any way to minimize introducing bias.

---

### Official Review · Reviewer_tCTj · 2025-11-06

**Soundness:** 2
**Presentation:** 3
**Contribution:** 2
**Rating:** 4
**Confidence:** 3

**Summary:**

This paper proposes Sampling-based Visual Projection (SVP), a method for improving vision-language alignment in VLMs by using a existing grounding model to provide feedback on self-generated captions. Specifically, the proposed method use iteratively sampling descriptions, scoring them based on grounding feedback, and fine-tuning the model on high-quality samples. Experiments using LLaVA series models demonstrate improvements on image captioning and hallucination reduction.

**Strengths:**

1. This paper investigates a well-understood problem in VLMs: the generated output may be inaccurate to the object, i.e., object hallucination.
2. The proposed method avoids the need for new, large-scale human annotations or explicit preference pairs, which can be bottlenecks for alignment methods such as SFT and DPO.
3. The experiments involve diverse benchmarks, tasks and model scales, which are relatively solid.

**Weaknesses:**

1. The major concern about this paper is the core idea looks like distillation, where it distills some knowledge from the grounding model into the target VLMs. That means we still have to rely on a grounding model instead of truly self-improvement. The grounding model obviously requires training data (or even annotated).
2. SVP's performance is capped by the quality of the "expert" grounding model, which should be further analyzed. If the grounding model provides incorrect feedback, SVP can make the target model degenerate. Also, it seems like the method does not work well on Multitasking in table 3. It is also suggested to experiment with different grounding models.
3. Additionally, SVP requires K\=20 samples per image and iterative fine-tuning, making it much more costly than standard method. It would be good to see more analysis on this.
4. While I don't carefully check Appendix F, I think the connection to ELBO is informal and may even be wrong. SVP doesn't actually optimize the ELBO in any way. And the claim about "posterior inference" is overclaimed.

**Questions:**

Please see the weakness.

---

> ### Author Response · Authors · 2025-11-22
>
> **Reviewer:**
>
> > The major concern about this paper is the core idea looks like distillation, where it distills some knowledge from the grounding model into the target VLMs.
>
> **Authors**:
>
> We appreciate this insightful framing. We agree that SVP shares some conceptual roots with distillation, specifically in transferring the visual-language alignment of an expert into the VLM. However, we characterize SVP as `Feedback-Driven Self-Training` rather than traditional knowledge distillation for three fundamental reasons:
>
> 1. **Self-Captioning**: In standard distillation, the student mimics the teacher's output tokens. In SVP, the VLM generates its own tokens. The grounding model acts as a “verifier” (providing spatial coordinates to check consistency), not as a “generator” providing target text.
> 2. **Nature of Supervision**: The grounding model does not provide ground-truth labels. Instead, it provides cross-modal feedback (spatial grounding) on the VLM's self-generated captions to guide the selection of high-quality samples.
> 3. **Learning Mechanism**: The VLM improves by learning from its own feedback-refined outputs, rather than mimicking the behavioral distribution of a teacher model.
>
> `[line 114-120]` We have revised the manuscript to explicitly connect SVP to Alignment Distillation via self-training. We clarify that the grounding model functions as a fixed feedback source - analogous to a compiler providing error signals or a reward model - rather than a generative teacher.
>
> **Reviewer**:
>
> > SVP's performance is capped by the quality of the "expert" grounding model, which should be further analyzed. If the grounding model provides incorrect feedback, SVP can make the target model degenerate. Also, it seems like the method does not work well on Multitasking in table 3. It is also suggested to experiment with different grounding models.
>
> **Authors**:
>
> `[Appendix E.3, Figure 17]` The key scientific question we address is whether spatial feedback improves alignment, not which grounding model is optimal. We view the grounding model as a fixed, low-cost tool. Our results with GDINO-tiny (a lightweight model not trained on COCO) demonstrate that SVP does not require strong feedback to be effective. The grounder only needs to identify discrepancies, not generate perfect captions. That said, we added to the Appendix a grounding robustness ablation using a "noisy" grounder (dropping a ratio of the grounded objects) to quantify this resilience.
>
> To our knowledge, we are the first to show that self-captioning and grounding feedback can effectively be amortized in a VLM, without requiring any *additional* source of annotation.
>
> **Reviewer**:
>
> > Additionally, SVP requires K=20 samples per image and iterative fine-tuning, making it much more costly than standard method. It would be good to see more analysis on this.
>
> **Authors**:
>
>  `[Appendix E.4, Figure 18]` We emphasize that the $K=20$ sampling overhead applies only to the one-time training phase, and sampling multiple textual continuation given a single (cached) image is fast and cheap no modern accelerators. We added ablations with K=[1, 5, 10, 20] in the Appendix..
>
> We challenge the notion that SVP is expensive for two main reasons:
>
> * **Efficiency**: SVP fine-tuning requires only ~1,000 images and completes in minutes on a single A100 GPU. Compared to SFT (typically performed with >100k of images) or RLHF, this compute cost is negligible. Sampling $K$ continuations given a single image is an extremely cheap task: given a single GPU, we can cache the image and sample (relatively short) textual content. It would be much more expensive to start with 20000 images and sample a single continuation per image: most GPU can only fit one or few high-resolution images, and parallelization would be impossible on standard hardware.
> * **Zero Inference Overhead**: Unlike search-based inference methods that require heavy compute per query, our adapted model uses standard sampling ($K=1$) during inference and deployment.
>
> **Reviewer**:
>
> > While I don't carefully check Appendix F, I think the connection to ELBO is informal
>
> **Authors**:
>
> We acknowledge the reviewer's concern and we removed this section in the Appendix. The core contributions of SVP remain in the empirical methodology and results. We wanted to use Appendix F to build intuition, but overall it seems to be confusing and not necessary for the core SVP framework formulation.

---

### Author Response · Authors · 2025-11-22
**General Response (i)**

We thank the reviewers for their insightful and constructive feedback. We are encouraged by your recognition of the critical importance of vision-language alignment and the potential of our feedback-driven framework. Guided by your consensus, we have significantly revised the manuscript to improve its clarity, positioning, and experimental robustness. New content is marked in `GREEN`.

We believe this revision effectively highlights our methodological contributions and demonstrates the breadth and depth of our experimental results. We welcome any further feedback during the discussion period.

### **Summary of Updates (Already in the Revised Manuscript)**

* **Reframing the Method**
  * **Reframing**: We have reframed the method from "guided self-improvement" to `Feedback-Driven Self-Training`. This better captures the essence of using the VLM's own outputs, refined by grounding feedback, to train itself.
  * **Clarification**: Our primary contribution is a framework for amortizing external spatial feedback into generative vision-language models. We demonstrate that leveraging feedback from a grounding model significantly improves vision-language alignment without requiring additional human annotation or external multimodal data.

* **Distinction from Knowledge Distillation**
  * We think SVP can be interpreted as a form of Alignment Distillation. Unlike traditional distillation, where a student mimics a teacher's generated tokens/distribution, SVP uses a grounding model as a verifier or guide to correct the VLM's own spatial reasoning. The VLM self-generates the initial description, the input for the grounding model, and the final grounded description.
  * There is no “generative teacher” providing correct or aligned captions/descriptions. However, we acknowledged the connection with alignment distillation in `[section 1, line  117-120]`. Using the teacher-student analogy common in distillation:
    * Knowledge Distillation: Teacher gives the answer (Tokens). Student copies.
    * SVP: Teacher checks the work (Grounding). Student fixes its own answer.

* **On Grounded-driven Data Augmentation**
  * While SVP shares superficial similarities with data augmentation, we argue that it is fundamentally distinct because all textual supervision is self-generated. The pipeline operates as a closed loop: the VLM generates a draft, the grounding model identifies spatial discrepancies, and the VLM uses this feedback to self-correct and generate a refined caption. Crucially, the model is then adapted using only these improved captions, without access to bounding boxes or external labels.
  * Unlike standard augmentation, which typically injects external noise or data, SVP extracts latent visual information already present in the image but initially missed by the model. Therefore, we believe Feedback-Driven Self-Training or Self-Correction is a more accurate framing of our contribution.

* **Method Clarity**
  * `[Section 3.1, Figure 5]` We restructured `Section 3.1` by adding a Symbol Table, a Simplified Algorithm block, and explicit Input/Output definitions and examples. We also rewrote the caption of `Figure 5` to better trace the data flow.

| Model                | Dataset Size | F1 (all) | F1 (adv) | F1 (pop) | F1 (rand) | Recall |
|----------------------|-------------------------|----------|----------|----------|-----------|--------|
| Qwen2.5-VL-7b        | >> 100M                 | 86.2     | **85.1**     | 86.1     | 87.1      | 77.9   |
| LLaVA-1.6-7b w/ SVP (ours)  | ~ 800K                  | **88.3**     | **85.9**     | **89.1**     | **90.1**      | **84.4**   |
|


* **New Experiments - Qwen-VL and SVP**
  * `[Section 4, Table 7, Table 8]` To address concerns about generality and relevance to modern models, we applied SVP to the recent Qwen2.5-VL with a focus on hallucination reduction and object recall.
  * **Experiment 1**. `[Section 4, Table 7]` Fine-tuning on Large Datasets -  Qwen2.5-VL vs LLaVA-1.6 w/ SVP: Simply scaling pre-training and post-training data is not enough to fix vision-language alignment (see also Table above). As shown in Table the base Qwen2.5-VL (trained on massive SFT data) still hallucinates more than our LLaVA-1.6-7B adapted with SVP. Even though Qwen2.5-VL has seen orders of magnitude more data, it still exhibits lower object recall than the much smaller LLaVA-1.6 adapted via SVP.

  * **Experiment 2**. `[Section 4, Table 8]` SVP on Stronger Models - Qwen2.5-VL vs Qwen2.5-VL w/ SVP (single iteration): LLaVA-1.6 adapted with SVP outperforms the much larger Qwen2.5-VL base on hallucination metrics. Furthermore, applying SVP to Qwen2.5-VL yields reduction in hallucination, showing that our framework is architecture-agnostic and can be effective even on VLMs utilizing large-scale pre-training and post-training datasets.

---

> ### Author Response · Authors · 2025-11-22
> **General Response (ii)**
>
> * **New Experiments: Comparison with Image Textualization**
>   * **Experiment 3**. `[Appendix E.1, Table 10]` We added a comparison with Image Textualization. SVP achieves higher hallucination reduction with significantly better data efficiency (4k samples vs. 10k+ for IT).
>
> * **New Experiments: IoU Ablation**
>   * **Experiment 4**. `[Appendix E.2, Figure 15, Figure 16]` We computed IoU scores for a subset of RefCOCO, to provide a non-textual metric for spatial alignment.
>
> * **New Experiments: Grounding Model Robustness Ablation**
>   * **Experiment 5**. `[Appendix E.3, Figure 17]`  To address questions on grounding dependency, we run SVP with: (a) a different grounding model architecture (GDINO-base), and (b) a "noisy" grounder (randomly dropping boxes) to test resilience.
>
>
> * **New Experiments: Generated Sample per Image (K) Ablation**
>   * **Experiment 6**. `[Appendix E.4, Figure 18]`  We studied the effect (over 4 datasets) of the number of samples K generated during the sampling loop, using K = [0, 1, 5, 10, 20].
>
> * **Expanded Experiment: Inference-Time Scaling variant for SVP (iSVP)**
>   * **Experiment 7**. `[Appendix C, Table 9]` We rewrote `Appendix C` to emphasize iSVP, demonstrating the framework's flexibility as an inference-time scaling mechanism without parameter tuning and leveraging in-context learning.
>
> * **Grounding Integration**
>   * `[Section 1, line 114-120]` We have added a paragraph clarifying that the grounding output is fed to the VLM without post-processing, a design choice intended to ensure objectivity and minimize engineered bias. Our integration strategy is intentionally minimalist: the VLM's raw output caption is passed directly to Grounding DINO, with no special prompting, pre-processing, or manual filtering. This approach mitigates human bias and ensures the system remains fully automated.
>
> * **SVP Efficiency**
>   * Compared to standard post-training paradigms (such as SFT, DPO, or RLHF) and inference-time scaling methods, SVP is computationally lightweight. As a targeted refinement strategy, it operates on a vastly smaller scale than full fine-tuning. Specifically, SVP requires adapting the model on only ~1,000 images for a limited number of epochs using LoRA - a process that completes in minutes on a single A100 GPU. When contrasted with expensive post-training methods based, SVP represents a negligible cost for substantial improvements in model reliability.

---

### Author Response · Authors · 2025-11-25
**Summary**

Dear Reviewers,

We thank you for your insightful and constructive feedback. We have uploaded a revised manuscript that we believe effectively addresses the concerns raised, specifically regarding method scoping, clarity, and generality. We welcome any further feedback during the discussion period.

-----
### **Key Updates in the Revised Manuscript**


**Reframing**
- We reframed the method as `Feedback-Driven Self-Training` to better scope our contribution.

**Clarification**
- We clarified the distinction with knowledge distillation (`[line 114-120]`) and data augmentation.

**Method**
- `[Section 3]` We improved the `Method` clarity and structure:
     * New `Algorithm 1` for SVP.
     * New notation `Table 1` with examples.
     * Improved `Figure 5` caption.
     * Improved overall structure of this section.

**Experiments**
- We added `7 additional experiments` as suggested by the Reviewers:

    * `[Section 4, Table 7]` **Qwen2-VL/Qwen2.5-VL** vs LLaVA-1.6 w/ SVP
    * `[Section 4, Table 8]` **Qwen2.5-VL w/ SVP** for hallucination reduction
    * `[Appendix E.1, Table 10]` **Image Textualization** and large SFT fine-tuning Ablation
    * `[Appendix E.2, Figure 15, Figure 16]` **IoU** Ablation
    * `[Appendix E.3, Figure 17]` **Grounding Model** Robustness Ablation
    * `[Appendix E.4, Figure 18]` Generated Samples per Image (**K**) Ablation
    * `[Appendix C, Table 9]` Inference-Time Scaling variant for SVP (**iSVP**)

**Details**
- We explained how grounding is integrated in SVP (`[line 382-384]`) and discussed SVP efficiency.

-----

*We believe this revision effectively solves most of the issues raised by the Reviewers and greatly improves, scopes, and clarifies our work. `The revised paper better highlights our methodological contributions and demonstrates the breadth and depth of our experimental results`.*

---

### Author Response · Authors · 2025-11-26
**Detailed Summary of Revisions**

### **Summary of Revisions: New Baselines, Ablations, and Method Generalization**

Dear Area Chair and Reviewers,

We thank the reviewers for their insightful feedback. **We have significantly revised the paper to address specific concerns regarding method scoping, experimental baselines, and clarity**.

------

Below is a **point-by-point summary of how our new experiments map to the issues raised by each Reviewer**:

* **Generality on Modern SOTA Models (Addressing Reviewer `nkcU`'s request for Qwen-VL)**:
  * To prove SVP applies to modern architectures, we applied it to `Qwen2.5-VL`.
  * Results show that SVP-adapted LLaVA (7B) outperforms the much larger Qwen2.5-VL (7B) on hallucination reduction metrics.
  * Applying SVP to Qwen2.5 yields further gains (`Section 4`, `Tables 7` and `Table 8`).

* **Comparison with Synthetic Data SFT (Addressing Reviewer `hcG4`'s query on Image Textualization)**:
  * To demonstrate data efficiency compared to standard SFT on synthetic data, we compared SVP against `Image Textualization`.
  * SVP achieves higher alignment (88.3 vs 86.4 F1) using less data (4k vs 10k/50k samples), validating the specific benefit of self-correction over external data fitting (`Appendix E.1`).

* **Sampling Efficiency ($K$ Ablation) (Addressing Reviewers `tCTj` and `hcG4`’s concerns on compute cost)**:
  * To quantify the computational trade-off, we ran ablations on the number of `generated samples per image` ($K \in [1, 5, 10, 20]$).
   * Results in `Appendix E.4` demonstrate that SVP is effective even with lower generated sample counts.

* **Spatial Alignment Metrics (Addressing Reviewer `CBKU`’s request for IoU)**:
  * To validate spatial precision beyond text metrics, we computed `IoU scores` on RefCOCO.
   * This confirms that text-based SVP improvements correlate directly with improved spatial localization (`Appendix E.2`).

* **Robustness to Grounding Feedback (Addressing Reviewers `tCTj` & `hcG4`’s concern on grounder dependency)**:
  * To show that SVP does not rely on an "oracle," we added ablations with "noisy" grounding models and different architectures (GDINO-tiny and GDINO-base).
  * SVP remains effective even with degraded feedback, proving robustness (`Appendix E.3`).

* **Inference-Time Adaptation (Addressing Reviewer `tCTj`’s comment on training necessity)**:
  * We formalized Inference-Time SVP (`iSVP`) to demonstrate that the SVP strategy can work without parameter updates (`Appendix C`).

* **Method Reframing and Clarity (Addressing Reviewers `tCTj` and `nkcU` on distillation and structure)**:
  * We reframed the contribution as `Feedback-Driven Self-Training` to clearly distinguish it from knowledge distillation and better scope our specific application (vision-language alignment) compared to generic guided self-improvement.
  * We also rewrote the beginning of `Section 3`, rewriting the caption for `Figure 5`, adding `Algorithm 1` and `Table 1`, collecting explicit symbol definitions and examples for each SVP components to streamline and clarify our method.
----

**We believe this comprehensive revision effectively resolves the reviewers' concerns regarding generality, efficiency, and clarity, significantly strengthening the manuscript**. We welcome any further feedback during the discussion period.

Best regards,

The Authors

---

### Author Response · Authors · 2025-11-29
**Summary of Revisions for Area Chair: Addressing Critiques with 7 New Experiments and Method Generalization**

Dear Area Chair,

First, we would like to thank the reviewers for the preliminary feedback. However, as the reviewers were unable to assess our rebuttal and revised manuscript due to the abrupt changes in the conference review schedule, we understand the current scores reflect the original submission rather than our revised manuscript.

While the initial reviews were constructive, they identified specific gaps in *experimental validation* and *areas where the methodological framing required clarification*.

**We believe  that our extensive revisions and [7 new experiments](https://openreview.net/forum?id=cu8z4V1nJt&noteId=WoJeUYhlqv)  directly close these gaps**.

Please see the [General Response (i)](https://openreview.net/forum?id=cu8z4V1nJt&noteId=9QPK3XP36R) and [General Response (ii)](https://openreview.net/forum?id=cu8z4V1nJt&noteId=Cmi0qso83K) for more details.

----

### **Summary of Revisions**

Here we provide a short summary of the core experiments and changes developed during the rebuttal period. Please see the [Detailed Summary of Revisions](https://openreview.net/forum?id=cu8z4V1nJt&noteId=WoJeUYhlqv) and [General Response](https://openreview.net/forum?id=cu8z4V1nJt&noteId=9QPK3XP36R) for more details.


**Addressing Generality** (Reviewer `nkcU`) - Table 7, Table 8

* **Critique**: Validation was limited to LLaVA architectures.
* **Experiment**: We applied SVP to the state-of-the-art Qwen2.5-VL.
* **Result**: SVP-adapted LLaVA (7B) outperforms the base Qwen2.5-VL (7B) on hallucination reduction (F1 88.3 vs 86.2), proving the method remains superior to simple scaling. Qwen2.5-VL w/ SVP reduces hallucinations (F1 89.1 vs 86.2) and improves object recall (83.7 vs 77.9) compared to Qwen2.5-VL.

**Addressing Baselines and Efficiency** (Reviewer `hcG4`) - Appendix E.1

* **Critique**: Lack of comparison against synthetic data baselines.
* **Experiment**: We benchmarked SVP against [Image Textualization](https://arxiv.org/abs/2406.07502).
* **Result**: SVP achieves higher alignment (88.3 vs 86.4 F1) using less data (4k vs 10k+ samples), validating that SVP is an effective alignment mechanism compared to generic SFT on synthetic data.

**Addressing Spatial Validation** (Reviewer `CBKU`) - Appendix E.2

* **Critique**: Request for explicit localization metrics (IoU) to substantiate spatial alignment claims.
* **Experiment**: We extended the evaluation to include IoU scores on RefCOCO benchmarks.
* **Result**: The analysis confirms that SVP improves spatial localization (IoU) alongside text generation, validating that the SVP feedback loop effectively refines the model's actual spatial understanding.


**Addressing Grounding Dependency** (Reviewers `tCTj` and `hcG4`) - Appendix E.3

* **Critique**: Concern that the method relies on perfect grounding feedback.
* **Experiment**: We conducted ablations with noisy grounders and different architectures (GDINO-tiny, GDINO-base).
* **Result**: The method remains effective even with degraded feedback, confirming that SVP relies on consistent spatial signals rather than requiring a perfect oracle.

**Addressing Method Framing** (Reviewers `tCTj` and `nkcU`) - Method

* **Critique**: Ambiguity regarding method scoping and distinction from distillation.
* **Revision**: We reframed the contribution as `Feedback-Driven Self-Training` and added `Algorithm 1` and `Table 1` to improve clarity.
* **Result**: We clarify that unlike distillation (which mimics a teacher's tokens), SVP uses grounding strictly as a verifier (for vision-language alignment) to guide the model's own generation distribution.

------

### **Final Remarks**

**We believe these comprehensive updates effectively resolve the reviewers' reservations**. `Feedback-Driven Vision-Language Alignment` represents a critical yet under-explored frontier in multimodal learning. SVP advances this direction by demonstrating a scalable mechanism to harness model self-generation and feedback for alignment, offering significant gains without the need for additional human annotation.


Best regards,

The Authors

---

### Meta-Review · Area_Chair_WnnK · 2025-12-29

**Summary:**

This paper studies an important problem, and shows solid experiments for a variety of tasks and model scales. The reviewer scores prior to the author response are unanimously below the acceptance threshold.

I agree with the reviewers about the method's weakness of relying on an accurate grounding model. Since the primary comparisons do not have access to this model (or equivalent level of supervision, like the training data of the grounding model), they are not apples-to-apples comparisons. While some of the author reviewers help provide other information, the core concern remains. The revised framing as "feedback-driven self-training" only seems further confusing & misleading, since self-training should generally only refer to scenarios where  only the model itself is used, rather than separate models or supervision.

Lack of clarity was also a significant concern, and it is difficult to judge if this concern is addressed without extensive further reviewing. Though, it appears that only the algorithm box and figures have been modified, with no modifications to the text of the methods section.

I think that the new experiments significantly improve the paper, and that the paper has a lot of promise, but given the low initial scores and the above remaining concerns, I recommend reject. I encourage the authors to revise the paper and resubmit it to a future venue with revisions for text clarity and core comparisons that leverage the extra information from the grounding model in different ways.

**Reviewer Concerns:**

I believe many of the reviewer concerns have been addressed, including:

- the concern about the derivation in appendix F because the entire section was removed.
- showing multiple grounding models (though an entirely different model would be better)
- using other VLM models
- the questions about noun list and object instance ambiguation
- the comparison to image textualization

I believe some concerns are outstanding, which are described above.

**Reviewer Scores:**

I expect some reviewers would have increased their score, but not all reviewers.

---

### Decision · Program_Chairs · 2026-01-26

Reject